# Theoretical guarantees in KL for Diffusion Flow Matching

**Marta Gentiloni Silveri**
École polytechnique
Route de Saclay, 91120 Palaiseau, France
marta.gentiloni-silveri@polytechnique.edu

**Giovanni Conforti**
Università degli Studi di Padova
Via Trieste, 63, 35131 Padova, Italia
giovanni.conforti@math.unipd.it

**Alain Durmus**
École polytechnique
Route de Saclay, 91120 Palaiseau, France
alain.durmus@polytechnique.edu

## Abstract

Flow Matching (FM) (also referred to as stochastic interpolants or rectified flows) stands out as a class of generative models that aims to bridge in finite time the target distribution $\nu^\star$ with an auxiliary distribution $\mu$, leveraging a fixed coupling $\pi$ and a bridge which can either be deterministic or stochastic. These two ingredients define a path measure which can then be approximated by learning the drift of its Markovian projection. The main contribution of this paper is to provide relatively mild assumptions on $\nu^\star$, $\mu$ and $\pi$ to obtain non-asymptotics guarantees for Diffusion Flow Matching (DFM) models using as bridge the conditional distribution associated with the Brownian motion. More precisely, we establish bounds on the Kullback-Leibler divergence between the target distribution and the one generated by such DFM models under moment conditions on the score of $\nu^\star$, $\mu$ and $\pi$, and a standard $L^2$-drift-approximation error assumption.

## 1   Introduction

A significant task in statistics and machine learning currently revolves around generating samples from a target distribution that is only accessible via a dataset. To tackle this challenge, generative models have become prominent as effective computational tools for learning to simulate new data. Essentially, these models involve learning a *generator* capable of mapping a source distribution into new approximate samples from the target distribution.

One of the most productive approaches to generative modeling is based on deterministic and stochastic transport dynamics, that connect a target distribution with a base distribution. Typically, the target distribution represents the data set from which we want to generate new samples, while the base distribution is one that can be easily simulated or sampled. Regarding the dynamics, they correspond to SDEs Stochastic Differential Equations (SDEs) or Ordinary Differential Equations (ODEs), where the drift (for SDEs) or velocity field (for ODEs) is determined by solving a regression problem. This regression problem is usually addressed with appropriate neural networks and related training techniques [SDWMG15a, OFLR21, FJNO20, DBMP19, CLT22, DBTHD21, SE19, LYB+23, GCBD19].

Among these methods, score-base generative models (SGMs) and in particular diffusion models based on score matching [SDWMG15b, HJA20, SE20, SE19] was an important milestone. In a nutshell, these models involve transforming an arbitrary density into a standard Gaussian model and consists in learning the drift of the corresponding reversal process. More precisely, the idea is to first consider

an Ornstein-Uhlenbeck (OU) process $(X_t^{\mathrm{OU}})_{t\in[0,T]}$, over a time interval $[0,T]$,

$$\mathrm{d}X_t^{\mathrm{OU}} = -(1/2)X_t^{\mathrm{OU}}\mathrm{d}t + \mathrm{d}B_t\ , \quad X_0^{\mathrm{OU}} \sim \nu^\star\ , \tag{1}$$

where $(B_t)_{t\geqslant 0}$ is a $d$-dimensional Brownian motion. Then, the reversal process $(\overleftarrow{X}_t^{\mathrm{OU}})_{t\in[0,T]}$, which is defined from the non-homogeneous SDE [And82]:

$$\mathrm{d}\overleftarrow{X}_t^{\mathrm{OU}} = \{(1/2)\overleftarrow{X}_t^{\mathrm{OU}} + \nabla\log p_{T-t}^{\mathrm{OU}}(\overleftarrow{X}_t^{\mathrm{OU}})\}\mathrm{d}t + \mathrm{d}B_t\ , \quad t\in[0,T] \quad , \text{ with } \overleftarrow{X}_0^{\mathrm{OU}} \sim \nu^\star P_T^{\mathrm{OU}}\ , \tag{2}$$

allows the law of $X_T^{\mathrm{OU}}$ to be transported to $\nu^\star$: [And82, Equations 3.11, 3.12] show that $\overleftarrow{X}_T^{\mathrm{OU}}$ has distribution $\nu^\star$. The initialization $\nu^\star P_T^{\mathrm{OU}}$ in (2) is the distribution of $X_T^{\mathrm{OU}}$ defined in (1). The drift in (2) can be decomposed as the sum of a linear function and the score associated with the density of $X_{T-t}^{\mathrm{OU}}$ with respect to the Lebesgue measure, denoted by $p_{T-t}^{\mathrm{OU}}$. From the Tweedie identity, this score is the solution to a regression problem that can be solved efficiently by score matching [HD05, Hyv05, Vin11]. Learning this score at different times can also be formulated as a sequence of denoising problems. Once the drift of the reversal process is learned or equivalently the scores $(\nabla\log p_t^{\mathrm{OU}})_{t\in[0,T]}$, score-based generative models consist in following the reversal dynamics over $[0,T]$ or, more commonly, a discretization of it, starting with a sample from $\mathrm{N}(0,\mathrm{Id})$. The final sample at time $T$ is then approximatively distributed according to $\nu^\star$. Note that an approximation is made even if the reversal dynamics were simulated exactly, because for full accuracy, the model would need to start from a sample of $\nu^\star P_T^{\mathrm{OU}}$. However, it is well known that for sufficiently large $T$, $\nu^\star P_T^{\mathrm{OU}}$ is (exponentially) close to $\mathrm{N}(0,\mathrm{Id})$.

When exploring diffusion models, it has been realized that the generation of approximate data samples could also be achieved using an ODE instead of the reversal diffusion:

$$\mathrm{d}\overleftarrow{\mathbf{x}}_t^{\mathrm{OU}}/\mathrm{d}t = (1/2)(\overleftarrow{\mathbf{x}}_t^{\mathrm{OU}} + \nabla\log p_{T-t}^{\mathrm{OU}}(\overleftarrow{\mathbf{x}}_t^{\mathrm{OU}}))\ .$$

Similarly to the drift of the reversal diffusion, the velocity fields at time $t\in[0,T]$ associated with this ODE is the sum of a linear function and the score of the density of $X_{T-t}^{\mathrm{OU}}$. This observation has prompted the introduction of the Probability Flow ODE implementation of diffusion models [CCL$^+$23a]. SGMs in their standard and probability flow ODE implementations have achieved notable success in a range of applications; see e.g., [RBL$^+$22, RDN$^+$22, PVG$^+$21].

While diffusion-based methods have now become popular generative models, they can suffer from two limitations. First, there is a trade-off in selecting the time horizon $T$ and second, they rely solely on Gaussian distributions as base distributions, in general. Therefore, there remains considerable interest in developing methods that consider a more general base distribution $\mu$ and that accomplish the transport between $\nu^\star$ and $\mu$ relying on dynamics defined on a fixed finite time interval. Defining a generative process in finite time by means of a coupling and an interpolating process, [Pel22], laid the foundation for Flow Matching (FM) models [AVE22, ABVE23, LCBH$^+$23, Liu22, LGL23], finally addressing this problem.

In its simplest form, the main strategy employed by FMs to bridge two distributions, involves a fixed coupling $\pi$ between $\nu^\star$ and $\mu$ and the use of a bridge, *i.e.*, a conditional distribution on the path space $\mathrm{C}([0,1],\mathbb{R}^d)$ of a reference process $(R_t)_{t\in[0,1]}$ given its starting point $R_0$ and end point $R_1$. In case $R_t$ is a deterministic function of $R_0$ and $R_1$, we say that the bridge is deterministic and stochastic otherwise. As $(R_t)_{t\in[0,1]}$ corresponds to the solution of a stochastic differential equation, we coin the term Diffusion Flow Matching (DFM) to distinguish the latter case from the former one and focus on it. Then, this bridge and the coupling $\pi$ between $\nu^\star$ and $\mu$ define an interpolated process $(X_t^{\mathrm{I}})_{t\in[0,1]}$, referred to as an interpolant, defined as $(X_0^{\mathrm{I}},X_1^{\mathrm{I}}) \sim \pi$ and $(X_t^{\mathrm{I}})_{t\in[0,1]}$ given $(X_0^{\mathrm{I}},X_1^{\mathrm{I}})$ has the same conditional distribution as $(R_t)_{t\in[0,1]}$ given $(R_0,R_1)$. However, $(X_t^{\mathrm{I}})_{t\in[0,1]}$ does not correspond in general to the distribution of a diffusion or even to the one of a Markov process. This characteristic poses a challenge when dealing with potential stochastic sampling procedures: indeed, similarly to SGMs, FMs and DFMs aim to design a Markov process that approximatively transport $\mu$ to $\nu^\star$. To address this issue, most works proposing FM and DFM models [SBCD23, LWYql23] rely on mimicking the marginal flow of the interpolated process $(X_t^{\mathrm{I}})_{t\in[0,1]}$ through a diffusion process known as the Markovian projection. A remarkable feature of this diffusion lies in the fact that its drift is also a solution of a regression problem that can be approximatively solved using only samples from the interpolant $(X_t^{\mathrm{I}})_{t\in[0,1]}$. Then, an approximate samples from $\nu^\star$ is obtained by following a discretization of the dynamics associated with the considered Markovian projection, starting with a sample of $\mu$.

While there exists now an important literature on theoretical guarantees for SGMs [CDS23, CLL23, CCL$^+$23b, PMM23, LLT23, Bor22a], only a few works have been considering FMs. In addition, up to our knowledge, these works on FMs only consider deterministic interpolants [AVE22, BDD23, GHJZ24]. The main objective of this paper is to fill this gap and to analyze DFMs using the $d$-dimensional Brownian motion as reference process, in which case the bridge is simply the $d$-dimensional standard Brownian bridge. We provide theoretical guarantees, upper bounding the Kullback Leibler divergence between the target distribution and the one resulting from the DFM. Our results consider the two sources of error coming from the DFM model, namely drift-estimation and time-discretization. This pursuit underscores the significance of comprehending and quantifying the factors influencing the performance of DFMs, paving the way for further advancements in generative modeling techniques.

**Our contribution.** In this work, we analyze a DFM model using as bridge the $d$-dimensional Brownian bridge and examine how it performs in two distinct scenarios: one without early-stopping and another with early-stopping. In our first main contribution Theorem 2, we establish an explicit and simple bound on the KL divergence between the data distribution and the distribution at the final time of the DFM model. We achieve our bound without early stopping, by assuming only (1) moment conditions on the target $\nu^\star$ and the base $\mu$ (**H**1); (2) integrability conditions on the scores associated with the data distribution $\nu^\star$, the base distribution $\mu$ and the coupling $\pi$ (**H**2); (3) a L$^2$-drift-approximation error (**H**3) (an assumption commonly considered in previous works). Note that condition (2) implies that $\nu^\star$ necessarily admits a density. We relax this condition in our second contribution. In Theorem 3, we establish an explicit bound on the KL divergence between a smoothed version of the target distribution and an early stopped version of the DFM model, assuming (1) and (3), but replacing the condition (2) by assuming (4) $\pi = \mu \otimes \nu^\star$ and integrability conditions only on the score associated with $\mu$.

To the best of our knowledge, our paper provides the first convergence analysis for diffusion-type FMs, that tackles all the sources of error, *i.e.*, the drift-approximation-error and the time-discretization error. In addition, previous studies concerning FMs and Probability Flow ODEs, with deterministic or mixed sampling procedure, either rely on at least some Lipschitz regularity of the flow velocity field or its estimator and/or do not take the time-discretization error into consideration. Also, in the context of SGMs with constant step-size, most of existing works without early-stopping are obtained assuming either the score (or its estimator) to be Lipschitz or the data distribution to satisfy some additional conditions (e.g., manifold hypothesis, bounded support, etc.); the unique exception being [CDS23]. We refer to Section 3.2 for a more in depth literature comparison.

**Notation.** Given a measurable space $(\mathsf{E}, \mathcal{E})$, we denote by $\mathcal{P}(\mathsf{E})$ the set of probability measures of $\mathsf{E}$. Also, given a topological space $(\mathsf{E}, \tau)$, we use $\mathcal{B}(\mathsf{E})$ to denote the Borel $\sigma$-algebra on $\mathsf{E}$. Given two random variables $Y, \tilde{Y}$, we write $Y \perp\!\!\!\perp \tilde{Y}$ to say that $Y$ and $\tilde{Y}$ are independent. Denote by $(B_t)_{t \in [0,1]}$ a $d$-dimensional Brownian motion. We denote by $\mathrm{Leb}^d$ the Lebesgue measure on $\mathbb{R}^d$. Given two real numbers $u, v \in \mathbb{R}$, we write $u \lesssim v$ (resp. $u \gtrsim v$) to mean $u \leq Cv$ (resp. $u \geq Cv$) for a universal constant $C > 0$. Also, we denote by $\|x\|$ the Euclidean norm of $x \in \mathbb{R}^d$, by $\langle x, y \rangle$ the scalar product between $x, y \in \mathbb{R}^d$, and by $x^\mathrm{T}$ the transpose of $x$. Given a matrix $\mathbf{A} \in \mathbb{R}^{d \times s}$, we denote by $\|\mathbf{A}\|_{\mathrm{op}}$ the operator norm of $\mathbf{A}$. For $f : [0,1] \times \mathbb{R}^d \to \mathbb{R}$ regular enough, we denote by $\nabla_x f(t, x), \nabla_x^2 f(t, x)$ and $\Delta_x f(t, x)$ respectively the gradient, hessian and laplacian of $f$, defined for $t, x \in [0,1] \times \mathbb{R}^d$ by $\nabla_x f(t, x) := (\partial_{x_i} f(t, x))_i, \nabla_x^2 f(t, x) := (\partial_{x_i} \partial_{x_j} f(t, x))_{i,j}, \Delta f(t, x) := \sum_{i=1}^d \partial_{x_i}^2 f(t, x)$, where $\partial_{x_j}$ denotes the partial derivative with respect to the $j$-th variable. For $F : [0,1] \times \mathbb{R}^d \to \mathbb{R}^d$ regular enough, we denote by $D_x F, \mathrm{div}_x F$ and $\Delta_x F$ respectively, the Jacobian matrix, the divergence and the vectorial laplacian of $F$, defined for $t, x \in [0,1] \times \mathbb{R}^d$ by $D_x F(t, x) = (\partial_{x_j} F_i(t, x))_{i,j}$, $\mathrm{div}_x F(t, x) := \sum_{j=1}^d \partial_{x_j} F_j(t, x), \Delta_x F(t, x) = (\Delta_x F_1(t, x), ..., \Delta_x F_d(t, x))$.

## 2 Diffusion Flow Matching.

Given a target distribution $\nu^\star \in \mathcal{P}(\mathbb{R}^d)$ and a base distribution $\mu \in \mathcal{P}(\mathbb{R}^d)$, the idea at the core of FM models is intuitively to construct a path between these two by considering two ingredients (1) a coupling $\pi$ and (2) a bridge (or an interpolant following [AVE22]) between $\mu$ and $\nu^\star$ (or more precisely a bridge with foundations $\pi$). More formally, here we say that $\pi$ is a coupling

between $\mu$ and $\nu^\star$ if for any $\mathsf{A} \in \mathcal{B}(\mathbb{R}^d)$, $\pi(\mathsf{A} \times \mathbb{R}^d) = \mu(\mathsf{A})$ and $\pi(\mathbb{R}^d \times \mathsf{A}) = \nu^\star(\mathsf{A})$, and denote by $\Pi(\mu, \nu^\star)$ the set of coupling between $\mu$ and $\nu^\star$. Then, based on a probability measure on $\mathbb{W} = \mathrm{C}([0,1], \mathbb{R}^d)$ the set of continuous functions from $[0,1]$ to $\mathbb{R}^d$, we define the bridge $\mathrm{b}\mathbb{Q}$ associated with $\mathbb{Q}$ as the Markov kernel $\mathrm{b}\mathbb{Q}$ on $\mathbb{R}^{2d} \times \mathbb{W}$, such that, for any $\mathsf{A} \in \mathcal{B}(\mathbb{W})$, $\mathbb{Q}(\mathsf{A}) = \int \mathbb{Q}_{0,1}\mathrm{d}(x_0, x_1)\mathrm{b}\mathbb{Q}((x_0, x_1), \mathsf{A})$ (see e.g., [Kle13, Theorem 8.37] for the existence of this kernel), where for any $\mathsf{I} = \{t_1, \ldots, t_n\} \subset [0,1]$, $t_1 < \cdots < t_n$, $\mathbb{Q}_\mathsf{I}$ is the $\mathsf{I}$-marginal distribution of $\mathbb{Q}$, $i.e.$, the pushforward measure of $\mathbb{Q}$ by $(x_t)_{t \in [0,1]} \mapsto (x_{t_1}, \ldots, x_{t_n})$. From a probabilistic perspective, it implies that if $(W_t)_{t \in [0,1]} \sim \mathbb{Q}$, then $\mathrm{b}\mathbb{Q}$ is a conditional distribution of $(W_t)_{t \in [0,1]} \sim \mathbb{Q}$ given $(W_0, W_1)$: for any bounded and measurable function on $\mathbb{W}$, $\mathbb{E}[f((W_t)_{t \in [0,1]})|X_0, X_1] = \int f((w_t)_{t \in [0,1]})\mathrm{b}\mathbb{Q}((W_0, W_1), \mathrm{d}(w_t)_{t \in [0,1]})$.

## 2.1 Definition of the interpolated process

Consider now a coupling $\pi$ and a bridge $\mathrm{b}\mathbb{Q}^\beta$ associated to $\mathbb{Q}^\beta \in \mathcal{P}(\mathbb{W})$. We suppose here that $\mathbb{Q}^\beta$ is the distribution of $(Y_t)_{t \in [0,1]}$ solution of the stochastic differential equation

$$\mathrm{d}Y_t = \beta(Y_t)\mathrm{d}t + \sqrt{2}\mathrm{d}B_t, \quad t \in [0,1], \quad Y_0 \sim \mu \in \mathcal{P}(\mathbb{R}^d), \tag{3}$$

where $(B_t)_{t \in \mathbb{R}_+}$ is a standard $d$-dimensional Brownian motion. In addition, we suppose $\beta \in \mathrm{C}^\infty(\mathbb{R}^d, \mathbb{R}^d)$ for simplicity and that (3) admits a unique strong solution. Consider now, the interpolated measure $\mathbb{I}(\pi, \mathbb{Q}^\beta)$[1] corresponding to the distribution of the process defined by $(X_0^\mathsf{I}, X_1^\mathsf{I}) \sim \pi$ and $(X_t^\mathsf{I})_{t \in [0,1]}|(X_0^\mathsf{I}, X_1^\mathsf{I}) \sim \mathrm{b}\mathbb{Q}^\beta((X_0^\mathsf{I}, X_1^\mathsf{I}), \cdot)$. In [ABVE23], $(X_t^\mathsf{I})_{t \in [0,1]}$ is referred to as a stochastic interpolant. Denote by $(s, t, x, y) \mapsto p_{t|s}^Y(y|x)$ the conditional density of $Y_t$ given $Y_s$ with respect to the Lebesgue measure and by $(p_t^\mathsf{I})_{t \in [0,1]}$ the time marginal densities of $(X_t^\mathsf{I})_{t \in [0,1]}$ with respect to the Lebesgue measure, that is $\mathbb{P}(Y_t \in \mathsf{A}|Y_s) = \int_\mathsf{A} p_{t|s}(x|Y_s)\mathrm{d}x$ and $\mathbb{P}(X_t^\mathsf{I} \in \mathsf{A}) = \int_\mathsf{A} p_t^\mathsf{I}(x)\mathrm{d}x$, for $\mathsf{A} \in \mathcal{B}(\mathbb{R}^d)$ and $s, t \in [0,1]$, $s \neq t$. Then, note that, as a straightforward consequence of the definition of $(X_t^\mathsf{I})_{t \in [0,1]}$, it holds

$$p_t^\mathsf{I}(x) = \int_{\mathbb{R}^{2d}} p_{t|0}^Y(x|x_0)p_{1|t}^Y(x_1|x)\tilde{\pi}(\mathrm{d}x_0, \mathrm{d}x_1), \tag{4}$$

where

$$\tilde{\pi}(\mathrm{d}x_0, \mathrm{d}x_1) = \frac{\pi(\mathrm{d}x_0, \mathrm{d}x_1)}{p_{1|0}^Y(x_1|x_0)}.$$

An example that we will focus on in this paper is $\mathbb{Q}^\beta = \mathbb{B}$ the distribution of the Brownian motion $(\sqrt{2}B_t)_{t \in \mathbb{R}}$ solution of (3) with $\beta \equiv 0$. Then, it is well known that $\mathrm{b}\mathbb{B}$ is then the Markov kernel associated with the Brownian bridge and the resulting stochastic interpolant satisfies for any $t \in [0,1]$

$$X_t^\mathsf{I} \stackrel{\mathrm{dist}}{=} (1-t)X_0^\mathsf{I} + tX_1^\mathsf{I} + \sqrt{2t(1-t)}Z, \quad Z \sim \mathrm{N}(0, \mathrm{Id}), \tag{5}$$

where $\stackrel{\mathrm{dist}}{=}$ denotes the equality in distribution.

It is well-known that for any $x_0, x_1 \in \mathbb{R}^d$, the distribution $\mathrm{b}\mathbb{Q}^\beta((x_0, x_1), \cdot)$ is diffusion-like under appropriate conditions. More precisely, $\mathrm{b}\mathbb{Q}^\beta((x_0, x_1), \cdot)$ is the distribution of $(\mathbf{Y}_t)_{t \in [0,1]}$ solution to

$$\mathrm{d}\mathbf{Y}_t = \{\beta(\mathbf{Y}_t) + 2\nabla\phi_t^{x_1}(\mathbf{Y}_t)\}\mathrm{d}t + \sqrt{2}\mathrm{d}B_t, \quad t \in [0,1], \quad \mathbf{Y}_0 = x_0, \tag{6}$$

where $\phi_t^{x_1}(y) = \log p_{1|t}^Y(x_1|y)$. For instance, for $x_0, x_1 \in \mathbb{R}^d$, the bridge $\mathrm{b}\mathbb{B}((x_0, x_1), \cdot)$ associated to $\mathbb{B}$ is the distribution of a Brownian bridge $(\bar{B}_t^{x_0, x_1})_{t \in [0,1]}$ solution to the SDE

$$\mathrm{d}\bar{B}_t^{x_0, x_1} = \frac{x_1 - \bar{B}_t^{x_0, x_1}}{1-t}\mathrm{d}t + \sqrt{2}\mathrm{d}B_t, \quad t \in [0,1], \quad \bar{B}_0^{x_0, x_1} = x_0.$$

From (6), it turns out that $(X_t^\mathsf{I})_{t \in [0,1]}$ given $(X_0^\mathsf{I}, X_1^\mathsf{I})$ is therefore solution to

$$\mathrm{d}X_t^\mathsf{I} = \{\beta(X_t^\mathsf{I}) + 2\nabla\phi_t^{X_1^\mathsf{I}}(X_t^\mathsf{I})\}\mathrm{d}t + \sqrt{2}\mathrm{d}B_t, \quad t \in [0,1], \quad X_0^\mathsf{I} \sim \mu. \tag{7}$$

---

[1]It corresponds to $\pi \otimes \mathrm{b}\mathbb{Q}^\beta$ the tensor product between $\pi$ and $\mathrm{b}\mathbb{Q}^\beta$: $\mathbb{I}(\pi, \mathbb{Q}^\beta)(\mathsf{A}) = \int \mathrm{b}\mathbb{Q}^\beta((x_0, x_1), \mathsf{A})\mathrm{d}\pi(x_0, x_1)$, $\mathsf{A} \in \mathcal{B}(\mathbb{R}^d)$. In other words, $\pi$-mixture of $\mathbb{Q}^\beta$-bridges.

Note that the drift coefficient in (7) depends on $X_1^I$, and therefore, $(X_t^I)_{t \in [0,1]}$ is not Markov, which is a natural property if we are interested in constructing a generative process. To circumvent this issue, based on $\mathbb{I}(\pi, \mathbb{Q}^\beta)$, we aim to define a distribution $\mathbb{M}(\pi, \mathbb{Q}^\beta)$ such that it has the same one-dimensional time marginals as $\mathbb{I}(\pi, \mathbb{Q}^\beta)$ and corresponds to a diffusion, *i.e.*, if $(X_t^I)_{t \in [0,1]} \sim \mathbb{I}(\pi, \mathbb{Q}^\beta)$ and $(X_t^M)_{t \in [0,1]} \sim \mathbb{M}(\pi, \mathbb{Q}^\beta)$, for any $t \in [0,1]$, $X_t^I \overset{\text{dist}}{=} X_t^M$ and $(X_t^M)_{\in \mathbb{N}}$ is solution of a diffusion with Markov coefficient. This can be done trough the Markovian projection.

## 2.2 Markovian projection and Diffusion Flow Matching

**Markovian projection.** The idea of Markovian projection originally dates back to [Gyö86] and [Kry]. Its main idea is in essence to define a diffusion Markov process which "mimics" the time-marginal of an Itô process:

$$\mathrm{d}\mathbf{X}_t = \mathbf{b}_t \mathrm{d}t + \sqrt{2}\mathrm{d}B_t , \quad t \in [0,1] , \quad \mathbf{X}_0 \sim \mu ,$$

for some adapted process $(\mathbf{b}_t)_{t \in [0,1]}$ and initial distribution $\mu$. Under appropriate conditions (see [BS13, Corollary 3.7]), this diffusion process, denoted by $(\mathbf{X}_t^M)_{t \in [0,1]}$, exists and is solution to an SDE with a (relatively) simple modification of the drift $\mathbf{b}_t$, namely

$$\mathrm{d}\mathbf{X}_t^M = \tilde{\mathbf{b}}_t(\mathbf{X}_t^M)\mathrm{d}t + \sqrt{2}\mathrm{d}B_t , \quad t \in [0,1] , \quad \mathbf{X}_0^M \sim \mu ,$$

where $\tilde{\mathbf{b}}_t(\mathbf{X}_t^M) = \mathbb{E}[\mathbf{b}_t | \mathbf{X}_t^I]$. This result can be applied to the non-Markov process $(X_t^I)_{t \in [0,1]}$ solution of (7): its Markovian projection is solution of

$$\mathrm{d}X_t^M = \tilde{\beta}_t(X_t^M)\mathrm{d}t + \sqrt{2}\mathrm{d}B_t , \quad t \in [0,1] ,$$

for some function $\tilde{\beta} : [0,1] \times \mathbb{R}^d \to \mathbb{R}^d$. It turns out that we can identify $\tilde{\beta}$, relying on the family of conditional densities $(p_{t|s}^Y)_{0 \le s \le t \le 1}$ and marginal densities $(p_t^I)_{0 \le t \le 1}$. This is the content of the following result.

**Theorem 1.** *Consider a* $\pi \in \Pi(\mu, \nu^\star)$ *and* $\mathbb{Q}^\beta$ *associated with* (3). *Consider the drift field*

$$\tilde{\beta}_t^Y(x) = \beta(x) + 2\frac{\int \nabla_x \log p_{1|t}^Y(x_1|x) p_{t|0}^Y(x|x_0) p_{1|t}^Y(x_1|x) \tilde{\pi}(\mathrm{d}x_0, \mathrm{d}x_1)}{p_t^I(x)} . \tag{8}$$

*Under appropriate conditions (see Appendix A.1), the Markov process* $(X_t^M)_{t \in [0,1]}$ *solution of*

$$\mathrm{d}X_t^M = \tilde{\beta}_t^Y(X_t^M)\mathrm{d}t + \sqrt{2}\mathrm{d}B_t , \quad t \in [0,1) , \quad X_0^M \sim \mu , \tag{9}$$

*mimics the one-dimensional time marginals of* $\mathbb{I}(\pi, \mathbb{Q}^\beta)$, *i.e., for any* $t \in [0,1)$, $X_t^I \overset{\text{dist}}{=} X_t^M$.

This result is well-know, but, for sake of completeness, we provide the proof in Appendix A.1. The process (9) is known as the *Markovian projection* of $\mathbb{I}(\pi, \mathbb{Q}^\beta)$, and its drift (8) as *mimicking drift*. In what follows, we denote by $\mathbb{M}(\pi, \mathbb{Q}^\beta)$ the distribution of $(X_t^M)_{t \in [0,1]}$ on $\mathbb{W}$.

*Remark* 1. It can be easily shown by continuity that $X_t^M = X_t^I \Rightarrow X_1^I$ for $t \to 1$, where $\Rightarrow$ denotes the convergence in distribution. As $\text{Law}(X_1^I) = \nu^\star$, the Markovian projection therefore gives a an *ideal* generative model which would consist in following the SDE (9) with initial distribution $\mu$.

*Remark* 2. Note that, because of (4), the mimicking drift (8) rewrites as

$$\tilde{\beta}_t^Y(X_t^I) = \mathbb{E}\Big[2\nabla_x \log p_{1|t}^Y(X_1^I|X_t^I) + \beta(X_t^I)|X_t^I\Big] .$$

**Diffusion Flow Matching.** Eventually, as pointed out in Remark 1, the Markovian projection gives a an *ideal* generative model. However, a) the mimicking drift (8) is intractable and b) the continuous-time SDE (9) can not be numerically simulated. Thus, in order to implement the proposed theoretical idea, we first need to address and overcome the aforementioned computational challenges. To circumvent a), observe that, because of Remark 2 and [Kle13, Corollary 8.17], we can approximate the mimicking drift via solving

$$\min_{\theta \in \Theta} \mathbb{E}\Big[ \big\| s_\theta^Y(t, X_t^I) - \tilde{\beta}_t^Y(X_t^I) \big\|^2 \Big] , \tag{10}$$

for a properly chosen class of neural networks $\{(t,x) \mapsto s_\theta^Y(t,x)\}_{\theta \in \Theta}$, and replace $\tilde{\beta}^Y$ in (9) with $s_{\theta^\star}^Y$, where $\theta^\star \in \Theta$ denotes a minimizer of (10). To deal with b), we simply make use of the Euler-Maruyama scheme, *i.e.*, for a choice of sequence of step sizes $\{h_k\}_{k=1}^N$, $N \geqslant 1$, and the corresponding time discretization $t_k = \sum_{i=1}^k h_i$, such that $t_0 = 0$ and $t_N = 1$, we define the continuous process $(X_t^{\theta^\star})_{t \in [0,1]}$ recursively on the intervals $[t_k, t_{k+1}]$ by

$$\mathrm{d}X_t^{\theta^\star} = s_{\theta^\star}^Y(t_k, X_{t_k}^{\theta^\star})\mathrm{d}t + \sqrt{2}\mathrm{d}B_t , \quad t \in [t_k, t_{k+1}] , \quad \text{with} \quad X_0^{\theta^\star} \sim \mu . \tag{11}$$

(11) is the DFM generative model we are going to analyze.

# 3 Main results

In this section, we provide convergence guarantees in Kullback-Leibler divergence for the Diffusion Flow Matching model (11), under mild assumptions on the $\mu, \nu^\star, \pi$ and $s_{\theta^\star}$, either within a non-early-stopping regime or within a early-stopping regime.

From now on, we consider the case $\beta \equiv 0$, *i.e.*, $\mathbb{Q}^\beta = \mathbb{B}$. We show in Appendix A.2, Remark 9, that, under out set of assumptions, the conditions of Theorem 1 hold for this setup. Moreover, in this case, for any $s, t \in [0,1]$, $s < t$ and $x, y \in \mathbb{R}^d$, the conditional density $p_{t|s}^Y(y|x) \equiv p_{t-s}(y|x)$ where $(t, x, y) \mapsto p_t(y|x)$ is the heat kernel:

$$p_t(y|x) = \frac{1}{(4\pi t)^{d/2}} \exp\left(-\frac{\|y - x\|^2}{4t}\right) , \quad t \in (0, 1] . \tag{12}$$

In the following, we set $\tilde{\beta}^Y \equiv \tilde{\beta}$ and $s_{\theta^\star}^Y \equiv s_{\theta^\star}$.

## 3.1 Convergence Bounds.

We assume moment conditions on the probability measures $\mu$ and $\nu^\star$, and mild integrability conditions on the probability distributions $\mu, \nu^\star$ and on the coupling $\pi$.

For $p \geqslant 1$, we denote for $\zeta \in \mathcal{P}(\mathbb{R}^d)$,

$$\mathbf{m}_p[\zeta] = \int \|x\|^p \, \mathrm{d}\zeta(x) .$$

**H1.** *The probability distributions $\mu, \nu^\star$ satisfy $\mathbf{m}_8[\mu] + \mathbf{m}_8[\nu^\star] < +\infty$.*

**H2.** *The probability distributions $\nu^\star, \mu$ and $\pi$ are absolutely continuous with respect to the Lebesgue measure on $\mathbb{R}^d$ and $\mathbb{R}^{2d}$ respectively, and satisfy*

*(i) The functions $\log \mathrm{d}\mu/\mathrm{dLeb}^d$ and $\log \mathrm{d}\nu^\star/\mathrm{dLeb}^d$ are continuously differentiable and satisfy $\|\nabla \log \nu^\star\|_{\mathrm{L}^8(\nu^\star)}^8 + \|\nabla \log \mu\|_{\mathrm{L}^8(\mu)}^8 < +\infty$ where for $\zeta \in \{\nu^\star, \mu\}$,*

$$\|\nabla \log \zeta\|_{\mathrm{L}^8(\zeta)}^8 = \int \left\|\nabla \log\left(\frac{\mathrm{d}\zeta}{\mathrm{dLeb}^d}\right)(x_0)\right\|^8 \mathrm{d}\zeta(x_0) .$$

*(ii) The function $\log \mathrm{d}\pi/\mathrm{dLeb}^{2d}$ is continuously differentiable and satisfies $\|\nabla \log \tilde{\pi}\|_{\mathrm{L}^8(\pi)}^8 < +\infty$ where*

$$\|\nabla \log \tilde{\pi}\|_{\mathrm{L}^8(\pi)}^8 = \int \|\nabla \log(\tilde{\pi})(x_0, x_1)\|^8 \mathrm{d}\pi(x_0, x_1) , \quad \tilde{\pi}(x_0, x_1) = \frac{1}{p_1(x_1|x_0)} \frac{\mathrm{d}\pi}{\mathrm{dLeb}^{2d}}(x_0, x_1) , \tag{13}$$

*and $p_1$ is defined in (12).*

*Remark* 3. Under **H1**, note that $\|\nabla \log \tilde{\pi}\|_{\mathrm{L}^8(\pi)}^8 < +\infty$ is equivalent by (12) and $\pi \in \Pi(\mu, \nu^\star)$ to

$$\int \|\nabla \log(\mathrm{d}\pi/\mathrm{dLeb}^{2d})(x_0, x_1)\|^8 \mathrm{d}\pi(x_0, x_1) < +\infty .$$

*Remark* 4. We can relax the condition that $\log \mathrm{d}\mu/\mathrm{dLeb}^d$, $\log \mathrm{d}\nu^\star/\mathrm{dLeb}^d$ and $\log \mathrm{d}\pi/\mathrm{dLeb}^{2d}$ are continuously differentiable assuming that $\sqrt[8]{\mathrm{d}\mu/\mathrm{dLeb}^d}$, $\sqrt[8]{\mathrm{d}\nu^\star/\mathrm{dLeb}^d}$ and $\sqrt[8]{\mathrm{d}\pi/\mathrm{dLeb}^{2d}}$ belongs to some Sobolev space, but, for ease of presentation, we prefer not to delve into these technical details.

Moreover, we assume to have estimated the mimicking drift with an $\varepsilon^2$-precision, for some $\varepsilon^2 > 0$ sufficiently small.

**H3.** *There exist $\theta^\star \in \Theta$ and $\varepsilon^2 > 0$ such that*

$$\sum_{k=0}^{N-1} h_{k+1} \mathbb{E}\left[ \left\| s_{\theta^\star}(t_k, X_{t_k}^{\mathrm{M}}) - \tilde{\beta}_{t_k}(X_{t_k}^{\mathrm{M}}) \right\|^2 \right] \leq \varepsilon^2 .$$

*Remark* 5. To be coherent with the previous section, observe that, as a consequence of Theorem 1, for any $k = 0, \cdots, N$, it holds

$$\mathbb{E}\left[ \left\| s_{\theta^\star}(t_k, X_{t_k}^{\mathrm{M}}) - \tilde{\beta}_{t_k}(X_{t_k}^{\mathrm{M}}) \right\|^2 \right] = \mathbb{E}\left[ \left\| s_{\theta^\star}(t_k, X_{t_k}^{\mathrm{I}}) - \tilde{\beta}_{t_k}(X_{t_k}^{\mathrm{I}}) \right\|^2 \right] .$$

Under such assumptions, we derive an upper bound on the KL divergence between the data distribution $\nu^\star$ and the output of the DFM (11):

**Theorem 2.** *Consider a uniform partition of $[0,1]$ with a constant stepsize $h_k \equiv h$, $h = 1/N_h > 0$, for $N_h \in \mathbb{N}^*$ and consider the corresponding process $(X_t^{\theta^\star})_{t\in[0,1]}$ defined in (11). Assume **H**1 to 3. Denoting by $\nu_1^{\theta^\star}$ the distribution of $X_1^{\theta^\star}$, we have that*

$$\mathrm{KL}(\nu^\star | \nu_1^{\theta^\star}) \lesssim \varepsilon^2 + h(h^{1/8} + 1)\Big( d^4 + \mathbf{m}_8[\mu] + \mathbf{m}_8[\nu^\star] + \|\nabla \log \tilde{\pi}\|_{\mathrm{L}^8(\pi)}^8$$

$$+ \|\nabla \log \mu\|_{\mathrm{L}^8(\mu)}^8 + \|\nabla \log \nu^\star\|_{\mathrm{L}^8(\nu^\star)}^8 \Big) . \quad (14)$$

*Remark* 6. Under almost the same conditions as Theorem 2, *i.e.*, **H**1 **H**2, **H3**, replacing $\tilde{\pi}$ in (13) by

$$\tilde{\pi}_T(x_0, x_1) = \frac{1}{p_T(x_1|x_0)} \frac{\mathrm{d}\pi}{\mathrm{dLeb}^{2d}}(x_0, x_1) , \quad (15)$$

our proofs apply also to DFM using a time horizon $T > 0$ and the Brownian bridge on $[0, T]$. In particular, we would have obtained similar bounds as the ones derived in Theorem 2 but with a factor $\max(1, T^8)$ in front of the second addend.

*Remark* 7. Choosing, in Theorem 2,

$$N_h = \frac{d^4 + \mathbf{m}_8[\mu] + \mathbf{m}_8[\nu^\star] + \|\nabla \log \tilde{\pi}\|_{\mathrm{L}^8(\pi)}^8 + \|\nabla \log \mu\|_{\mathrm{L}^8(\mu)}^8 + \|\nabla \log \nu^\star\|_{\mathrm{L}^8(\nu^\star)}^8}{\varepsilon^2}$$

makes the approximation error of order $\mathcal{O}(\varepsilon^2)$ and the complexity of order $\mathcal{O}(\varepsilon^{-2})$.

Using an early-stopping procedure, we obtain in the case $\pi$ is the independent coupling:

**Theorem 3.** *Fix $0 < \delta < 1/2$. Consider a uniform partition of $[0,1]$ with a constant stepsize $h_k \equiv h$, $h = 1/N_h > 0$, for $N_h \in \mathbb{N}^*$ and consider the corresponding process $(X_t^{\theta^\star})_{t\in[0,1]}$ defined in (11). Assume **H**1, **H3** and $\pi = \mu \otimes \nu^\star$ to be the independent coupling. Suppose in addition that $\mu$ is absolutely continuously with respect to the Lebesgue measure, $\log \mathrm{d}\mu/\mathrm{dLeb}^d$ is continuously differentiable and satisfies $\|\nabla \log \mu\|_{\mathrm{L}^8(\mu)}^8 < +\infty$.*

*Then, denoting by $\nu_{1-\delta}^\star$ and $\nu_{1-\delta}^{\theta^\star}$ the distribution of $X_{1-\delta}^{\mathrm{M}}$ and $X_{1-\delta}^{\theta^\star}$ respectively, we have that*

$$\mathrm{KL}(\nu_{1-\delta}^\star | \nu_{1-\delta}^{\theta^\star}) \lesssim \varepsilon^2 + h(h^{1/8} + 1)\Big( \frac{d^4}{\delta^4} + \mathbf{m}_8[\mu] + \mathbf{m}_8[\nu^\star]\frac{1}{\delta^8} + \|\nabla \log \mu\|_{\mathrm{L}^8(\mu)}^8 \Big) .$$

**Corollary 1.** *Fix $\delta = \mathcal{O}(\sqrt{\varepsilon})$. Consider a uniform partition of $[0,1]$ with a constant stepsize $h = \mathcal{O}(\min(\varepsilon^4/d^4, \varepsilon^6))$, and consider the corresponding process $(X_t^{\theta^\star})_{t\in[0,1]}$ defined in (11). Assume **H**1, **H3** and $\pi = \mu \otimes \nu^\star$ to be the independent coupling. Suppose in addition that $\mu$*

is absolutely continuously with respect to the Lebesgue measure, $\log \mathrm{d}\mu/\mathrm{dLeb}^d$ is continuously differentiable and satisfies $\|\nabla \log \mu\|^8_{\mathrm{L}^s(\mu)} < +\infty$.

Then, denoting by $\nu^\star_{1-\delta}$ and $\nu^{\theta^\star}_{1-\delta}$ the distribution of $X^{\mathrm{M}}_{1-\delta}$ and $X^{\theta^\star}_{1-\delta}$ respectively, we have that

$$\mathscr{W}^2_{2,FM}(\nu^\star_{1-\delta}|\nu^{\theta^\star}_{1-\delta}) \lesssim \varepsilon^2 \ ,$$

where $\mathcal{W}_{2,FM}$ denotes the Fortet- Mourier distance of order 2, i.e.

$$\mathscr{W}^2_{2,FM}(\mu,\nu) = \inf_{\pi \in \Pi(\mu,\nu)} \int \min\{\|x-y\|^2, 1\} \mathrm{d}\pi(x,y) \ .$$

## 3.2 Related works and comparison with existing literature.

FMs stand at the forefront of innovation in generative modeling, offering a practical solution to the longstanding challenge of bridging two arbitrary distributions within a finite time interval. Their consequent immense potential has prompted substantial research efforts aimed at providing a theoretical explanation for their effectiveness and has put SGMs and Probability Flow ODEs all in perspective. In this section we report and discuss previous researches on FMs, SGMs and Probability Flow ODEs with the purpose of highlighting the links and differences with our work and contextualizing our contribution.

**Non-early-stopping setting.** In the context of FMs, [AVE22] and [BDD23] seek convergence guarantees in 2-Wasserstein distance. Both consider more general designs for the stochastic interpolant and a deterministic sampling procedure, rather than a stochastic one. However, both works rely on some regularity condition on the approximated flow velocity filed, *i.e.*, that it is Lipschitz. Namely, [AVE22] works under a $K$-Lipschitz (uniform in time and space) assumption on the estimator of the exact flow velocity field. How such assumption pertains to the flow matching framework for generative modeling is not articulated and remains unclear. On the other hand, [BDD23] assumes the estimator of the exact flow velocity field to be $L_t$-Lipschtz for any $t \in [0,1]$ and discuss in [BDD23, Theorem 2] how such assumption relate to the setting : under the additional assumption [BDD23, Assumption 4], the true flow velocity field is proven to be $L_t$-Lipschitz in space for any $t \in [0,1]$. Therefore, [BDD23] enhances the findings of [AVE22]. However, [BDD23, Assumption 4] is not an usual conditions considered in papers about convergence guarantees for SGM and it is unclear which type of distributions satisfy [BDD23, Assumption 4]. Moreover, both works [BDD23, AVE22] do not take into account the discretization error in their analysis.

In the context of Probability Flow ODEs, [LWCC24] and [GZ24] investigate the performance of such models in Total Vartiation distance and 2-Wasserstein distance. In contrast to [BDD23] and [AVE22], [LWCC24] and [GZ24] examine the error coming from the (prerequisite when implementing an algorithm) introduction of a time-discretization scheme. However, once again, the provided bounds work under smoothness assumptions either on the score or on its estimator. More precisely, the result reported in [LWCC24] depends on a small $\mathrm{L}^2$-Jacobian-estimation error assumption, besides a classical small $\mathrm{L}^2$-score-estimation error assumption. As for [GZ24], they assume the score to be Lipschitz in time and the data to be smooth and log-concave. In contrast, our result do not make such assumptions. Finally, to the best of our knowledge, the recent work [CDS23] represents the state of art in the context of SGMs without early-stopping procedure: [CDS23, Theorem 2.1] provides a sharp bound in KL divergence between the data distribution and the law of the SGM both in the overdamped and kinetic setting under the sole assumptions of an $\mathrm{L}^2$-score-approximation error and that the data distribution has finite Fisher information with respect to the standard Gaussian distribution. All previous results are obtained assuming either some Lipschitz condition on the score and/or its estimator ([CCL+23b], [CLL23]) or a manifold hypothesis on the data distribution ([Bor22b]). However, we underline that the FM framework enables to consider a significantly wider range of interpolating paths compared to SGMs and to avoid the trade-off concerning the time horizon $T$ which is inevitable when dealing with SGMs.

**Early-stopping procedure.** The recent work [GHJZ24] establishes convergence guarantees in 2-Wasserstein distance for FMs based on a deterministic sampling procedure. However, the results of [GHJZ24] requires to interpolate with a Gaussian distribution and applies only to data distributions which either have a bounded support, are strongly log-concave, or are the convolution between a Gaussian and an other probability distribution supported on an Euclidean Ball. In contrast, for our bound to hold true we only need the data distribution $\nu^\star$ and its score to have finite eight-order

moment. Furthermore, even if [GHJZ24] goes into depth when dealing with the statistical analysis of the estimator and the L$^2$-estimation error, the entire investigation therein pursued depends on the choice of ReLUnetworks with Lipschitz regularity control to approximate the velocity field. They motivate such choice by proving (see [GHJZ24, Theorem 5.1]) Lipschitz properties in time and space of the true velocity field under the aforementioned assumptions on the data. On the contrary, we do not assume any regularity on the estimator of the mimicking drift. In the context of Probability Flow ODEs, [CCL$^+$23a] provides bound in Total Variation distance, but assuming both the score and its estimator to be Lipschitz in space. So, also in the early-stopping regime, our bound improves previously obtained one.

To conclude, in the context of SGMs, [CLL23, CDS23, BDBDD23] are able to cover any data distribution with bounded second moment at the cost of using exponentially decreasing step-sizes. However, [CDS23, Corollary 2.4] and [BDBDD23] improves upon [CLL23, Theorem 2.2]: the term that takes track of the time-discretization error is linearly dependent on the dimension $d$ in the former works, whereas quadratically dependent on $d$ in the latter.

### 3.3   The proposed methodology.

In what follows, we provide a sketch of the proofs of Theorem 2 and Theorem 3 in order to outline and delineate our methodology.

The starting point of our proof of Theorem 2 is the following (by now) standard [CCL$^+$23b, CLL23, CDS23] decomposition of the KL divergence which is derived from Girsanov theorem:

$$\mathrm{KL}(\nu^\star|\nu_1^{\theta^\star}) \leq \mathrm{KL}(\mathbb{M}(\pi,\mathbb{B})|\mathrm{Law}(X_{[0,1]}^{\theta^\star})) \lesssim \sum_{k=0}^{N-1} \int_{t_k}^{t_{k+1}} \mathbb{E}\Big[\Big\|s_{\theta^\star}(t_k,X_{t_k}^{\mathrm{M}}) - \tilde{\beta}_t(X_t^{\mathrm{M}})\Big\|^2\Big]\mathrm{d}t$$

$$\lesssim \varepsilon^2 + \sum_{k=0}^{N-1} \int_{t_k}^{t_{k+1}} \mathbb{E}\Big[\Big\|\tilde{\beta}_{t_k}(X_{t_k}^{\mathrm{M}}) - \tilde{\beta}_t(X_t^{\mathrm{M}})\Big\|^2\Big]\mathrm{d}t \ ,$$

where, for the first inequality, we used the data processing inequality [Nut21, Lemma 1.6] and the last inequality follows from the triangle inequality and the assumption **H**3. In order to conclude, we should bound the L$^2$ norm of the adjoint process in the Pontryagin system associated with the Markovian projection of the interpolant. We do so by introducing a novel quantity in the generative model literature (see [Kre97]), namely the so-called *reciprocal characteristic* of the mimicking drift, *i.e.*,

$$(\partial_t + \mathcal{L}_t^{\mathrm{M}})\tilde{\beta}_t \ ,$$

where $\mathcal{L}^{\mathrm{M}}$ denotes the generator of $(X_t^{\mathrm{M}})_{t\in[0,1]}$. This quantity may be viewed as some sort of mean acceleration field and guides the time evolution of the mimicking drift, as

$$\mathrm{d}\tilde{\beta}_t(X_t^{\mathrm{M}}) = (\partial_t + \mathcal{L}_t^{\mathrm{M}})\tilde{\beta}_t(X_t^{\mathrm{M}})\mathrm{d}t + \sqrt{2}D_x\tilde{\beta}_t(X_t^{\mathrm{M}})\mathrm{d}B_t \ , \quad t \in [0,1] \ .$$

The main efforts in our proof are directed towards bounding the L$^2$ norm of the reciprocal characteristic whose representation in terms of either conditional moments or higher-order logarithmic derivatives of conditional densities is quite intricate, see (39). Trying to bound each of these terms separately requires strong assumptions on the initial distributions and couplings leading to suboptimal results. However, using integration by parts both in time and space and a double change of measure argument, and profiting from symmetry properties of the heat kernel, we managed to bound these terms under assumptions comparable to the minimal ones required in the analysis of SGMs. Note that the analysis of the reciprocal characteristic is not required for SGMs (it is always 0) and that controlling it also requires new tricks and ideas, since its representation contains up to three logarithmic derivatives of conditional distributions, whereas the analysis of SGMs requires at most two such derivatives to be analyzed.

Regarding the proof of Theorem 3, we consider the interpolated process $(X_t^{\mathrm{I}})_{t\in[0,\delta]}$ restricted to $[0,1-\delta]$. Denoting by $\pi_{1-\delta}$, the coupling between $\mu$ and $\nu_{1-\delta}^\star$ corresponding to the distribution of the couple $(X_0^{\mathrm{I}}, X_{1-\delta}^{\mathrm{I}})$. By the property of the Brownian bridge, $(X_t^{\mathrm{I}})_{t\in[0,1-\delta]}$ is a stochastic interpolant resulting from $\pi_{1-\delta}$ and the Brownian bridge on $[0,1-\delta]$. Therefore based on Remark 6,

we only have to show $\nu^\star_{1-\delta}$ and $\pi_{1-\delta}$ satisfy **H**1 and **H**2, replacing $\tilde{\pi}$ in **H**2 by $\tilde{\pi}_{1-\delta}$ defined in (15). More precisely, we show that they hold and that

$$\left\|\nabla \log \tilde{\pi}_{1-\delta}\right\|^8_{\mathrm{L}^8(\pi_{1-\delta})} \leqslant \|\nabla \log \mu\|^8_{\mathrm{L}^8(\mu)} + \mathbf{m}_8[\mu]\frac{1}{(1-\delta)^8} + \mathbf{m}_8[\nu^\star]\frac{1}{\delta^8} + d^4\frac{1}{\delta^4(1-\delta)^4}$$

$$\left\|\nabla \log \nu^\star_{1-\delta}\right\|^8_{\mathrm{L}^8(\nu^\star_{1-\delta})} \leqslant \mathbf{m}_8[\mu]\frac{1}{(1-\delta)^8} + \mathbf{m}_8[\nu^\star]\frac{1}{\delta^8} + d^4\frac{1}{\delta^4(1-\delta)^4}\,.$$

## 4   Conclusion

In this work, we have investigated a Diffusion Flow Matching model built around the Markovian projection of the $d$-dimensional Brownian bridge between the data distribution $\nu^\star$ and the base distribution $\mu$. In particular, we have derived convergence guarantees in Kullback-Leibler divergence, which take account of all the sources of error - time-discretization error and drift-estimation error - that arise when implementing the model, and which hold under mild moments conditions on $\mu, \nu^\star$, the scores of $\mu, \nu^\star$ and the score of the coupling $\pi$ between $\mu$ and $\nu^\star$. However, there are several questions remaining open. First, it would be worthy to understand if we could lower the order of integrability of the score associated with $\mu, \nu$ and $\pi$. Second, it would be interesting to complement our analysis by a statistical analysis of DFM (11), similarly to what have been achieved in [GHJZ24] for a particular deterministic FM model. Finally, it would be valuable to obtain better dimension dependence with respect to the space dimension $d$ when applying early-stopping procedure.

## Acknowledgments and Disclosure of Funding

The work of Marta Gentiloni-Silveri has been supported by the Paris Ile-de-France Région in the framework of DIM AI4IDF. The work by Alain Durmus is partially funded by the European Union (ERC-2022-SYG-OCEAN-101071601). Views and opinions expressed are however those of the author(s) only and do not necessarily reflect those of the European Union or the European Research Council Executive Agency. Neither the European Union nor the granting authority can be held responsible for them.

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

# A Postponed proofs

## A.1 The Markovian Projection

First, we introduce the set of assumptions under which Theorem 1 holds true.

**H4.** *For $t > s$, $(s, t, x, y) \mapsto p^Y_{t|s}(y|x)$ is continuously differentiable in the $t$ and $s$ variables and twice continuously differentiable in the $x$ and $y$ variables. Furthermore, $\partial_t p^Y_{t|s}(y|x)$, $\partial_s p^Y_{t|s}(y|x)$, $\nabla_x p^Y_{t|s}(y|x)$, $\nabla_y p^Y_{t|s}(y|x)$, $\nabla^2_x p^Y_{t|s}(y|x)$, $\nabla^2_y p^Y_{t|s}(y|x)$ are bounded.*

**H**4 ensures that $(s, t, x, y) \mapsto p^Y_{t|s}(y|x)$ is enough regular to allow a series of algebraic manipulations.

**H5.** *$\tilde{\beta}^Y$ is a locally bounded Borel vector field on $\mathbb{R}^d \times (0, 1)$ such that, for at least one probability solution $\mu_t$ to the Fokker-Planck equation*

$$\partial_t \mu_t + \mathrm{div}(\tilde{\beta}^Y_t \mu_t) - \Delta \mu_t = 0 \,, \quad t \in (0, 1) \,, \quad \mu_0 = \mu \,, \tag{16}$$

*it holds $\int \|\tilde{\beta}^Y_t(x)\| \mu_t(\mathrm{d}x)\mathrm{d}t < +\infty$.*

**H**5 provides uniqueness of the solution to the Fokker Planck equation with drift field $\tilde{\beta}^Y$, see [BKRS15, Theorem 9.4.3].

We are now ready to rigorously state and prove Theorem 1:

**Theorem 4.** *Consider a $\pi \in \Pi(\mu, \nu^\star)$, $\mathbb{Q}^\beta$ associated with (3) and the drift field defined in (8). Under **H**4 and 5, the Markov process $(X^M_t)_{t \in [0,1]}$ solution of (9) is such that, for any $t \in [0, 1)$, $X^I_t \overset{dist}{=} X^M_t$.*

*Proof of Theorem 4:* We start by reminding that $(s, t, x, y) \mapsto p^Y_{t|s}(y|x)$ satisfies for any $x, y \in \mathbb{R}^d$ and $s, t \in [0, 1]$ with $s < t$ both the Fokker-Planck equation

$$\partial_t p^Y_{t|s}(y|x) + \mathrm{div}_y(p^Y_{t|s}(y|x)\beta(x)) - \Delta_y p^Y_{t|s}(y|x) = 0 \,,$$

and the Kolmogorov backward equation

$$\partial_s p^Y_{t|s}(y|x) + \langle \beta(x), \nabla_x p^Y_{t|s}(y|x) \rangle + \Delta_x p^Y_{t|s}(y|x) = 0 \,.$$

If we exploit these well-known results, (4) and **H**4, we get that for $t \in (0, 1)$

$$
\begin{aligned}
\partial_t p^I_t(x) &= \partial_t \Big( \int_{\mathbb{R}^{2d}} p^Y_{t|0}(x|x_0)p^Y_{1|t}(x_1|x)\tilde{\pi}(\mathrm{d}x_0, \mathrm{d}x_1) \Big) \\
&= \int_{\mathbb{R}^{2d}} \Big( \Delta_x p^Y_{t|0}(x|x_0)p^Y_{1|t}(x_1|x) - p^Y_{t|0}(x|x_0)\Delta_x p^Y_{1|t}(x_1|x) \Big) \tilde{\pi}(\mathrm{d}x_0, \mathrm{d}x_1) \\
&\quad - \int_{\mathbb{R}^{2d}} \mathrm{div}_x(p^Y_{t|0}(x|x_0)\beta(x))p^Y_{1|t}(x_1|x)\tilde{\pi}(\mathrm{d}x_0, \mathrm{d}x_1) \\
&\quad - \int_{\mathbb{R}^{2d}} \langle \beta(x), \nabla_x p^Y_{1|t}(x_1|x) \rangle p^Y_{t|0}(x|x_0)\tilde{\pi}(\mathrm{d}x_0, \mathrm{d}x_1) \\
&= \int_{\mathbb{R}^{2d}} \Big( \Delta_x p^Y_{t|0}(x|x_0)p^Y_{1|t}(x_1|x) + \nabla_x p^Y_{t|0}(x|x_0)\nabla_x p^Y_{1|t}(x_1|x) \Big) \tilde{\pi}(\mathrm{d}x_0, \mathrm{d}x_1) \\
&\quad - \int_{\mathbb{R}^{2d}} \Big( \nabla_x p^Y_{t|0}(x|x_0)\nabla_x p^Y_{1|t}(x_1|x) + p^Y_{t|0}(x|x_0)\Delta_x p^Y_{1|t}(x_1|x) \Big) \tilde{\pi}(\mathrm{d}x_0, \mathrm{d}x_1) \\
&\quad - \langle \beta(x), \nabla_x p^I_t(x) \rangle - \mathrm{div}_x \beta(x)p^I_t(x) \\
&= \mathrm{div}_x \Big( \int_{\mathbb{R}^{2d}} \{\nabla_x p^Y_{t|0}(x|x_0)p^Y_{1|t}(x_1|x) - p^Y_{t|0}(x|x_0)\nabla_x p^Y_{1|t}(x_1|x)\}\tilde{\pi}(\mathrm{d}x_0, \mathrm{d}x_1) \Big) \\
&\quad - \mathrm{div}_x(\beta(x)p^I_t(x)) \\
&= \mathrm{div}_x \Big( -\beta(x)p^I_t(x) \\
&\quad + \int_{\mathbb{R}^{2d}} \{\nabla_x \log p^Y_{t|0}(x|x_0) - \nabla_x \log p^Y_{1|t}(x_1|x)\}p^Y_{t|0}(x|x_0)p^Y_{1|t}(x_1|x)\tilde{\pi}(\mathrm{d}x_0, \mathrm{d}x_1) \Big) \,.
\end{aligned}
$$

Therefore, if we set

$$v_t^Y(x) = \frac{\int_{\mathbb{R}^{2d}} \{\nabla_x \log p_{1|t}^Y(x_1|x) - \nabla_x \log p_{t|0}^Y(x|x_0)\} p_{t|0}^Y(x|x_0) p_{1|t}^Y(x_1|x) \tilde{\pi}(\mathrm{d}x_0, \mathrm{d}x_1)}{p_t^{\mathrm{I}}(x)} + \beta(x) \,,$$

we have proven that $p_t^{\mathrm{I}}(x)$ satisfies the following continuity equation

$$\partial_t p_t^{\mathrm{I}}(x) + \mathrm{div}_x(v_t^Y(x) p_t^{\mathrm{I}}(x)) = 0 \,, \quad t \in (0,1) \,, \ x \in \mathbb{R}^d \,.$$

Consequently, if we define

$$b_t^Y(x) = v_t^Y(x) + \nabla_x \log p_t^{\mathrm{I}}(x) \,,$$

we have that, under **H**4, $p_t^{\mathrm{I}}(x)$ satisfies the following Fokker-Planck equation

$$\partial_t p_t^{\mathrm{I}}(x) + \mathrm{div}_x(b_t^Y(x) p_t^{\mathrm{I}}(x)) - \Delta_x p_t^{\mathrm{I}}(x) = 0 \,, \quad t \in (0,1) \,, \ x \in \mathbb{R}^d \,. \tag{17}$$

So, $b_t(x)$ is a mimicking drift. It remains to show that $b^Y \equiv \tilde{\beta}^Y$: the thesis will then follows from the uniqueness of the solution to the Fokker Planck equation (17) under **H**5, see [BKRS15, Theorem 9.4.3]. To this aim, note that

$$\begin{aligned}
\nabla_x \log p_t^{\mathrm{I}}(x) &= \frac{\int_{\mathbb{R}^{2d}} \{\nabla_x p_{t|0}^Y(x|x_0) p_{1|t}^Y(x_1|x) + p_{t|0}^Y(x|x_0) \nabla_x p_{1|t}^Y(x_1|x)\} \tilde{\pi}(\mathrm{d}x_0, \mathrm{d}x_1)}{p_t^{\mathrm{I}}(x)} \\
&= \frac{\int_{\mathbb{R}^{2d}} \{\nabla_x \log p_{t|0}^Y(x|x_0) + \nabla_x \log p_{1|t}^Y(x_1|x)\} p_{t|0}^Y(x|x_0) p_{1|t}^Y(x_1|x) \tilde{\pi}(\mathrm{d}x_0, \mathrm{d}x_1)}{p_t^{\mathrm{I}}(x)} \,.
\end{aligned}$$

Therefore, for any $t \in [0,1)$ and $x \in \mathbb{R}^d$, we have that

$$b_t^Y(x) = 2 \frac{\int_{\mathbb{R}^{2d}} \nabla_x \log p_{1|t}^Y(x_1|x) p_{t|0}^Y(x|x_0) p_{1|t}^Y(x_1|x) \tilde{\pi}(\mathrm{d}x_0, \mathrm{d}x_1)}{p_t^{\mathrm{I}}(x)} + \beta(x) = \tilde{\beta}_t^Y(x) \,.$$

$\square$

## A.2 Preliminary results

We begin this section with two remarks aiming at highlighting some of the properties of the heat kernels (12) and some of their important consequences.

*Remark* 8. It is well-known that $(s,x,y) \mapsto p_s(y|x)$ defined in (12) is twice continuously differentiable in the space variables $x$ and $y$ and satisfies for $x,y \in \mathbb{R}^d, s \in (0,1]$,

$$\nabla_x p_s(y|x) = -\frac{x-y}{2s} p_s(y|x) = -\nabla_y p_s(y|x) \,, \tag{18}$$

$$\nabla_x^2 p_s(y|x) = -\frac{1}{2s} p_s(y|x) \,\mathrm{Id} + \frac{(x-y)(x-y)^{\mathrm{T}}}{4s^2} p_s(y|x) = \nabla_y^2 p_s(y|x) \,, \tag{19}$$

$$\Delta_x p_s(y|x) = -\frac{d}{2s} p_s(y|x) + \left\| \frac{x-y}{2s} \right\|^2 p_s(y|x) = \Delta_y p_s(y|x) \,. \tag{20}$$

Moreover (12) satisfies the heat equation, *i.e.*,

$$\partial_s p_s(y|x) = \Delta_x p_s(y|x) \,, \quad s \in (0,1] \,, \ x,y \in \mathbb{R}^d \,. \tag{21}$$

Thus, in particular, $(s,x,y) \mapsto p_s(y|x)$ is continuously differentiable in the time variable $s$.

*Remark* 9. In the case $\beta \equiv 0$, *i.e.*, $\mathbb{Q}^\beta = \mathbb{B}$, under **H**1, the conditions of Theorem 1 hold. Indeed, as highlighted in Remark 8, $(s,x,y) \mapsto p_s(y|x)$ defined in (12) is continuously differentiable in the time variable $s$ and twice continuously differentiable in the space variables $x$ and $y$. Also, using Equation (18), Equation (19) and (12), it's straightforward to verify that $\nabla_x p_{t|s}^Y(y|x), \nabla_y p_{t|s}^Y(y|x)$, $\nabla_x^2 p_{t|s}^Y(y|x), \nabla_y^2 p_{t|s}^Y(y|x)$ are bounded. Additionally, using (20) and (21), it's easy to argue that also $\partial_t p_{t|s}^Y(y|x)$ and $\partial_s p_{t|s}^Y(y|x)$ are bounded. So **H**4 is verified and $p_t^{\mathrm{I}}$ solves the Fokker-Planck equation

(16). Moreover $\tilde{\beta}$ is clearly locally bounded on $\mathbb{R}^d \times (0,1)$ and, as a consequence of (18), (5), and Jensen inequality, it satisfies uniformly in time

$$
\begin{aligned}
\int_{\mathbb{R}^d} \left\| \tilde{\beta}_t(x) \right\|^2 p_t^{\mathrm{I}}(x) \mathrm{d}x &= \int_{\mathbb{R}^d} \left\| \frac{1}{p_t^{\mathrm{I}}(x)} \int_{\mathbb{R}^{2d}} \frac{x_1 - x}{1-t} p_t(x|x_0) p_{1-t}(x_1|x) \tilde{\pi}(\mathrm{d}x_0, \mathrm{d}x_1) \right\|^2 p_t^{\mathrm{I}}(x) \mathrm{d}x \\
&= \mathbb{E}\left[ \left\| \mathbb{E}\left[ \frac{X_1^{\mathrm{I}} - X_t^{\mathrm{I}}}{1-t} \middle| X_t^{\mathrm{I}} \right] \right\|^2 \right] \\
&= \mathbb{E}\left[ \left\| \mathbb{E}\left[ X_1^{\mathrm{I}} - X_0^{\mathrm{I}} - \frac{\sqrt{2t}}{\sqrt{1-t}} \mathrm{Z} \middle| X_t^{\mathrm{I}} \right] \right\|^2 \right] \\
&\lesssim \mathbb{E}\left[ \left\| X_1^{\mathrm{I}} - X_0^{\mathrm{I}} - \sqrt{2t}\mathrm{Z} \right\|^2 \right] \\
&\lesssim \mathbf{m}_2[\mu] + \mathbf{m}_2[\nu^\star] + d \ ,
\end{aligned}
$$

which is finite under **H**1. It follows that $\int \|\tilde{\beta}_t^Y(x)\| p_t^{\mathrm{I}}(x) \mathrm{d}x \mathrm{d}t < +\infty$. So, **H**5 is verified.

Additionally, hereunder, we state three lemmas that will be crucial in the derivation of the bounds provided in Theorems 2 and 3.

**Lemma 1.** *For any $p \geq 1$, they hold*

$$
\mathbb{E}[\|X_s^{\mathrm{I}} - X_0^{\mathrm{I}}\|^{2p}] \lesssim s^{2p}\mathbf{m}_{2p}[\mu] + s^{2p}\mathbf{m}_{2p}[\nu^\star] + d^p s^p (1-s)^p \ ,
$$

*and*

$$
\mathbb{E}[\|X_1^{\mathrm{I}} - X_s^{\mathrm{I}}\|^{2p}]\mathrm{d}s \lesssim (1-s)^{2p}\mathbf{m}_{2p}[\mu] + (1-s)^{2p}\mathbf{m}_{2p}[\nu^\star] + d^p s^p (1-s)^p \ .
$$

*Proof of Lemma 1:* As a direct consequence of (5) and Young inequality, it holds

$$
\begin{aligned}
\mathbb{E}[\|X_s^{\mathrm{I}} - X_0^{\mathrm{I}}\|^{2p}] &= \mathbb{E}\left[ \left\| s(X_1^{\mathrm{I}} - X_0^{\mathrm{I}}) + \sqrt{2s(1-s)}\mathrm{Z} \right\|^{2p} \right] \\
&\lesssim s^{2p}\mathbb{E}[\|X_0^{\mathrm{I}} - X_1^{\mathrm{I}}\|^{2p}] + (2s(1-s))^p \mathbb{E}[\|\mathrm{Z}\|^{2p}] \\
&\lesssim s^{2p}\mathbf{m}_{2p}[\mu] + s^{2p}\mathbf{m}_{2p}[\nu^\star] + d^p s^p (1-s)^p \ .
\end{aligned}
$$

A similar argument holds for $\mathbb{E}[\|X_1^{\mathrm{I}} - X_s^{\mathrm{I}}\|^{2p}]$. $\qquad\square$

We preface the next lemma with a definition.

**Definition 1.** Consider $\mathbb{Q} \in \mathcal{P}(\mathbb{W})$. The reverse time measure $\mathbb{Q}^{\mathrm{R}} \in \mathcal{P}(\mathbb{W})$ of $\mathbb{Q}$ is defined as follows: for any $\mathsf{A} \in \mathcal{B}(\mathbb{W})$

$$
\mathbb{Q}^{\mathrm{R}}(\mathsf{A}) = \mathbb{Q}(\mathsf{A}^{\mathrm{R}}), \ ,
$$

where $\mathsf{A}^{\mathrm{R}} := \{t \mapsto \omega(1-t) \ : \ \omega \in \mathsf{A}\}$.

**Lemma 2.** *Assume $\pi \ll \mathrm{Leb}^{2d}$ and $\mu, \nu^\star \ll \mathrm{Leb}^d$. Then $(X_t^{\mathrm{I}})_{t \in [0,1]}$ solves weakly*

$$
\mathrm{d}\overrightarrow{X}_t = 2\overrightarrow{b}_t(\overrightarrow{X}_0, \overrightarrow{X}_t)\mathrm{d}t + \sqrt{2}\mathrm{d}\overrightarrow{B}_t \ , \quad t \in [0,1] \ , \quad \overrightarrow{X}_0 \sim \mu \ . \tag{22}
$$

*with $(\overrightarrow{B}_t)_{t \in [0,1]}$ $d$-dimensional Brownian motion independent of $\overrightarrow{X}_0$ and*

$$
\overrightarrow{b}_t(x_0, x) = \nabla_x \psi_t^{x_0}(x) \ ,
$$

*where $\psi_t^{x_0}(x)$ solves*

$$
\partial_t \psi_t^{x_0} + \Delta_x \psi_t^{x_0} + \|\nabla_x \psi_t^{x_0}\|^2 = 0 \ , \quad t \in [0,1) \ , \quad \psi_1^{x_0} = \log \tilde{\pi}_0^{x_0} \ , \tag{23}
$$

*with*

$$
\tilde{\pi}_0^{x_0}(x) := \frac{\mathrm{d}\pi(x_0, x)}{\mathrm{d}\mathrm{Leb}^{2d}} \frac{1}{p_1(x|x_0)} \bigg/ \frac{\mathrm{d}\mu(x_0)}{\mathrm{d}\mathrm{Leb}^d} \ , \tag{24}
$$

*and* $p_1(x|x_0)$ *defined in* (12).

*Similarly,* $((X_t^I)_{t\in[0,1]})^R$ *solves weakly*

$$\mathrm{d}\overleftarrow{X}_t = 2\overleftarrow{b}_t(\overleftarrow{X}_0, \overleftarrow{X}_t)\mathrm{d}t + \sqrt{2}\mathrm{d}\overleftarrow{B}_t\,, \quad t \in [0,1]\,, \quad \overleftarrow{X}_0 \sim \nu^\star\,.$$

*with* $(\overleftarrow{B}_t)_{t\in[0,1]}$ *d-dimensional Brownian motion independent of* $\overleftarrow{X}_0$ *and*

$$\overleftarrow{b}_t(x_1, x) = \nabla_x \varphi_{1-t}^{x_1}(x)\,,$$

*where* $\varphi_t^{x_1}(x)$ *solves*

$$-\partial_t \varphi_t^{x_1} + \Delta_x \varphi_t^{x_1} + \|\nabla_x \varphi_t^{x_1}\|^2 = 0\,, \quad t \in [0,1)\,, \quad \varphi_0^{x_1} = \log \tilde{\pi}_1^{x_1}\,,$$

*with*

$$\tilde{\pi}_1^{x_1}(x) := \frac{\mathrm{d}\pi(x, x_1)}{\mathrm{dLeb}^{2d}} \frac{1}{p_1(x_1|x)} \bigg/ \frac{\mathrm{d}\nu^\star(x_1)}{\mathrm{dLeb}^d}\,, \tag{25}$$

*and* $p_1(x_1|x)$ *defined in* (12).

*Proof of Lemma 2:* We just show that $(X_t^I)_{t\in[0,1]}$ solves weakly (22). The argument for $((X_t^I)_{t\in[0,1]})^R$ is similar and therefore omitted.

For a fixed $x_0 \in \mathbb{R}^d$, we denote by $(x_0, \mathsf{A}) \mapsto \mathbb{I}_{\pi,\mathbb{B}}(x_0, \mathsf{A})$ the conditional distribution of $(X_t^I)_{t\in[0,1]}$ given $X_0^I$ (see e.g. [Kle13, Theorem 8.37] for the existence of this conditional distribution) and by $(B_t^{x_0})_{t\in[0,1]}$ the solution to

$$\mathrm{d}B_t^{x_0} = \sqrt{2}\mathrm{d}B_t\,, \quad t \in [0,1]\,, \quad B_0^{x_0} = x_0\,.$$

Also, we denote by $(W_t)_{t\in[0,1]}$ the canonical process on the Wiener space $\mathbb{W}$ and by $(\mathcal{F}_t)_{t\in[0,1]}$ the corresponding natural filtration. Note that, as a consequence of the very definition of $(X_t^I)_{t\in[0,1]}$, (23) and Ito's formula applied to $(\psi_t^{x_0}(B_t^{x_0}))_{t\in[0,1]}$, it holds

$$\frac{\mathrm{d}\mathbb{I}_{\pi,\mathbb{B}}(x_0, \cdot)}{\mathrm{dLaw}((B_t^{x_0})_{t\in[0,1]})}((B_t^{x_0})_{t\in[0,1]})$$

$$= \tilde{\pi}_0^{x_0}(B_1^{x_0})$$

$$= \exp\left(\psi_1^{x_0}(B_1^{x_0})\right)$$

$$= \exp\left(\psi_1^{x_0}(B_1^{x_0}) - \psi_0^{x_0}(x_0)\right)$$

$$= \exp\left(\psi_1^{x_0}(B_1^{x_0}) - \psi_0^{x_0}(B_0^{x_0}) - \int_0^1 \left\{\partial_t \psi_t^{x_0} + \Delta_x \psi_t^{x_0} + \|\nabla_x \psi_t^{x_0}\|^2\right\}(B_t^{x_0})\mathrm{d}t\right)$$

$$= \exp\left(\int_0^1 \nabla_x \psi_t^{x_0}(B_t^{x_0})\mathrm{d}B_t^{x_0} - \int_0^1 \|\nabla_x \psi_t^{x_0}(B_t^{x_0})\|^2\,\mathrm{d}t\right)\,.$$

Hence, for any $t \in [0,1]$ we have that

$$\left.\frac{\mathrm{d}\mathbb{I}_{\pi,\mathbb{B}}(x_0, \cdot)}{\mathrm{dLaw}((B_t^{x_0})_{t\in[0,1]})}\right|_{\mathcal{F}_t} = D_t\,,$$

where

$$D_t = \exp\left(\int_0^t \nabla_x \psi_t^{x_0}(W_s)\mathrm{d}W_s - \int_0^1 \|\nabla_x \psi_t^{x_0}(W_s)\|^2\,\mathrm{d}s\right)$$

is continuous $\mathrm{Law}((B_t^{x_0})_{t\in[0,1]})$- almost surely and is such that

$$\mathrm{d}D_t = D_t \nabla_x \psi_s^{x_0}(W_s)\mathrm{d}W_t\,.$$

Therefore, being $(W_t)_{t\in[0,1]}$ a martingale under $\mathrm{Law}((B_t^{x_0})_{t\in[0,1]})$, as a consequence of Girsanov theorem (see [RY13, Theorem 1.4]), we have that

$$\left(W_t - \left(D^{-1}\langle W, D\rangle\right)_t\right)_{t\in[0,1]} = \left(W_t - 2\int_0^t \nabla_x \psi_s^{x_0}(W_s)\mathrm{d}s\right)_{t\in[0,1]}$$

is a $((\mathcal{F}_t)_t, \mathbb{I}_{\pi,\mathbb{B}}(x_0,\cdot))$- martingale with bracket $\sqrt{2}t$. It follows that, under $\mathbb{I}_{\pi,\mathbb{B}}(x_0,\cdot)$, $(W_t)_{t\in[0,1]}$ solves

$$\mathrm{d}W_t = 2\nabla_x \psi_t^{x_0}(W_s)\mathrm{d}t + \sqrt{2}\mathrm{d}B_t\ , \quad t \in [0,1]\ , \quad W_0 = x_0\ .$$

The thesis is now a direct consequence of the fact that

$$\mathbb{I}(\pi,\mathbb{B})(\mathsf{A}) = \int_{\mathbb{R}^d} \mathbb{P}(X^{\mathrm{I}} \in \mathsf{A}|X_0^{\mathrm{I}} = x_0)\mu(\mathrm{d}x_0) = \int_{\mathbb{R}^d} \mathbb{I}_{\pi,\mathbb{B}}(x_0,\mathsf{A})\mu(\mathrm{d}x_0)\ ,$$

for any measurable $\mathsf{A} \in \mathbb{W}$. $\qquad\square$

**Lemma 3.** *Assume H2. Then, almost surely, it holds*

$$\overrightarrow{b}_t(X_0^{\mathrm{I}}, X_t^{\mathrm{I}}) = \mathbb{E}\left[\left.\frac{\nabla \tilde{\pi}_0^{X_0^{\mathrm{I}}}(X_1^{\mathrm{I}})}{\tilde{\pi}_0^{X_0^{\mathrm{I}}}(X_1^{\mathrm{I}})}\right| (X_0^{\mathrm{I}}, X_t^{\mathrm{I}})\right]\ , \tag{26}$$

*where $\tilde{\pi}_0^{x_0}$ is defined as in* (24)*. Moreover, for any $u \in [0, 1-s]$, $s \in [0,1]$ and $p \in \{2,4,8\}$ it holds*

$$\mathbb{E}\left[\left\|\int_u^{u+s} \overrightarrow{b}_r(\overrightarrow{X}_0, \overrightarrow{X}_r)\mathrm{d}r\right\|^p\right] \lesssim s^p\left(\|\nabla\log\tilde{\pi}\|_{\mathrm{L}^p(\pi)}^p + \|\nabla\log\mu\|_{\mathrm{L}^p(\mu)}^p\right)\ . \tag{27}$$

*Similarly, almost surely it holds*

$$\overleftarrow{b}_t(X_0^{\mathrm{I}}, X_t^{\mathrm{I}}) = \mathbb{E}\left[\left.\frac{\nabla \tilde{\pi}_1^{X_1^{\mathrm{I}}}(X_0^{\mathrm{I}})}{\tilde{\pi}_1^{X_1^{\mathrm{I}}}(X_0^{\mathrm{I}})}\right| (X_0^{\mathrm{I}}, X_t^{\mathrm{I}})\right]\ , \tag{28}$$

*where $\tilde{\pi}_1^{x_1}$ is defined as in* (25)*. Moreover, for any $u \in [0, 1-s]$, $s \in [0,1]$ and $p \in \{2,4,8\}$ it holds*

$$\mathbb{E}\left[\left\|\int_u^{u+s} \overleftarrow{b}_r(\overleftarrow{X}_0, \overleftarrow{X}_r)\mathrm{d}r\right\|^p\right] \lesssim s^p\left(\|\nabla\log\tilde{\pi}\|_{\mathrm{L}^p(\pi)}^p + \|\nabla\log\nu^\star\|_{\mathrm{L}^p(\nu^\star)}^p\right)\ . \tag{29}$$

*Proof of Lemma 3:* We just show (26) and (27). The proof of (28) and (29) is similar and therefore omitted. First of all, note that (26) is trivial for $t = 1$. We therefore focus on $t \in [0,1)$. It's well known that the solution $\psi_t^{x_0}(x)$ to the Hamilton-Jacobi-Bellman equation (23) is given by

$$\psi_t^{x_0}(x) = \log\left(\int \tilde{\pi}_0^{x_0}(x_1)p_{1-t}(x_1|x)\mathrm{d}x_1\right)\ , \quad t \in [0,1)\ .$$

Therefore, we have that for $t \in [0,1)$

$$\overrightarrow{b}_t(x_0, x) = \nabla_x \psi_t^{x_0}(x) = \frac{\int \tilde{\pi}_0^{x_0}(x_1)\nabla_x p_{1-t}(x_1|x)\mathrm{d}x_1}{\int \tilde{\pi}_0^{x_0}(\tilde{x}_1)p_{1-t}(\tilde{x}_1|x)\mathrm{d}\tilde{x}_1}\ .$$

Using (18) and integrating by parts, we get for $t \in [0,1)$ that

$$\overrightarrow{b}_t(x_0, x) = -\frac{\int \tilde{\pi}_0^{x_0}(x_1)\nabla_{x_1} p_{1-t}(x_1|x)\mathrm{d}x_1}{\int \tilde{\pi}_0^{x_0}(\tilde{x}_1)p_{1-t}(\tilde{x}_1|x)\mathrm{d}\tilde{x}_1} = \frac{\int (\nabla\tilde{\pi}_0^{x_0}(x_1)/\tilde{\pi}_0^{x_0}(x_1))\tilde{\pi}_0^{x_0}(x_1)p_{1-t}(x_1|x)\mathrm{d}x_1}{\int \tilde{\pi}_0^{x_0}(\tilde{x}_1)p_{1-t}(\tilde{x}_1|x)\mathrm{d}\tilde{x}_1}\ .$$

But then, it suffices to prove that for $t \in [0,1)$ it holds

$$\frac{\tilde{\pi}_0^{x_0}(x_1)p_{1-t}(x_1|x)}{\int \tilde{\pi}_0^{x_0}(\tilde{x}_1)p_{1-t}(\tilde{x}_1|x)\mathrm{d}\tilde{x}_1} = p_{1|0,t}^{\mathrm{I}}(x_1|x_0,x)\ ,$$

to conclude. To do so, we simply use (4).

$$\begin{aligned}
p_{1|0,t}^{\mathrm{I}}(x_1|x_0,x) &= \frac{p_{0,1,t}^{\mathrm{I}}(x_0,x_1,x)}{\int p_{0,1,t}^{\mathrm{I}}(x_0,\tilde{x}_1,x)\mathrm{d}\tilde{x}_1}\\
&= \frac{\pi(x_0,x_1)p_t(x|x_0)p_{1-t}(x_1|x)}{p_1(x_1|x_0)}\bigg/ \int \frac{\pi(x_0,\tilde{x}_1)p_t(x|x_0)p_{1-t}(\tilde{x}_1|x)}{p_1(\tilde{x}_1|x_0)}\mathrm{d}\tilde{x}_1\\
&= \frac{\pi(x_0,x_1)p_{1-t}(x_1|x)}{p_1(x_1|x_0)}\bigg/ \int \frac{\pi(x_0,\tilde{x}_1)p_{1-t}(\tilde{x}_1|x)}{p_1(\tilde{x}_1|x_0)}\mathrm{d}\tilde{x}_1\\
&= \frac{\pi(x_0,x_1)p_{1-t}(x_1|x)}{\mu(x_0)p_1(x_1|x_0)}\bigg/ \int \frac{\pi(x_0,\tilde{x}_1)p_{1-t}(\tilde{x}_1|x)}{\mu(x_0)p_1(\tilde{x}_1|x_0)}\mathrm{d}\tilde{x}_1\\
&= \frac{\tilde{\pi}_0^{x_0}(x_1)p_{1-t}(x_1|x)}{\int \tilde{\pi}_0^{x_0}(\tilde{x}_1)p_{1-t}(\tilde{x}_1|x)\mathrm{d}\tilde{x}_1}\ .
\end{aligned}$$

We are left with the bounds (27).

We prove (27) only for $p = 1$. The other two cases are analogous. To this aim, we simply make use of (22), Hölder and Jensen inequalities and the properties of the conditional expectation.

$$
\mathbb{E}\left[\left\|\int_u^{u+s} \overrightarrow{b}_r(\overrightarrow{X}_0, \overrightarrow{X}_r)\mathrm{d}r\right\|^2\right] = \mathbb{E}\left[\left\|\int_u^{u+s} \mathbb{E}\left[\frac{\nabla \tilde{\pi}_0^{X_0^{\mathrm{I}}}(X_1^{\mathrm{I}})}{\tilde{\pi}_0^{X_0^{\mathrm{I}}}(X_1^{\mathrm{I}})}\bigg|(X_0^{\mathrm{I}}, X_r^{\mathrm{I}})\right]\mathrm{d}r\right\|^2\right]
$$

$$
\lesssim s\mathbb{E}\left[\int_u^{u+s}\left\|\mathbb{E}\left[\frac{\nabla \tilde{\pi}_0^{X_0^{\mathrm{I}}}(X_1^{\mathrm{I}})}{\tilde{\pi}_0^{X_0^{\mathrm{I}}}(X_1^{\mathrm{I}})}\bigg|(X_0^{\mathrm{I}}, X_r^{\mathrm{I}})\right]\right\|^2\mathrm{d}r\right]
$$

$$
\lesssim s\mathbb{E}\left[\int_u^{u+s}\mathbb{E}\left[\left\|\frac{\nabla \tilde{\pi}_0^{X_0^{\mathrm{I}}}(X_1^{\mathrm{I}})}{\tilde{\pi}_0^{X_0^{\mathrm{I}}}(X_1^{\mathrm{I}})}\right\|^2\bigg|(X_0^{\mathrm{I}}, X_r^{\mathrm{I}})\right]\mathrm{d}r\right]
$$

$$
= s\mathbb{E}\left[\int_u^{u+s}\left\|\frac{\nabla \tilde{\pi}_0^{X_0^{\mathrm{I}}}(X_1^{\mathrm{I}})}{\tilde{\pi}_0^{X_0^{\mathrm{I}}}(X_1^{\mathrm{I}})}\right\|^2\mathrm{d}r\right]
$$

$$
\lesssim s^2\int \|\nabla \log \tilde{\pi}_0^{x_0}(x_1)\|^2\,\mathrm{d}\pi(x_0, x_1)
$$

$$
\lesssim s^2\left(\|\nabla \log \tilde{\pi}\|_{\mathrm{L}^2(\pi)}^2 + \|\nabla \log \mu\|_{\mathrm{L}^2(\mu)}^2\right),
$$

where, in the last inequality, we have used the very definition of $\tilde{\pi}_0^{x_0}$ given in (24). $\qquad\square$

## A.3 Main results

We preface the proofs of our main results with some extra notation. For sake of brevity, we denote by

$$
\overrightarrow{f}_u^t = 2\int_u^{t+u}\overrightarrow{b}_r(\overrightarrow{X}_0, \overrightarrow{X}_r)\mathrm{d}r, \quad \overrightarrow{g}_u^t = \overrightarrow{B}_{t+u} - \overrightarrow{B}_u,
$$

and similarly

$$
\overleftarrow{f}_u^t = 2\int_u^{t+u}\overleftarrow{b}_r(\overleftarrow{X}_0, \overleftarrow{X}_r)\mathrm{d}r, \quad \overleftarrow{g}_u^t = \overleftarrow{B}_{t+u} - \overleftarrow{B}_u.
$$

*Remark* 10. With this new notation, according to Lemma 2, we have that

$$
(X_t^{\mathrm{I}})_{t\in[0,1]} \stackrel{\mathrm{dist}}{=} (\overrightarrow{X}_t)_{t\in[0,1]}, \quad (X_t^{\mathrm{I}})_{t\in[0,1]} \stackrel{\mathrm{dist}}{=} ((\overleftarrow{X}_t)_{t\in[0,1]})^{\mathrm{R}},
$$

and that for any $u \in [0, 1]$ and $t \in [0, 1 - u]$,

$$
X_{t+u}^{\mathrm{I}} - X_u^{\mathrm{I}} \stackrel{\mathrm{dist}}{=} \overrightarrow{f}_u^t + \overrightarrow{g}_u^t, \quad X_{1-(t+u)}^{\mathrm{I}} - X_{1-u}^{\mathrm{I}} \stackrel{\mathrm{dist}}{=} \overleftarrow{f}_u^t + \overleftarrow{g}_u^t.
$$

Furthermore, according to Lemma 3, under **H**2, for any $u \in [0, 1]$, $t \in [0, 1 - u]$, $p \in \{2, 4, 8\}$, we have that

$$
\mathbb{E}\left[\left\|\overrightarrow{f}_u^t\right\|^p\right] \lesssim t^p(\|\nabla \log \mu\|_{\mathrm{L}^p(\mu)}^p + \|\nabla \log \tilde{\pi}\|_{\mathrm{L}^p(\pi)}^p),
$$

and

$$
\mathbb{E}\left[\left\|\overleftarrow{f}_u^t\right\|^p\right] \lesssim t^p(\|\nabla \log \nu^\star\|_{\mathrm{L}^p(\nu^\star)}^p + \|\nabla \log \tilde{\pi}\|_{\mathrm{L}^p(\pi)}^p).
$$

*Remark* 11. Note that for any $u \in [0, 1]$ and $t \in [0, 1 - u]$

$$
\overrightarrow{g}_u^t \perp\!\!\!\perp (\overrightarrow{X}_r)_{r\leq u}, \quad \overleftarrow{g}_u^t \perp\!\!\!\perp (\overleftarrow{X}_r)_{r\leq u}.
$$

This fact is an almost immediate consequence of the Markov property of $(\overrightarrow{B}_t)_{t\in[0,1]}$, see [BB17, Theorem 3.3]: consider the filtration $(\mathcal{F}_t)_{t\in[0,1]}$ defined by $\mathcal{F}_t = \sigma(\overrightarrow{X}_0, (\overrightarrow{B}_u)_{u\leq t})$. Then, being $(\overrightarrow{B}_t)_{t\in[0,1]} \perp\!\!\!\perp \overrightarrow{X}_0$, $(\overrightarrow{B}_t)_{t\in[0,1]}$ is $(\mathcal{F}_t)_{t\in[0,1]}$-adapted and, as a consequence of the Markov property of $(\overrightarrow{B}_t)_{t\in[0,1]}$, $(\overrightarrow{g}_u^t)_{t\in[0,1-u]} = (\overrightarrow{B}_{t+u} - \overrightarrow{B}_u)_{t\in[0,1-u]} \perp\!\!\!\perp \mathcal{F}_u$. On the other hand, it's well known that $(\overrightarrow{X}_t)_{t\in[0,1]}$ is $(\mathcal{F}_t)_{t\in[0,1]}$-adapted. It therefore follows that $\overrightarrow{g}_u^t \perp\!\!\!\perp (\overrightarrow{X}_r)_{r\leq u}$. A similar arguments holds for $(\overleftarrow{X}_t)_{t\in[0,1]}$.

## A.4 Proof of Theorem 2

We fix $0 < \epsilon < \min\{h, 1/2\}$ and, for any $t \in [0, 1 - \epsilon]$, we denote by $\nu_t^\star = \text{Law}(X_t^{\text{M}})$ and $\nu_t^{\theta^\star} = \text{Law}(X_t^{\theta^\star})$.

First, using the data processing inequality [Nut21, Lemma 1.6], the standard decomposition of the KL divergence [CCL⁺23b, CLL23, CDS23] based on Girsanov theorem, triangle inequality and **H3**, we bound the KL divergence between $\nu_{1-\epsilon}^\star$ and $\nu_{1-\epsilon}^{\theta^\star}$ as follows

$$
\begin{aligned}
&\text{KL}(\nu_{1-\epsilon}^\star | \nu_{1-\epsilon}^{\theta^\star}) \\
&\leq \text{KL}(\text{Law}((X_t^{\text{M}})_{t\in[0,1-\epsilon]}) | \text{Law}((X_t^{\theta^\star})_{t\in[0,1-\epsilon]})) \\
&\lesssim \sum_{k=0}^{N-2} \int_{t_k}^{t_{k+1}} \mathbb{E}\Big[\Big\| s_{\theta^\star}(t_k, X_{t_k}^{\text{M}}) - \tilde{\beta}_t(X_t^{\text{M}}) \Big\|^2\Big] \mathrm{d}t + \int_{1-h}^{1-\epsilon} \mathbb{E}\Big[\Big\| s_{\theta^\star}(1-h, X_{1-h}^{\text{M}}) - \tilde{\beta}_t(X_t^{\text{M}}) \Big\|^2\Big] \mathrm{d}t \\
&\lesssim \varepsilon^2 + \sum_{k=0}^{N-2} \int_{t_k}^{t_{k+1}} \mathbb{E}\Big[\Big\| \tilde{\beta}_{t_k}(X_{t_k}^{\text{M}}) - \tilde{\beta}_t(X_t^{\text{M}}) \Big\|^2\Big] \mathrm{d}t + \int_{1-h}^{1-\epsilon} \mathbb{E}\Big[\Big\| \tilde{\beta}_{1-h}(X_{1-h}^{\text{M}}) - \tilde{\beta}_t(X_t^{\text{M}}) \Big\|^2\Big] \mathrm{d}t \;.
\end{aligned}
\tag{30}
$$

Second, we aim at bounding the RHS of (30) uniformly in $\epsilon$. Indeed, if we assume to be able to bound it with a constant $A$ independent of $\epsilon$, then, using the weak convergence of $X_{1-\epsilon}^{\text{M}}$ to $X_1^{\text{I}}$ (whose law is given by $\nu^\star$) as $\epsilon \to 0$, the continuity of $(X_t^{\theta^\star})_{t\in[0,1]}$, hence the weak convergence of $X_{1-\epsilon}^{\theta^\star}$ to $X_1^{\theta^\star}$ (whose law is given by $\nu_1^{\theta^\star}$) as $\epsilon \to 0$, and the lower semi-continuity of the KL-divergence with respect to the weak convergence [VEH14, Theorem 19], we will get

$$
\text{KL}(\nu^\star | \nu_1^{\theta^\star}) \leq \liminf_{\epsilon\to 0} \text{KL}(\nu_{1-\epsilon}^\star | \nu_{1-\epsilon}^{\theta^\star}) \lesssim \liminf_{\epsilon\to 0} A = A \;.
\tag{31}
$$

Let us therefore bound the RHS of (30). We will do so by using stochastic calculus tools, and, more precisely, Ito's formula. To this aim let us introduce the generator of $(X_t^{\text{M}})_{t\in[0,1-\epsilon]}$, which is defined for any $t \in [0, 1 - \epsilon]$ and $\rho \in C^2(\mathbb{R}^d)$ as

$$
\mathcal{L}_t^{\text{M}}\rho := \langle \nabla_x \rho, \tilde{\beta}_t \rangle + \Delta_x \rho \;.
$$

Using Ito's formula, we get that

$$
\mathrm{d}\tilde{\beta}_t(X_t^{\text{M}}) = (\partial_t + \mathcal{L}_t^{\text{M}})\tilde{\beta}_t(X_t^{\text{M}})\mathrm{d}t + \sqrt{2}D_x\tilde{\beta}_t(X_t^{\text{M}})\mathrm{d}B_t \;, \quad t \in [0, 1-\epsilon] \;.
$$

So, applying Young inequality and Ito's isometry, we have that, for any $k = 0, \cdots, N-1$

$$
\begin{aligned}
&\mathbb{E}\Big[\Big\| \tilde{\beta}_{t_k}(X_{t_k}^{\text{M}}) - \tilde{\beta}_t(X_t^{\text{M}}) \Big\|^2\Big] \\
&= \mathbb{E}\Big[\Big\| \int_{t_k}^{t} (\partial_s + \mathcal{L}_s^{\text{M}})\tilde{\beta}_s(X_s^{\text{M}})\mathrm{d}s + \sqrt{2}\int_{t_k}^{t} D_x\tilde{\beta}_s(X_s^{\text{M}})\mathrm{d}B_s \Big\|^2\Big] \\
&\lesssim \mathbb{E}\Big[\Big\| \int_{t_k}^{t} (\partial_s + \mathcal{L}_s^{\text{M}})\tilde{\beta}_s(X_s^{\text{M}})\mathrm{d}s \Big\|^2\Big] + 2\int_{t_k}^{t} \mathbb{E}\Big[\Big\| D_x\tilde{\beta}_s(X_s^{\text{M}}) \Big\|^2\Big]\mathrm{d}s \;.
\end{aligned}
$$

We now bound separately the two upper addends. To do so, we introduce the auxiliary measures $\lambda_k^h(\mathrm{d}s) \in \mathcal{P}([t_k, t_{k+1}])$ for $k = 0, ..., N-2$ and $\lambda_{N-1}^h(\mathrm{d}s) \in \mathcal{P}([1-h, 1-\epsilon])$ which will help us, via a double change of measure argument, to mitigate the bad behaviour at $t = 0$ and $t = 1$ of the reciprocal characteristic of the mimicking drift (i.e., $\partial_s + \mathcal{L}_s^{\text{M}}$), which is the trickiest addend. Namely, for $k = 0, ..., N-2$ we consider the measures $\lambda_k^h(\mathrm{d}s) \in \mathcal{P}([t_k, t_{k+1}])$ defined as

$$
\lambda_k^h(\mathrm{d}s) = \frac{\rho(s)^{-1}}{Z_k}\mathbb{1}_{[t_k, t_{k+1}]}\mathrm{d}s \;,
$$

with

$$
\rho(s)^{-1} = s^{-7/8}\mathbb{1}_{\{s\leq 1/2\}} + (1-s)^{-7/8}\mathbb{1}_{\{s>1/2\}} \;,
$$

and

$$
Z_k = \int_{\min\{t_k, 1/2\}}^{\min\{t_{k+1}, 1/2\}} r^{-7/8}\mathrm{d}r + \int_{\max\{t_k, 1/2\}}^{\max\{t_{k+1}, 1/2\}} (1-r)^{-7/8}\mathrm{d}r \;.
$$

Whereas, for $k = N - 1$, we consider the measures $\lambda^h_{N-1}(\mathrm{d}s) \in \mathcal{P}([1 - h, 1 - \epsilon])$ defined as

$$\lambda^h_{N-1}(\mathrm{d}s) = \frac{\rho(s)^{-1}}{Z_{N-1}} \mathbb{1}_{[1-h,1-\epsilon]} \mathrm{d}s \, ,$$

with $\rho(s)^{-1}$ as above and

$$Z_{N-1} = \int_{\min\{1-h,1/2\}}^{\min\{1-\epsilon,1/2\}} r^{-7/8} \mathrm{d}r + \int_{\max\{1-h,1/2\}}^{\max\{1-\epsilon,1/2\}} (1 - r)^{-7/8} \mathrm{d}r \, .$$

Note that, for any $s \in [0, 1 - \epsilon]$ and for any $k = 0, ..., N - 1$, they hold

$$\rho(s) \lesssim 1 \, , \quad Z_k \lesssim h^{1/8} \, , \quad \int_0^{1-\epsilon} \rho^{-1}(s)\mathrm{d}s = 16\left(\frac{1}{2}\right)^{1/8} - 8\epsilon^{1/8} \lesssim 1 \, . \tag{32}$$

We start by bounding the first addend, that is the one that involves the reciprocal characteristic of the mimicking drift. With a first change of measure argument, we get for any $k = 0, ..., N - 1$,

$$\mathbb{E}\left[\left\|\int_{t_k}^t (\partial_s + \mathcal{L}^{\mathrm{M}}_s)\tilde{\beta}_s(X^{\mathrm{M}}_s)\mathrm{d}s\right\|^2\right] = Z_k^2 \mathbb{E}\left[\left\|\int_{t_k}^t (\partial_s + \mathcal{L}^{\mathrm{M}}_s)\tilde{\beta}_s(X^{\mathrm{M}}_s)\rho(s)\lambda^h_k(\mathrm{d}s)\right\|^2\right] \, ,$$

where, in the last inequality, we used (32). But then, if we apply Jensen inequality and use an other change of measure argument, we get

$$\mathbb{E}\left[\left\|\int_{t_k}^t (\partial_s + \mathcal{L}^{\mathrm{M}}_s)\tilde{\beta}_s(X^{\mathrm{M}}_s)\mathrm{d}s\right\|^2\right] \leq Z_k^2 \mathbb{E}\left[\int_{t_k}^t \left\|(\partial_s + \mathcal{L}^{\mathrm{M}}_s)\tilde{\beta}_s(X^{\mathrm{M}}_s)\right\|^2 \rho(s)^2 \lambda^h_k(\mathrm{d}s)\right]$$

$$\leq Z_k \int_{t_k}^t \mathbb{E}\left[\left\|(\partial_s + \mathcal{L}^{\mathrm{M}}_s)\tilde{\beta}_s(X^{\mathrm{M}}_s)\right\|^2\right]\rho(s)\mathrm{d}s \tag{33}$$

$$\lesssim h^{1/8} \int_{t_k}^t \mathbb{E}\left[\left\|(\partial_s + \mathcal{L}^{\mathrm{M}}_s)\tilde{\beta}_s(X^{\mathrm{M}}_s)\right\|^2\right]\rho(s)\mathrm{d}s \, .$$

Let us now focus on the second addend. Remarkably, this addend can be bounded via the reciprocal characteristic of $\tilde{\beta}$: because of Ito's formula, for $t \in [0, 1 - \epsilon]$, it holds true

$$\mathrm{d}\left\|\tilde{\beta}_t(X^{\mathrm{M}}_t)\right\|^2 = \left\{2\langle\tilde{\beta}_t, (\partial_t + \mathcal{L}^{\mathrm{M}}_t)\tilde{\beta}_t\rangle + 2\left\|D_x\tilde{\beta}_t\right\|^2\right\}(X^{\mathrm{M}}_t)\mathrm{d}t + 2\sqrt{2}\langle\tilde{\beta}_t, D_x\tilde{\beta}_t\rangle(X^{\mathrm{M}}_t)\mathrm{d}B_t \, .$$

Consequently, if we assume that the process $(\int_0^s \langle\tilde{\beta}_t, D_x\tilde{\beta}_t\rangle(X^{\mathrm{M}}_t)\mathrm{d}B_t)_{s\in[0,1-\epsilon]}$ is a true martingale (see Lemma 4 below), we have that

$$2\int_{t_k}^t \mathbb{E}\left[\left\|D_x\tilde{\beta}_s(X^{\mathrm{M}}_s)\right\|^2\right]\mathrm{d}s$$

$$\leq \mathbb{E}\left[\left\|\tilde{\beta}_t(X^{\mathrm{M}}_t)\right\|^2\right] - \mathbb{E}\left[\left\|\tilde{\beta}_{t_k}(X^{\mathrm{M}}_{t_k})\right\|^2\right] + 2\left|\int_{t_k}^t \mathbb{E}[\langle\tilde{\beta}_s(X^{\mathrm{M}}_s), (\partial_s + \mathcal{L}^{\mathrm{M}}_s)\tilde{\beta}_s(X^{\mathrm{M}}_s)\rangle]\mathrm{d}s\right| \, .$$

But then, using, as before, a double change of measure argument and applying Cauchy-Schwartz inequality, we can bound the above expression as follows

$$2\int_{t_k}^t \mathbb{E}\left[\left\|D_x\tilde{\beta}_s(X^{\mathrm{M}}_s)\right\|^2\right]\mathrm{d}s$$

$$\leq \mathbb{E}\left[\left\|\tilde{\beta}_t(X^{\mathrm{M}}_t)\right\|^2\right] - \mathbb{E}\left[\left\|\tilde{\beta}_{t_k}(X^{\mathrm{M}}_{t_k})\right\|^2\right]$$

$$+ 2Z_k\left|\int_{t_k}^t \mathbb{E}[\langle\tilde{\beta}_s(X^{\mathrm{M}}_s), (\partial_s + \mathcal{L}^{\mathrm{M}}_s)\tilde{\beta}_s(X^{\mathrm{M}}_s)\rangle]\rho(s)\lambda^h_k(\mathrm{d}s)\right|$$

$$\leq \mathbb{E}\left[\left\|\tilde{\beta}_t(X^{\mathrm{M}}_t)\right\|^2\right] - \mathbb{E}\left[\left\|\tilde{\beta}_{t_k}(X^{\mathrm{M}}_{t_k})\right\|^2\right] + Z_k\int_{t_k}^t \mathbb{E}\left[\left\|\tilde{\beta}_s(X^{\mathrm{M}}_s)\right\|^2\right]\lambda^h_k(\mathrm{d}s)$$

$$+ Z_k\int_{t_k}^t \mathbb{E}\left[\left\|(\partial_s + \mathcal{L}^{\mathrm{M}}_s)\tilde{\beta}_s(X^{\mathrm{M}}_s)\right\|^2\right]\rho(s)^2\lambda^h_k(\mathrm{d}s)$$

$$= \mathbb{E}\left[\left\|\tilde{\beta}_t(X^{\mathrm{M}}_t)\right\|^2\right] - \mathbb{E}\left[\left\|\tilde{\beta}_{t_k}(X^{\mathrm{M}}_{t_k})\right\|^2\right] + \int_{t_k}^t \mathbb{E}\left[\left\|\tilde{\beta}_s(X^{\mathrm{M}}_s)\right\|^2\right]\rho(s)^{-1}\mathrm{d}s$$

$$+ \int_{t_k}^t \mathbb{E}\left[\left\|(\partial_s + \mathcal{L}^{\mathrm{M}}_s)\tilde{\beta}_s(X^{\mathrm{M}}_s)\right\|^2\right]\rho(s)\mathrm{d}s \, .$$

Plugging this bound and (33) in (30), we get

$$\mathrm{KL}(\nu^{\star}_{1-\epsilon}|\nu^{\theta^{\star}}_{1-\epsilon}) \lesssim \varepsilon^2 + h\mathbb{E}\Big[\Big\|\tilde{\beta}_{1-\epsilon}(X^{\mathrm{M}}_{1-\epsilon})\Big\|^2\Big] + h\int_0^{1-\epsilon}\mathbb{E}\Big[\Big\|\tilde{\beta}_s(X^{\mathrm{M}}_s)\Big\|^2\Big]\rho(s)^{-1}\mathrm{d}s$$

$$+ h(h^{1/8}+1)\int_0^{1-\epsilon}\mathbb{E}\Big[\Big\|(\partial_s + \mathcal{L}^{\mathrm{M}}_s)\tilde{\beta}_s(X^{\mathrm{M}}_s)\Big\|^2\Big]\rho(s)\mathrm{d}s \ . \quad (34)$$

We now compute explicitly and upper bound each term appearing in the RHS of (34), recalling that, because of Theorem 1, for any $s \in [0, 1-\epsilon]$, $\nu^{\star}_s = \mathrm{Law}(X^{\mathrm{I}}_s)$. We start with

$$\mathbb{E}\Big[\Big\|\tilde{\beta}_{1-\epsilon}(X^{\mathrm{M}}_{1-\epsilon})\Big\|^2\Big] \lesssim \int_{\mathbb{R}^d}\left\|\frac{\int_{\mathbb{R}^{2d}}p_{1-\epsilon}(x|x_0)\nabla_x p_\epsilon(x_1|x)\tilde{\pi}(x_0,x_1)\mathrm{d}x_1\mathrm{d}x_0}{p^{\mathrm{I}}_{1-\epsilon}(x)}\right\|^2 p^{\mathrm{I}}_{1-\epsilon}(x)\mathrm{d}x \ .$$

First we use (18), second we integrate by part, third we apply Jensen inequality and last we rely on **H 2**.

$$\mathbb{E}\Big[\Big\|\tilde{\beta}_{1-\epsilon}(X^{\mathrm{M}}_{1-\epsilon})\Big\|^2\Big]$$

$$\lesssim \int_{\mathbb{R}^d}\left\|\frac{\int_{\mathbb{R}^{2d}}p_{1-\epsilon}(x|x_0)p_\epsilon(x_1|x)(\nabla_{x_1}\tilde{\pi}(x_0,x_1)/\tilde{\pi}(x_0,x_1))\tilde{\pi}(x_0,x_1)\mathrm{d}x_0\mathrm{d}x_1}{p^{\mathrm{I}}_{1-\epsilon}(x)}\right\|^2 p^{\mathrm{I}}_{1-\epsilon}(x)\mathrm{d}x$$

$$= \mathbb{E}\left[\left\|\mathbb{E}\left[\frac{\nabla_{x_1}\tilde{\pi}}{\tilde{\pi}}(X^{\mathrm{I}}_0,X^{\mathrm{I}}_1)\Big|X^{\mathrm{I}}_{1-\epsilon}\right]\right\|^2\right] \leq \mathbb{E}\left[\mathbb{E}\left[\left\|\frac{\nabla_{x_1}\tilde{\pi}}{\tilde{\pi}}(X^{\mathrm{I}}_0,X^{\mathrm{I}}_1)\right\|^2\Big|X^{\mathrm{I}}_{1-\epsilon}\right]\right]$$

$$= \mathbb{E}\left[\left\|\frac{\nabla_{x_1}\tilde{\pi}}{\tilde{\pi}}(X^{\mathrm{I}}_0,X^{\mathrm{I}}_1)\right\|^2\right] = \left\|\frac{\nabla_{x_1}\tilde{\pi}}{\tilde{\pi}}\right\|^2_{\mathrm{L}^2(\pi)} \leq \|\nabla\log\tilde{\pi}\|^2_{\mathrm{L}^2(\pi)} \ .$$

$$(35)$$

In the very same way we deal with the third term of the RHS of (34).

$$\int_0^{1-\epsilon}\mathbb{E}\Big[\Big\|\tilde{\beta}_s(X^{\mathrm{M}}_s)\Big\|^2\Big]\rho(s)^{-1}\mathrm{d}s$$

$$\lesssim \int_0^{1-\epsilon}\rho(s)^{-1}\int_{\mathbb{R}^d}\left\|\frac{\int_{\mathbb{R}^{2d}}p_s(x|x_0)\nabla_x p_{1-s}(x_1|x)\tilde{\pi}(x_0,x_1)\mathrm{d}x_0\mathrm{d}x_1}{p^{\mathrm{I}}_s(x)}\right\|^2 p^{\mathrm{I}}_s(x)\mathrm{d}x\mathrm{d}s$$

$$= \int_0^{1-\epsilon}\rho(s)^{-1}\int_{\mathbb{R}^d}\left\|\frac{1}{p^{\mathrm{I}}_s(x)}\int_{\mathbb{R}^{2d}}p_s(x|x_0)p_{1-s}(x_1|x)\frac{\nabla_{x_1}\tilde{\pi}}{\tilde{\pi}}(x_0,x_1)\tilde{\pi}(x_0,x_1)\mathrm{d}x_0,\mathrm{d}x_1\right\|^2 p^{\mathrm{I}}_s(x)\mathrm{d}x$$

$$= \int_0^{1-\epsilon}\rho(s)^{-1}\mathbb{E}\left[\left\|\mathbb{E}\left[\frac{\nabla_{x_1}\tilde{\pi}}{\tilde{\pi}}(X^{\mathrm{I}}_0,X^{\mathrm{I}}_1)\Big|X^{\mathrm{I}}_s\right]\right\|^2\right]\mathrm{d}s$$

$$\leq \int_0^{1-\epsilon}\rho(s)^{-1}\mathbb{E}\left[\mathbb{E}\left[\left\|\frac{\nabla_{x_1}\tilde{\pi}}{\tilde{\pi}}(X^{\mathrm{I}}_0,X^{\mathrm{I}}_1)\right\|^2\Big|X^{\mathrm{I}}_s\right]\right]\mathrm{d}s$$

$$= \int_0^{1-\epsilon}\rho(s)^{-1}\mathbb{E}\left[\left\|\frac{\nabla_{x_1}\tilde{\pi}}{\tilde{\pi}}(X^{\mathrm{I}}_0,X^{\mathrm{I}}_1)\right\|^2\right]\mathrm{d}s$$

$$= \int_0^{1-\epsilon}\rho(s)^{-1}\mathrm{d}s\ \|\nabla\log\tilde{\pi}\|^2_{\mathrm{L}^2(\pi)} \lesssim \|\nabla\log\tilde{\pi}\|^2_{\mathrm{L}^2(\pi)} \ ,$$

$$(36)$$

where, in the last inequality, we used (32). We now turn to the last term, *i.e.*, to

$$\int_0^{1-\epsilon}\mathbb{E}\Big[\Big\|(\partial_s + \mathcal{L}^{\mathrm{M}}_s)\tilde{\beta}_s(X^{\mathrm{M}}_s)\Big\|^2\Big]\rho(s)\mathrm{d}s = \int_0^{1-\epsilon}\int_{\mathbb{R}^d}\Big\|(\partial_s + \mathcal{L}^{\mathrm{M}}_s)\tilde{\beta}_s(x)\Big\|^2\rho(s)p^{\mathrm{I}}_s(x)\mathrm{d}x\ \mathrm{d}s \ .$$

Some computations and (21) lead to

$$
\partial_s \tilde{\beta}_s(x)
$$
$$
= 2 \frac{\int_{\mathbb{R}^{2d}} \Delta_x p_s(x|x_0) \nabla_x p_{1-s}(x_1|x) \tilde{\pi}(x_0, x_1) \mathrm{d}x_0 \mathrm{d}x_1}{p_s^{\mathrm{I}}(x)}
$$
$$
- 2 \frac{\int_{\mathbb{R}^{2d}} p_s(x|x_0) \nabla_x \Delta_x p_{1-s}(x_1|x) \tilde{\pi}(x_0, x_1) \mathrm{d}x_0 \mathrm{d}x_1}{p_s^{\mathrm{I}}(x)}
$$
$$
- 2 \frac{\int_{\mathbb{R}^{2d}} p_s(x|x_0) \nabla_x p_{1-s}(x_1|x) \tilde{\pi}(x_0, x_1) \mathrm{d}x_0 \mathrm{d}x_1 \int_{\mathbb{R}^{2d}} \Delta_x p_s(x|x_0) p_{1-s}(x_1|x) \tilde{\pi}(x_0, x_1) \mathrm{d}x_0 \mathrm{d}x_1}{(p_s^{\mathrm{I}}(x))^2}
$$
$$
+ 2 \frac{\int_{\mathbb{R}^{2d}} p_s(x|x_0) \nabla_x p_{1-s}(x_1|x) \tilde{\pi}(x_0, x_1) \mathrm{d}x_0 \mathrm{d}x_1 \int_{\mathbb{R}^{2d}} p_s(x|x_0) \Delta_x p_{1-s}(x_1|x) \tilde{\pi}(x_0, x_1) \mathrm{d}x_0 \mathrm{d}x_1}{(p_s^{\mathrm{I}}(x))^2} \; ;
$$

$$
D_x \tilde{\beta}_s(x)
$$
$$
= 2 \frac{\int_{\mathbb{R}^{2d}} \nabla_x p_{1-s}(x_1|x) (\nabla_x p_s(x|x_0))^{\mathrm{T}} \tilde{\pi}(x_0, x_1) \mathrm{d}x_0 \mathrm{d}x_1}{p_s^{\mathrm{I}}(x)}
$$
$$
+ 2 \frac{\int_{\mathbb{R}^{2d}} p_s(x|x_0) \nabla_x^2 p_{1-s}(x_1|x) \tilde{\pi}(x_0, x_1) \mathrm{d}x_0 \mathrm{d}x_1}{p_s^{\mathrm{I}}(x)}
$$
$$
- 2 \left( \frac{\int_{\mathbb{R}^{2d}} p_s(x|x_0) \nabla_x p_{1-s}(x_1|x) \tilde{\pi}(x_0, x_1) \mathrm{d}x_0 \mathrm{d}x_1}{p_s^{\mathrm{I}}(x)} \right) \left( \frac{\int_{\mathbb{R}^{2d}} \nabla_x p_s(x|x_0) p_{1-s}(x_1|x) \tilde{\pi}(x_0, x_1) \mathrm{d}x_0 \mathrm{d}x_1}{p_s^{\mathrm{I}}(x)} \right)^{\mathrm{T}}
$$
$$
- 2 \left( \frac{\int_{\mathbb{R}^{2d}} p_s(x|x_0) \nabla_x p_{1-s}(x_1|x) \tilde{\pi}(x_0, x_1) \mathrm{d}x_0 \mathrm{d}x_1}{p_s^{\mathrm{I}}(x)} \right) \left( \frac{\int_{\mathbb{R}^{2d}} p_s(x|x_0) \nabla_x p_{1-s}(x_1|x) \tilde{\pi}(x_0, x_1) \mathrm{d}x_0 \mathrm{d}x_1}{p_s^{\mathrm{I}}(x)} \right)^{\mathrm{T}} ,
\tag{37}
$$

hence

$$
D_x \tilde{\beta}_s(x) \tilde{\beta}_s(x)
$$
$$
= 4 \frac{\int_{\mathbb{R}^{2d}} \nabla_x p_{1-s}(x_1|x) (\nabla_x p_s(x|x_0))^{\mathrm{T}} \tilde{\pi}(x_0, x_1) \mathrm{d}x_0 \mathrm{d}x_1}{p_s^{\mathrm{I}}(x)} \frac{\int_{\mathbb{R}^{2d}} p_s(x|x_0) \nabla_x p_{1-s}(x_1|x) \tilde{\pi}(x_0, x_1) \mathrm{d}x_0 \mathrm{d}x_1}{p_s^{\mathrm{I}}(x)}
$$
$$
+ 4 \frac{\int_{\mathbb{R}^{2d}} p_s(x|x_0) \nabla_x^2 p_{1-s}(x_1|x) \tilde{\pi}(x_0, x_1) \mathrm{d}x_0 \mathrm{d}x_1}{p_s^{\mathrm{I}}(x)} \frac{\int_{\mathbb{R}^{2d}} p_s(x|x_0) \nabla_x p_{1-s}(x_1|x) \tilde{\pi}(x_0, x_1) \mathrm{d}x_0 \mathrm{d}x_1}{p_s^{\mathrm{I}}(x)}
$$
$$
- 4 \left( \frac{\int_{\mathbb{R}^{2d}} p_s(x|x_0) \nabla_x p_{1-s}(x_1|x) \tilde{\pi}(x_0, x_1) \mathrm{d}x_0 \mathrm{d}x_1}{p_s^{\mathrm{I}}(x)} \right) \left( \frac{\int_{\mathbb{R}^{2d}} \nabla_x p_s(x|x_0) p_{1-s}(x_1|x) \tilde{\pi}(x_0, x_1) \mathrm{d}x_0 \mathrm{d}x_1}{p_s^{\mathrm{I}}(x)} \right)^{\mathrm{T}}
$$
$$
\cdot \frac{\int_{\mathbb{R}^{2d}} p_s(x|x_0) \nabla_x p_{1-s}(x_1|x) \tilde{\pi}(x_0, x_1) \mathrm{d}x_0 \mathrm{d}x_1}{p_s^{\mathrm{I}}(x)}
$$
$$
- 4 \frac{\int_{\mathbb{R}^{2d}} p_s(x|x_0) \nabla_x p_{1-s}(x_1|x) \tilde{\pi}(x_0, x_1) \mathrm{d}x_0 \mathrm{d}x_1}{p_s^{\mathrm{I}}(x)} \left\| \frac{\int_{\mathbb{R}^{2d}} p_s(x|x_0) \nabla_x p_{1-s}(x_1|x) \tilde{\pi}(x_0, x_1) \mathrm{d}x_0 \mathrm{d}x_1}{p_s^{\mathrm{I}}(x)} \right\|^2 \; ;
\tag{38}
$$

$$\Delta_x \tilde{\beta}_s(x)$$

$$= 2 \frac{\int_{\mathbb{R}^{2d}} \Delta_x p_s(x|x_0) \nabla_x p_{1-s}(x_1|x) \tilde{\pi}(x_0, x_1) \mathrm{d}x_0 \mathrm{d}x_1}{p_s^{\mathrm{I}}(x)}$$

$$+ 4 \frac{\int_{\mathbb{R}^{2d}} \nabla_x^2 p_{1-s}(x|x_0) \nabla_x p_s(x_1|x) \tilde{\pi}(x_0, x_1) \mathrm{d}x_0 \mathrm{d}x_1}{p_s^{\mathrm{I}}(x)}$$

$$- 4 \frac{\int_{\mathbb{R}^{2d}} \nabla_x p_{1-s}(x_1|x) (\nabla_x p_s(x|x_0))^{\mathrm{T}} \tilde{\pi}(x_0, x_1) \mathrm{d}x_0 \mathrm{d}x_1}{p_s^{\mathrm{I}}(x)}$$

$$\cdot \frac{\int_{\mathbb{R}^{2d}} \nabla_x p_s(x|x_0) p_{1-s}(x_1|x) \tilde{\pi}(x_0, x_1) \mathrm{d}x_0 \mathrm{d}x_1}{p_s^{\mathrm{I}}(x)}$$

$$- 4 \frac{\int_{\mathbb{R}^{2d}} \nabla_x p_{1-s}(x_1|x) (\nabla_x p_s(x|x_0))^{\mathrm{T}} \tilde{\pi}(x_0, x_1) \mathrm{d}x_0 \mathrm{d}x_1}{p_s^{\mathrm{I}}(x)}$$

$$\cdot \frac{\int_{\mathbb{R}^{2d}} p_s(x|x_0) \nabla_x p_{1-s}(x_1|x) \tilde{\pi}(x_0, x_1) \mathrm{d}x_0 \mathrm{d}x_1}{p_s^{\mathrm{I}}(x)}$$

$$- 4 \frac{\int_{\mathbb{R}^{2d}} \langle \nabla_x p_{1-s}(x_1|x), \nabla_x p_s(x|x_0) \rangle \tilde{\pi}(x_0, x_1) \mathrm{d}x_0 \mathrm{d}x_1}{p_s^{\mathrm{I}}(x)}$$

$$\cdot \frac{\int_{\mathbb{R}^{2d}} p_s(x|x_0) \nabla_x p_{1-s}(x_1|x) \tilde{\pi}(x_0, x_1) \mathrm{d}x_0 \mathrm{d}x_1}{p_s^{\mathrm{I}}(x)}$$

$$+ 2 \frac{\int_{\mathbb{R}^{2d}} p_s(x|x_0) \nabla_x \Delta_x p_{1-s}(x_1|x) \tilde{\pi}(x_0, x_1) \mathrm{d}x_0 \mathrm{d}x_1}{p_s^{\mathrm{I}}(x)}$$

$$- 4 \frac{\int_{\mathbb{R}^{2d}} p_s(x|x_0) \nabla_x^2 p_{1-s}(x_1|x) \tilde{\pi}(x_0, x_1) \mathrm{d}x_0 \mathrm{d}x_1 \int_{\mathbb{R}^{2d}} \nabla_x p_s(x|x_0) p_{1-s}(x_1|x) \tilde{\pi}(x_0, x_1) \mathrm{d}x_0 \mathrm{d}x_1}{(p_s^{\mathrm{I}}(x))^2}$$

$$- 4 \frac{\int_{\mathbb{R}^{2d}} p_s(x|x_0) \nabla_x^2 p_{1-s}(x_1|x) \tilde{\pi}(x_0, x_1) \mathrm{d}x_0 \mathrm{d}x_1 \int_{\mathbb{R}^{2d}} p_s(x|x_0) \nabla_x p_{1-s}(x_1|x) \tilde{\pi}(x_0, x_1) \mathrm{d}x_0 \mathrm{d}x_1}{(p_s^{\mathrm{I}}(x))^2}$$

$$- 2 \frac{\int_{\mathbb{R}^{2d}} \Delta_x p_s(x|x_0) p_{1-s}(x_1|x) \tilde{\pi}(x_0, x_1) \mathrm{d}x_0 \mathrm{d}x_1 \int_{\mathbb{R}^{2d}} p_s(x|x_0) \nabla_x p_{1-s}(x_1|x) \tilde{\pi}(x_0, x_1) \mathrm{d}x_0 \mathrm{d}x_1}{(p_s^{\mathrm{I}}(x))^2}$$

$$- 2 \frac{\int_{\mathbb{R}^{2d}} p_s(x|x_0) \Delta_x p_{1-s}(x_1|x) \tilde{\pi}(x_0, x_1) \mathrm{d}x_0 \mathrm{d}x_1 \int_{\mathbb{R}^{2d}} p_s(x|x_0) \nabla_x p_{1-s}(x_1|x) \tilde{\pi}(x_0, x_1) \mathrm{d}x_0 \mathrm{d}x_1}{(p_s^{\mathrm{I}}(x))^2}$$

$$+ 8 \frac{\int_{\mathbb{R}^{2d}} p_s(x|x_0) \nabla_x p_{1-s}(x_1|x) \tilde{\pi}(x_0, x_1) \mathrm{d}x_0 \mathrm{d}x_1}{p_s^{\mathrm{I}}(x)} \left( \frac{\int_{\mathbb{R}^{2d}} \nabla_x p_s(x|x_0) p_{1-s}(x_1|x) \tilde{\pi}(x_0, x_1) \mathrm{d}x_0 \mathrm{d}x_1}{p_s^{\mathrm{I}}(x)} \right)^{\mathrm{T}}$$

$$\cdot \frac{\int_{\mathbb{R}^{2d}} p_s(x|x_0) \nabla_x p_{1-s}(x_1|x) \tilde{\pi}(x_0, x_1) \mathrm{d}x_0 \mathrm{d}x_1}{p_s^{\mathrm{I}}(x)}$$

$$+ 4 \left\| \frac{\int_{\mathbb{R}^{2d}} p_s(x|x_0) \nabla_x p_{1-s}(x_1|x) \tilde{\pi}(x_0, x_1) \mathrm{d}x_0 \mathrm{d}x_1}{p_s^{\mathrm{I}}(x)} \right\|^2 \cdot \frac{\int_{\mathbb{R}^{2d}} p_s(x|x_0) \nabla_x p_{1-s}(x_1|x) \tilde{\pi}(x_0, x_1) \mathrm{d}x_0 \mathrm{d}x_1}{p_s^{\mathrm{I}}(x)}$$

$$+ 4 \left\| \frac{\int_{\mathbb{R}^{2d}} \nabla_x p_s(x|x_0) p_{1-s}(x_1|x) \tilde{\pi}(x_0, x_1) \mathrm{d}x_0 \mathrm{d}x_1}{p_s^{\mathrm{I}}(x)} \right\|^2 \cdot \frac{\int_{\mathbb{R}^{2d}} p_s(x|x_0) \nabla_x p_{1-s}(x_1|x) \tilde{\pi}(x_0, x_1) \mathrm{d}x_0 \mathrm{d}x_1}{p_s^{\mathrm{I}}(x)} .$$

So, we get

$$(\partial_s + \mathcal{L}_s^{\mathrm{M}}) \tilde{\beta}_s(x) = \sum_{k=1}^{6} A_s^k(x) , \tag{39}$$

where we have defined

$$A_s^1(x) = -4 \frac{\int_{\mathbb{R}^{2d}} \nabla_x p_{1-s}(x_1|x) (\nabla_x p_s(x|x_0))^{\mathrm{T}} \tilde{\pi}(x_0, x_1) \mathrm{d}x_0 \mathrm{d}x_1}{p_s^{\mathrm{I}}(x)}$$

$$\cdot \frac{\int_{\mathbb{R}^{2d}} \nabla_x p_s(x|x_0) p_{1-s}(x_1|x) \tilde{\pi}(x_0, x_1) \mathrm{d}x_0 \mathrm{d}x_1}{p_s^{\mathrm{I}}(x)} ,$$

$$A_s^2(x) = -4\frac{\int_{\mathbb{R}^{2d}}\langle\nabla_x p_{1-s}(x_1|x), \nabla_x p_s(x|x_0)\rangle\tilde{\pi}(x_0,x_1)\mathrm{d}x_0\mathrm{d}x_1}{p_s^{\mathrm{I}}(x)}$$
$$\cdot\frac{\int_{\mathbb{R}^{2d}} p_s(x|x_0)\nabla_x p_{1-s}(x_1|x)\tilde{\pi}(x_0,x_1)\mathrm{d}x_0\mathrm{d}x_1}{p_s^{\mathrm{I}}(x)},$$

$$A_s^3(x) = 4\frac{\left\|\int_{\mathbb{R}^{2d}}\nabla_x p_s(x|x_0)p_{1-s}(x_1|x)\tilde{\pi}(x_0,x_1)\mathrm{d}x_0\mathrm{d}x_1\right\|^2}{(p_s^{\mathrm{I}}(x))^2}$$
$$\cdot\frac{\int_{\mathbb{R}^{2d}} p_s(x|x_0)\nabla_x p_{1-s}(x_1|x)\tilde{\pi}(x_0,x_1)\mathrm{d}x_0\mathrm{d}x_1}{p_s^{\mathrm{I}}(x)},$$

$$A_s^4(x) = 4\frac{\int_{\mathbb{R}^{2d}} p_s(x|x_0)\nabla_x p_{1-s}(x_1|x)\tilde{\pi}(x_0,x_1)\mathrm{d}x_0\mathrm{d}x_1}{p_s^{\mathrm{I}}(x)}$$
$$\cdot\left(\frac{\int_{\mathbb{R}^{2d}}\nabla_x p_s(x|x_0)p_{1-s}(x_1|x)\tilde{\pi}(x_0,x_1)\mathrm{d}x_0\mathrm{d}x_1}{p_s^{\mathrm{I}}(x)}\right)^{\mathrm{T}}\frac{\int_{\mathbb{R}^{2d}} p_s(x|x_0)\nabla_x p_{1-s}(x_1|x)\tilde{\pi}(x_0,x_1)\mathrm{d}x_0\mathrm{d}x_1}{p_s^{\mathrm{I}}(x)}.$$

$$A_s^5(x) = 4\frac{\int_{\mathbb{R}^{2d}}\Delta_x p_s(x|x_0)\nabla_x p_{1-s}(x_1|x)\tilde{\pi}(x_0,x_1)\mathrm{d}x_0\mathrm{d}x_1}{p_s^{\mathrm{I}}(x)}$$
$$-4\frac{\int_{\mathbb{R}^{2d}}\Delta_x p_s(x|x_0)p_{1-s}(x_1|x)\tilde{\pi}(x_0,x_1)\mathrm{d}x_0\mathrm{d}x_1\int_{\mathbb{R}^{2d}} p_s(x|x_0)\nabla_x p_{1-s}(x_1|x)\tilde{\pi}(x_0,x_1)\mathrm{d}x_0\mathrm{d}x_1}{(p_s^{\mathrm{I}}(x))^2},$$

$$A_s^6(x) = 4\frac{\int_{\mathbb{R}^{2d}}\nabla_x^2 p_{1-s}(x_1|x)\nabla_x p_s(x|x_0)\tilde{\pi}(x_0,x_1)\mathrm{d}x_0\mathrm{d}x_1}{p_s^{\mathrm{I}}(x)}$$
$$-4\frac{\int_{\mathbb{R}^{2d}} p_s(x|x_0)\nabla_x^2 p_{1-s}(x_1|x)\tilde{\pi}(x_0,x_1)\mathrm{d}x_0\mathrm{d}x_1\int_{\mathbb{R}^{2d}}\nabla_x p_s(x|x_0)p_{1-s}(x_1|x)\tilde{\pi}(x_0,x_1)\mathrm{d}x_0\mathrm{d}x_1}{(p_s^{\mathrm{I}}(x))^2}.$$

Therefore

$$\int_0^{1-\epsilon}\mathbb{E}\Big[\big\|(\partial_s + \mathcal{L}_s^{\mathrm{M}})\tilde{\beta}_s(X_s^{\mathrm{M}})\big\|^2\Big]\rho(s)\mathrm{d}s \lesssim \sum_{k=1}^6\int_0^{1-\epsilon}\int_{\mathbb{R}^d}\big\|A_s^k(x)\big\|^2\rho(s)p_s^{\mathrm{I}}(x)\mathrm{d}x\ \mathrm{d}s.$$

We now bound each term $A_s^k$ in the sum.
Using (32) and Young inequality, we have

$$\int_0^{1-\epsilon}\int_{\mathbb{R}^d}\big\|A_s^1(x)\big\|^2\rho(s)p_s^{\mathrm{I}}(x)\mathrm{d}x\ \mathrm{d}s$$
$$\lesssim \int_0^{1-\epsilon}\int_{\mathbb{R}^d}\left\|\frac{\int_{\mathbb{R}^{2d}}\nabla_x p_s(x|x_0)p_{1-s}(x_1|x)\tilde{\pi}(x_0,x_1)\mathrm{d}x_0\mathrm{d}x_1}{p_s^{\mathrm{I}}(x)}\right\|^4 p_s^{\mathrm{I}}(x)\mathrm{d}x\mathrm{d}s$$
$$+\int_0^{1-\epsilon}\int_{\mathbb{R}^d}\left\|\frac{\int_{\mathbb{R}^{2d}}\nabla_x p_s(x|x_0)(\nabla_x p_{1-s}(x_1|x))^{\mathrm{T}}\tilde{\pi}(x_0,x_1)\mathrm{d}x_0\mathrm{d}x_1}{p_s^{\mathrm{I}}(x)}\right\|_{\mathrm{op}}^4 p_s^{\mathrm{I}}(x)\mathrm{d}x\mathrm{d}s.$$

To bound the first term we proceed as in (36), that is, we first exploit (18) and the integration by part formula and we then use Jensen inequality and the properties of the conditional expectation.

$$\int_0^{1-\epsilon}\int_{\mathbb{R}^d}\left\|\frac{\int_{\mathbb{R}^{2d}}\nabla_x p_s(x|x_0)p_{1-s}(x_1|x)\tilde{\pi}(x_0,x_1)\mathrm{d}x_0\mathrm{d}x_1}{p_s^{\mathrm{I}}(x)}\right\|^4 p_s^{\mathrm{I}}(x)\mathrm{d}x\mathrm{d}s \leq \|\nabla\log\tilde{\pi}\|_{\mathrm{L}^4(\pi)}^4.$$
$$\tag{40}$$

To bound the second term we first split the time interval $[0, 1 - \epsilon]$ in two, $[0, 1/2]$ and $[1/2, 1 - \epsilon]$, we second make either $\nabla_x p_s(x|x_0)$ or $\nabla_x p_{1-s}(x_1|x)$ explicit and we last proceed as before, *i.e.*, we exploit (18), we integrate by parts and we use Young and Jensen inequalities.

$$
\int_0^{1-\epsilon} \int_{\mathbb{R}^d} \left\| \frac{\int_{\mathbb{R}^{2d}} \nabla_x p_{1-s}(x_1|x)(\nabla_x p_s(x|x_0))^{\mathrm{T}} \tilde{\pi}(x_0, x_1) \mathrm{d}x_0 \mathrm{d}x_1}{p_s^{\mathrm{I}}(x)} \right\|_{\mathrm{op}}^4 p_s^{\mathrm{I}}(x) \mathrm{d}x \mathrm{d}s
$$

$$
= \int_0^{1/2} \int_{\mathbb{R}^d} \left\| \frac{\int_{\mathbb{R}^{2d}} \nabla_x p_s(x|x_0)(\nabla_x p_{1-s}(x_1|x))^{\mathrm{T}} \tilde{\pi}(x_0, x_1) \mathrm{d}x_0 \mathrm{d}x_1}{p_s^{\mathrm{I}}(x)} \right\|_{\mathrm{op}}^4 p_s^{\mathrm{I}}(x) \mathrm{d}x \mathrm{d}s
$$

$$
+ \int_{1/2}^{1-\epsilon} \int_{\mathbb{R}^d} \left\| \frac{\int_{\mathbb{R}^{2d}} \nabla_x p_{1-s}(x_1|x)(\nabla_x p_s(x|x_0))^{\mathrm{T}} \tilde{\pi}(x_0, x_1) \mathrm{d}x_0 \mathrm{d}x_1}{p_s^{\mathrm{I}}(x)} \right\|_{\mathrm{op}}^4 p_s^{\mathrm{I}}(x) \mathrm{d}x \mathrm{d}s
$$

$$
\lesssim \int_0^{1/2} \int_{\mathbb{R}^d} \left\| \frac{1}{p_s^{\mathrm{I}}(x)} \int_{\mathbb{R}^{2d}} \frac{\nabla_{x_0} \tilde{\pi}}{\tilde{\pi}}(x_0, x_1) \frac{(x_1 - x)^{\mathrm{T}}}{1 - s} p_s(x|x_0) p_{1-s}(x_1|x) \tilde{\pi}(x_0, x_1) \mathrm{d}x_0 \mathrm{d}x_1 \right\|_{\mathrm{op}}^4
$$
$$
\cdot p_s^{\mathrm{I}}(x) \mathrm{d}x \mathrm{d}s
$$

$$
+ \int_{1/2}^{1-\epsilon} \int_{\mathbb{R}^d} \left\| \frac{1}{p_s^{\mathrm{I}}(x)} \int_{\mathbb{R}^{2d}} \frac{\nabla_{x_1} \tilde{\pi}}{\tilde{\pi}}(x_0, x_1) \frac{(x - x_0)^{\mathrm{T}}}{s} p_s(x|x_0) p_{1-s}(x_1|x) \tilde{\pi}(x_0, x_1) \mathrm{d}x_0 \mathrm{d}x_1 \right\|_{\mathrm{op}}^4
$$
$$
\cdot p_s^{\mathrm{I}}(x) \mathrm{d}x \mathrm{d}s
$$

$$
\lesssim \int_0^{1/2} \mathbb{E}\left[ \left\| \left( \frac{\nabla_{x_0} \tilde{\pi}}{\tilde{\pi}}(X_0^{\mathrm{I}}, X_1^{\mathrm{I}}) \right)(X_1^{\mathrm{I}} - X_s^{\mathrm{I}})^{\mathrm{T}} \right\|_{\mathrm{op}}^4 \right] \mathrm{d}s
$$

$$
+ \int_{1/2}^1 \mathbb{E}\left[ \left\| \left( \frac{\nabla_{x_1} \tilde{\pi}}{\tilde{\pi}}(X_0^{\mathrm{I}}, X_1^{\mathrm{I}}) \right)(X_s^{\mathrm{I}} - X_0^{\mathrm{I}})^{\mathrm{T}} \right\|_{\mathrm{op}}^4 \right] \mathrm{d}s
$$

$$
\lesssim \int_0^{1/2} \mathbb{E}\left[ \left\| X_1^{\mathrm{I}} - X_s^{\mathrm{I}} \right\|^8 \right] \mathrm{d}s + \int_{1/2}^1 \mathbb{E}\left[ \left\| X_s^{\mathrm{I}} - X_0^{\mathrm{I}} \right\|^8 \right] \mathrm{d}s + \|\nabla \log \tilde{\pi}\|_{\mathrm{L}^8(\pi)}^8
$$

$$
\lesssim d^4 + \mathbf{m}_8[\mu] + \mathbf{m}_8[\nu^\star] + \|\nabla \log \tilde{\pi}\|_{\mathrm{L}^8(\pi)}^8 \ ,
$$

where, in the last inequality, we have used Lemma 1.
In a similar way we get

$$
\int_0^{1-\epsilon} \int_{\mathbb{R}^d} \left\| A_s^2(x) \right\|^2 \rho(s) p_s^{\mathrm{I}}(x) \mathrm{d}x \ \mathrm{d}s \lesssim d^4 + \mathbf{m}_8[\mu] + \mathbf{m}_8[\nu^\star] + \|\nabla \log \tilde{\pi}\|_{\mathrm{L}^8(\pi)}^8 \ .
$$

The argument to bound $A_s^2$ resembles the one used to bound $A_s^1$ and we therefore omit it.

We now focus on $A_s^3$. To bound such term, we first use (32) and Young inequality and we second proceed as in (40) and (36).

$$
\int_0^{1-\epsilon} \int_{\mathbb{R}^d} \left\| A_s^3(x) \right\|^2 \rho(s) p_s^{\mathrm{I}}(x) \mathrm{d}x \ \mathrm{d}s
$$

$$
\lesssim \int_0^{1-\epsilon} \int_{\mathbb{R}^d} \left\| \frac{\int_{\mathbb{R}^{2d}} \nabla_x p_s(x|x_0) p_{1-s}(x_1|x) \tilde{\pi}(x_0, x_1) \mathrm{d}x_0 \mathrm{d}x_1}{p_s^{\mathrm{I}}(x)} \right\|^8 p_s^{\mathrm{I}}(x) \mathrm{d}x \mathrm{d}s
$$

$$
+ \int_0^{1-\epsilon} \int_{\mathbb{R}^d} \left\| \frac{\int_{\mathbb{R}^{2d}} p_s(x|x_0) \nabla_x p_{1-s}(x_1|x) \tilde{\pi}(x_0, x_1) \mathrm{d}x_0 \mathrm{d}x_1}{p_s^{\mathrm{I}}(x)} \right\|^4 p_s^{\mathrm{I}}(x) \mathrm{d}x \mathrm{d}s
$$

$$
\lesssim \mathbb{E}\left[ \left\| \frac{\nabla_{x_0} \tilde{\pi}}{\tilde{\pi}}(X_0^{\mathrm{I}}, X_1^{\mathrm{I}}) \right\|^8 \right] + \mathbb{E}\left[ \left\| \frac{\nabla_{x_1} \tilde{\pi}}{\tilde{\pi}}(X_0^{\mathrm{I}}, X_1^{\mathrm{I}}) \right\|^4 \right]
$$

$$
\lesssim \|\nabla \log \tilde{\pi}\|_{\mathrm{L}^8(\pi)}^8 + \|\nabla \log \tilde{\pi}\|_{\mathrm{L}^4(\pi)}^4 \ .
$$

Proceeding in a similar way (we omit the argument, as it is almost a duplication of the previous one), we get

$$\int_0^{1-\epsilon} \int_{\mathbb{R}^d} \left\| A_s^4(x) \right\|^2 \rho(s) p_s^{\mathrm{I}}(x) \mathrm{d}x \, \mathrm{d}s \lesssim \| \nabla \log \tilde{\pi} \|_{\mathrm{L}^8(\pi)}^8 + \| \nabla \log \tilde{\pi} \|_{\mathrm{L}^4(\pi)}^4 \ .$$

We now turn to $A_s^5$. Because of (20), $A_s^5$ rewrites as

$$
\begin{aligned}
A_s^5(x) = {} & \frac{1}{p_s^{\mathrm{I}}(x)} \int_{\mathbb{R}^{2d}} \left\{ \frac{-d}{2s} + \frac{\|x - x_0\|^2}{4s^2} \right\} \frac{x_1 - x}{2(1-s)} p_s(x|x_0) p_{1-s}(x_1|x) \tilde{\pi}(x_0, x_1) \mathrm{d}x_0 \mathrm{d}x_1 \\
& - \left( \frac{1}{p_s^{\mathrm{I}}(x)} \int_{\mathbb{R}^{2d}} \left\{ \frac{-d}{2s} + \frac{\|x - x_0\|^2}{4s^2} \right\} p_s(x|x_0) p_{1-s}(x_1|x) \tilde{\pi}(x_0, x_1) \mathrm{d}x_0 \mathrm{d}x_1 \right) \\
& \cdot \left( \frac{1}{p_s^{\mathrm{I}}(x)} \int_{\mathbb{R}^{2d}} \frac{x_1 - x}{2(1-s)} p_s(x|x_0) p_{1-s}(x_1|x) \tilde{\pi}(x_0, x_1) \mathrm{d}x_0 \mathrm{d}x_1 \right) \\
\lesssim {} & \frac{1}{p_s^{\mathrm{I}}(x)} \int_{\mathbb{R}^{2d}} \frac{\|x - x_0\|^2}{s^2} \frac{x_1 - x}{1 - s} p_s(x|x_0) p_{1-s}(x_1|x) \tilde{\pi}(x_0, x_1) \mathrm{d}x_0 \mathrm{d}x_1 \\
& - \left( \frac{1}{p_s^{\mathrm{I}}(x)} \int_{\mathbb{R}^{2d}} \frac{\|x - x_0\|^2}{s^2} p_s(x|x_0) p_{1-s}(x_1|x) \tilde{\pi}(x_0, x_1) \mathrm{d}x_0 \mathrm{d}x_1 \right) \\
& \cdot \left( \frac{1}{p_s^{\mathrm{I}}(x)} \int_{\mathbb{R}^{2d}} \frac{x_1 - x}{1 - s} p_s(x|x_0) p_{1-s}(x_1|x) \tilde{\pi}(x_0, x_1) \mathrm{d}x_0 \mathrm{d}x_1 \right) \ .
\end{aligned}
$$

Therefore, we have that

$$
\begin{aligned}
\int_0^{1-\epsilon} \int_{\mathbb{R}^d} \left\| A_s^5(x) \right\|^2 \rho(s) p_s^{\mathrm{I}}(x) \mathrm{d}x \, \mathrm{d}s \lesssim {} & \int_0^{1-\epsilon} \mathbb{E}\left[ \left\| \mathbb{E}\left[ \frac{\|X_s^{\mathrm{I}} - X_0^{\mathrm{I}}\|^2}{s^2} \frac{X_1^{\mathrm{I}} - X_s^{\mathrm{I}}}{1 - s} \Big| X_s^{\mathrm{I}} \right] \right. \right. \\
& \left. \left. - \mathbb{E}\left[ \frac{\|X_s^{\mathrm{I}} - X_0^{\mathrm{I}}\|^2}{s^2} \Big| X_s^{\mathrm{I}} \right] \mathbb{E}\left[ \frac{X_1^{\mathrm{I}} - X_s^{\mathrm{I}}}{1 - s} \Big| X_s^{\mathrm{I}} \right] \right\|^2 \right] \rho(s) \mathrm{d}s \ .
\end{aligned}
$$

We now split the time interval $[0, 1 - \epsilon]$ in two, $[0, 1/2]$, $[1/2, 1 - \epsilon]$ and we focus on the first one. So we look at $s \in [0, 1/2]$ and we try to bound the integrand. Using Lemma 1, Lemma 2, Lemma 3 and standard and well-known inequalities (Cauchy-Schwarz, Young and Jensen inequalities), we get

that for $s \in [0, 1/2]$

$$\mathbb{E}\left[\left\|\mathbb{E}\left[\frac{\left\|X_s^{\mathrm{I}} - X_0^{\mathrm{I}}\right\|^2}{s^2}\frac{X_1^{\mathrm{I}} - X_s^{\mathrm{I}}}{1-s}\Big| X_s^{\mathrm{I}}\right] - \mathbb{E}\left[\frac{\left\|X_s^{\mathrm{I}} - X_0^{\mathrm{I}}\right\|^2}{s^2}\Big| X_s^{\mathrm{I}}\right]\mathbb{E}\left[\frac{X_1^{\mathrm{I}} - X_s^{\mathrm{I}}}{1-s}\Big| X_s^{\mathrm{I}}\right]\right\|^2\right]s^{7/8}$$

$$= \mathbb{E}\Bigg[\Bigg\|\mathbb{E}\left[\frac{\left\|\overleftarrow{X}_{1-s} - \overleftarrow{X}_1\right\|^2}{s^2}\frac{\overleftarrow{X}_0 - \overleftarrow{X}_{1-s}}{1-s}\Big| \overleftarrow{X}_{1-s}\right]$$

$$-\mathbb{E}\left[\frac{\left\|\overleftarrow{X}_{1-s} - \overleftarrow{X}_1\right\|^2}{s^2}\Big| \overleftarrow{X}_{1-s}\right]\mathbb{E}\left[\frac{\overleftarrow{X}_0 - \overleftarrow{X}_{1-s}}{1-s}\Big| \overleftarrow{X}_{1-s}\right]\Bigg\|^2\Bigg]s^{7/8}$$

$$= \mathbb{E}\Bigg[\Bigg\|\mathbb{E}\left[\frac{\left\|\overleftarrow{f}_{1-s}^{\,s}\right\|^2 + 2\langle\overleftarrow{f}_{1-s}^{\,s}, \overleftarrow{g}_{1-s}^{\,s}\rangle + \left\|\overleftarrow{g}_{1-s}^{\,s}\right\|^2}{s^2}(\overleftarrow{X}_0 - \overleftarrow{X}_{1-s})\Big| \overleftarrow{X}_{1-s}\right]$$

$$-\mathbb{E}\left[\frac{\left\|\overleftarrow{f}_{1-s}^{\,s}\right\|^2 + 2\langle\overleftarrow{f}_{1-s}^{\,s}, \overleftarrow{g}_{1-s}^{\,s}\rangle + \left\|\overleftarrow{g}_{1-s}^{\,s}\right\|^2}{s^2}\Big| \overleftarrow{X}_{1-s}\right]\mathbb{E}\left[\overleftarrow{X}_0 - \overleftarrow{X}_{1-s}\Big| \overleftarrow{X}_{1-s}\right]\Bigg\|^2\Bigg]s^{7/8}$$

$$= \mathbb{E}\Bigg[\Bigg\|\mathbb{E}\left[\frac{\left\|\overleftarrow{f}_{1-s}^{\,s}\right\|^2 + 2\langle\overleftarrow{f}_{1-s}^{\,s}, \overleftarrow{g}_{1-s}^{\,s}\rangle}{s^2}(\overleftarrow{X}_0 - \overleftarrow{X}_{1-s})\Big| \overleftarrow{X}_{1-s}\right]$$

$$-\mathbb{E}\left[\frac{\left\|\overleftarrow{f}_{1-s}^{\,s}\right\|^2 + 2\langle\overleftarrow{f}_{1-s}^{\,s}, \overleftarrow{g}_{1-s}^{\,s}\rangle}{s^2}\Big| \overleftarrow{X}_{1-s}\right]\mathbb{E}\left[\overleftarrow{X}_0 - \overleftarrow{X}_{1-s}\Big| \overleftarrow{X}_{1-s}\right]\Bigg\|^2\Bigg]s^{7/8}$$

$$\lesssim \mathbb{E}\Bigg[\Bigg\|\mathbb{E}\left[\frac{\left\|\overleftarrow{f}_{1-s}^{\,s}\right\|^2 + 2\langle\overleftarrow{f}_{1-s}^{\,s}, \overleftarrow{g}_{1-s}^{\,s}\rangle}{s^2}(\overleftarrow{X}_0 - \overleftarrow{X}_{1-s})\Big| \overleftarrow{X}_{1-s}\right]\Bigg\|^2\Bigg]s^{7/8}$$

$$+ \mathbb{E}\Bigg[\Bigg\|\mathbb{E}\left[\frac{\left\|\overleftarrow{f}_{1-s}^{\,s}\right\|^2 + 2\langle\overleftarrow{f}_{1-s}^{\,s}, \overleftarrow{g}_{1-s}^{\,s}\rangle}{s^2}\Big| \overleftarrow{X}_{1-s}\right]\mathbb{E}\left[\overleftarrow{X}_0 - \overleftarrow{X}_{1-s}\Big| \overleftarrow{X}_{1-s}\right]\Bigg\|^2\Bigg]s^{7/8}$$

$$\lesssim \mathbb{E}\Bigg[\Bigg\|\frac{\left\|\overleftarrow{f}_{1-s}^{\,s}\right\|^2 + 2\langle\overleftarrow{f}_{1-s}^{\,s}, \overleftarrow{g}_{1-s}^{\,s}\rangle}{s^2}(\overleftarrow{X}_0 - \overleftarrow{X}_{1-s})\Bigg\|^2\Bigg]s^{7/8}$$

$$+ \mathbb{E}\Bigg[\Bigg\|\mathbb{E}\left[\frac{\left\|\overleftarrow{f}_{1-s}^{\,s}\right\|^2 + 2\langle\overleftarrow{f}_{1-s}^{\,s}, \overleftarrow{g}_{1-s}^{\,s}\rangle}{s^2}\Big| \overleftarrow{X}_{1-s}\right]\Bigg\|^4\Bigg]s^{7/4} + \mathbb{E}\left[\left\|\mathbb{E}\left[\overleftarrow{X}_0 - \overleftarrow{X}_{1-s}\Big| \overleftarrow{X}_{1-s}\right]\right\|^4\right]$$

$$\lesssim \mathbb{E}\Bigg[\Bigg\|\frac{\left\|\overleftarrow{f}_{1-s}^{\,s}\right\|^2 + 2\langle\overleftarrow{f}_{1-s}^{\,s}, \overleftarrow{g}_{1-s}^{\,s}\rangle}{s^2}\Bigg\|^4\Bigg]s^{7/4} + \mathbb{E}\left[\left\|\overleftarrow{X}_0 - \overleftarrow{X}_{1-s}\right\|^4\right]$$

$$\lesssim \mathbb{E}\left[\frac{\left\|\overleftarrow{f}_{1-s}^{\,s}\right\|^8}{s^8}\right] + \mathbb{E}\left[\frac{\langle\overleftarrow{f}_{1-s}^{\,s}, \overleftarrow{g}_{1-s}^{\,s}\rangle^4}{s^8}\right]s^{7/4} + \mathbb{E}\left[\left\|\overleftarrow{X}_0 - \overleftarrow{X}_{1-s}\right\|^4\right]$$

$$\lesssim \mathbb{E}\left[\frac{\left\|\overleftarrow{f}_{1-s}^{\,s}\right\|^8}{s^8}\right] + \mathbb{E}\left[\frac{\left\|\overleftarrow{g}_{1-s}^{\,s}\right\|^8}{s^8}\right] s^{7/2} + \mathbb{E}\left[\left\|\overleftarrow{X}_0 - \overleftarrow{X}_{1-s}\right\|^4\right]$$

$$\lesssim \mathbb{E}\left[\frac{\left\|\overleftarrow{f}_{1-s}^{\,s}\right\|^8}{s^8}\right] + d^4 s^{-1/2} + \mathbb{E}\left[\left\|X_1^{\mathrm{I}} - X_s^{\mathrm{I}}\right\|^4\right]$$

$$\lesssim \|\nabla \log \tilde{\pi}\|_{\mathrm{L}^8(\pi)}^8 + \|\nabla \log \nu^\star\|_{\mathrm{L}^8(\nu^\star)}^8 + d^4 s^{-1/2} + d^2 + \mathbf{m}_4[\mu] + \mathbf{m}_4[\nu^\star] .$$

But then, we obtain that

$$\int_0^{1/2} \int_{\mathbb{R}^d} \left\|A_s^5(x)\right\|^2 \rho(s) p_s^{\mathrm{I}}(x) \mathrm{d}x \, \mathrm{d}s \lesssim d^4 + \mathbf{m}_4[\mu] + \mathbf{m}_4[\nu^\star] + \|\nabla \log \tilde{\pi}\|_{\mathrm{L}^8(\pi)}^8$$
$$+ \|\nabla \log \nu^\star\|_{\mathrm{L}^8(\nu^\star)}^8 .$$

We now focus on the second time interval, that is we look at $s \in [1/2, 1 - \epsilon]$ and try to bound the integrand. Using (32) and proceeding in a similar way, we get that

$$\mathbb{E}\left[\left\|\mathbb{E}\left[\frac{\|X_s^{\mathrm{I}} - X_0^{\mathrm{I}}\|^2}{s^2}\frac{X_1^{\mathrm{I}} - X_s^{\mathrm{I}}}{1-s}\Big| X_s^{\mathrm{I}}\right] - \mathbb{E}\left[\frac{\|X_s^{\mathrm{I}} - X_0^{\mathrm{I}}\|^2}{s^2}\Big| X_s^{\mathrm{I}}\right]\mathbb{E}\left[\frac{X_1^{\mathrm{I}} - X_s^{\mathrm{I}}}{1-s}\Big| X_s^{\mathrm{I}}\right]\right\|^2\right] \rho(s)$$

$$\lesssim \mathbb{E}\left[\left\|\mathbb{E}\left[\left\|\overrightarrow{X}_s - \overrightarrow{X}_0\right\|^2 \frac{\overrightarrow{f}_s^{\,1-s} + \overrightarrow{g}_s^{\,1-s}}{1-s}\Big|\overrightarrow{X}_s\right]\right.\right.$$

$$\left.\left. - \mathbb{E}\left[\left\|\overrightarrow{X}_s - \overrightarrow{X}_0\right\|^2\Big|\overrightarrow{X}_s\right]\mathbb{E}\left[\frac{\overrightarrow{f}_s^{\,1-s} + \overrightarrow{g}_s^{\,1-s}}{1-s}\Big|\overrightarrow{X}_s\right]\right\|^2\right]$$

$$= \mathbb{E}\left[\left\|\mathbb{E}\left[\left\|\overrightarrow{X}_s - \overrightarrow{X}_0\right\|^2 \frac{\overrightarrow{f}_s^{\,1-s}}{1-s}\Big|\overrightarrow{X}_s\right] - \mathbb{E}\left[\left\|\overrightarrow{X}_s - \overrightarrow{X}_0\right\|^2\Big|\overrightarrow{X}_s\right]\mathbb{E}\left[\frac{\overrightarrow{f}_s^{\,1-s}}{1-s}\Big|\overrightarrow{X}_s\right]\right\|^2\right]$$

$$\lesssim \mathbb{E}\left[\left\|\overrightarrow{X}_s - \overrightarrow{X}_0\right\|^8\right] + \mathbb{E}\left[\frac{\left\|\overrightarrow{f}_s^{\,1-s}\right\|^4}{(1-s)^4}\right] = \mathbb{E}\left[\left\|X_s^{\mathrm{I}} - X_0^{\mathrm{I}}\right\|^8\right] + \mathbb{E}\left[\frac{\left\|\overrightarrow{f}_s^{\,1-s}\right\|^4}{(1-s)^4}\right]$$

$$\lesssim d^4 + \mathbf{m}_8[\mu] + \mathbf{m}_8[\nu^\star] + \|\nabla \log \tilde{\pi}\|_{\mathrm{L}^4(\pi)}^4 + \|\nabla \log \mu\|_{\mathrm{L}^4(\mu)}^4 .$$

But then, we have that

$$\int_{1/2}^{1-\epsilon} \int_{\mathbb{R}^d} \left\|A_s^5(x)\right\|^2 \rho(s) p_s^{\mathrm{I}}(x) \mathrm{d}x \, \mathrm{d}s \lesssim d^4 + \mathbf{m}_8[\mu] + \mathbf{m}_8[\nu^\star] + \|\nabla \log \tilde{\pi}\|_{\mathrm{L}^8(\pi)}^8$$
$$+ \|\nabla \log \mu\|_{\mathrm{L}^8(\mu)}^8 .$$

To conclude, we can bound $A_s^5$ as follows

$$\int_0^{1-\epsilon} \int_{\mathbb{R}^d} \left\|A_s^5(x)\right\|^2 \rho(s) p_s^{\mathrm{I}}(x) \mathrm{d}x \, \mathrm{d}s \lesssim d^4 + \mathbf{m}_8[\mu] + \mathbf{m}_8[\nu^\star] + \|\nabla \log \tilde{\pi}\|_{\mathrm{L}^8(\pi)}^8$$
$$+ \|\nabla \log \mu\|_{\mathrm{L}^8(\mu)}^8 + \|\nabla \log \nu^\star\|_{\mathrm{L}^8(\nu^\star)}^8 .$$

We are left with $A_s^6$. Using (19), we can rewrite $A_s^6$ as follows

$$A_s^6(x)$$
$$= \frac{1}{p_s^{\mathrm{I}}(x)} \int_{\mathbb{R}^{2d}} \left\{ \frac{-1}{2(1-s)} \mathrm{Id} + \frac{(x_1-x)(x_1-x)^{\mathrm{T}}}{4(1-s)^2} \right\} \frac{x-x_0}{2s} p_s(x|x_0)p_{1-s}(x_1|x)\tilde{\pi}(x_0,x_1)\mathrm{d}x_0\mathrm{d}x_1$$
$$- \left( \frac{1}{p_s^{\mathrm{I}}(x)} \int_{\mathbb{R}^{2d}} \left\{ \frac{-1}{2(1-s)} \mathrm{Id} + \frac{(x_1-x)(x_1-x)^{\mathrm{T}}}{4(1-s)^2} \right\} p_s(x|x_0)p_{1-s}(x_1|x)\tilde{\pi}(x_0,x_1)\mathrm{d}x_0\mathrm{d}x_1 \right)$$
$$\cdot \left( \frac{1}{p_s^{\mathrm{I}}(x)} \int_{\mathbb{R}^{2d}} \frac{x-x_0}{2s} p_s(x|x_0)p_{1-s}(x_1|x)\tilde{\pi}(x_0,x_1)\mathrm{d}x_0\mathrm{d}x_1 \right)$$
$$\lesssim \left( \frac{1}{p_s^{\mathrm{I}}(x)} \int_{\mathbb{R}^{2d}} \frac{(x_1-x)(x_1-x)^{\mathrm{T}}}{(1-s)^2} \frac{x-x_0}{s} p_s(x|x_0)p_{1-s}(x_1|x)\tilde{\pi}(x_0,x_1)\mathrm{d}x_0\mathrm{d}x_1 \right.$$
$$\left. - \left( \frac{1}{p_s^{\mathrm{I}}(x)} \int_{\mathbb{R}^{2d}} \frac{(x_1-x)(x_1-x)^{\mathrm{T}}}{(1-s)^2} p_s(x|x_0)p_{1-s}(x_1|x)\tilde{\pi}(x_0,x_1)\mathrm{d}x_0\mathrm{d}x_1 \right) \right.$$
$$\left. \cdot \left( \frac{1}{p_s^{\mathrm{I}}(x)} \int_{\mathbb{R}^{2d}} \frac{x-x_0}{s} p_s(x|x_0)p_{1-s}(x_1|x)\tilde{\pi}(x_0,x_1)\mathrm{d}x_0\mathrm{d}x_1 \right) \right. .$$

It follows that

$$\int_0^{1-\epsilon} \int_{\mathbb{R}^d} \left\| A_s^6(x) \right\|^2 \rho(s)p_s^{\mathrm{I}}(x)\mathrm{d}x\,\mathrm{d}s$$
$$\lesssim \int_0^{1-\epsilon} \mathbb{E}\left[ \left\| \mathbb{E}\left[ \frac{(X_1^{\mathrm{I}} - X_s^{\mathrm{I}})(X_1^{\mathrm{I}} - X_s^{\mathrm{I}})^{\mathrm{T}}}{(1-s)^2} \frac{X_s^{\mathrm{I}} - X_0^{\mathrm{I}}}{s} \middle| X_s^{\mathrm{I}} \right] \right. \right.$$
$$\left. \left. - \mathbb{E}\left[ \frac{(X_1^{\mathrm{I}} - X_s^{\mathrm{I}})(X_1^{\mathrm{I}} - X_s^{\mathrm{I}})^{\mathrm{T}}}{(1-s)^2} \middle| X_s^{\mathrm{I}} \right] \mathbb{E}\left[ \frac{X_s^{\mathrm{I}} - X_0^{\mathrm{I}}}{s} \middle| X_s^{\mathrm{I}} \right] \right\|^2 \right] \rho(s)\mathrm{d}s$$

At this point, we proceed as for $A_s^5$, that is we split the time interval in two and we use Lemma 1, Lemma 2 and Lemma 3. By doing so and by using (32), we get that for $s \in [0, 1/2]$

$$\mathbb{E}\left[ \left\| \mathbb{E}\left[ \frac{(X_1^{\mathrm{I}} - X_s^{\mathrm{I}})(X_1^{\mathrm{I}} - X_s^{\mathrm{I}})^{\mathrm{T}}}{(1-s)^2} \frac{X_s^{\mathrm{I}} - X_0^{\mathrm{I}}}{s} \middle| X_s^{\mathrm{I}} \right] \right. \right.$$
$$\left. \left. - \mathbb{E}\left[ \frac{(X_1^{\mathrm{I}} - X_s^{\mathrm{I}})(X_1^{\mathrm{I}} - X_s^{\mathrm{I}})^{\mathrm{T}}}{(1-s)^2} \middle| X_s^{\mathrm{I}} \right] \mathbb{E}\left[ \frac{X_s^{\mathrm{I}} - X_0^{\mathrm{I}}}{s} \middle| X_s^{\mathrm{I}} \right] \right\|^2 \right] \rho(s)$$
$$\lesssim \mathbb{E}\left[ \left\| \mathbb{E}\left[ (\overleftarrow{X}_0 - \overleftarrow{X}_{1-s})(\overleftarrow{X}_0 - \overleftarrow{X}_{1-s})^{\mathrm{T}} \frac{\overleftarrow{X}_1 - \overleftarrow{X}_{1-s}}{s} \middle| \overleftarrow{X}_{1-s} \right] \right. \right.$$
$$\left. \left. - \mathbb{E}\left[ (\overleftarrow{X}_0 - \overleftarrow{X}_{1-s})(\overleftarrow{X}_0 - \overleftarrow{X}_{1-s})^{\mathrm{T}} \middle| \overleftarrow{X}_{1-s} \right] \mathbb{E}\left[ \frac{\overleftarrow{X}_1 - \overleftarrow{X}_{1-s}}{s} \middle| \overleftarrow{X}_{1-s} \right] \right\|^2 \right]$$
$$= \mathbb{E}\left[ \left\| \mathbb{E}\left[ (\overleftarrow{X}_0 - \overleftarrow{X}_{1-s})(\overleftarrow{X}_0 - \overleftarrow{X}_{1-s})^{\mathrm{T}} \frac{\overleftarrow{f}_{1-s}^{\,s}}{s} \middle| \overleftarrow{X}_{1-s} \right] \right. \right.$$
$$\left. \left. - \mathbb{E}\left[ (\overleftarrow{X}_0 - \overleftarrow{X}_{1-s})(\overleftarrow{X}_0 - \overleftarrow{X}_{1-s})^{\mathrm{T}} \middle| \overleftarrow{X}_{1-s} \right] \mathbb{E}\left[ \frac{\overleftarrow{f}_{1-s}^{\,s}}{s} \middle| \overleftarrow{X}_{1-s} \right] \right\|^2 \right]$$
$$\lesssim \mathbb{E}\left[ \left\| (\overleftarrow{X}_0 - \overleftarrow{X}_{1-s})(\overleftarrow{X}_0 - \overleftarrow{X}_{1-s})^{\mathrm{T}} \right\|_{\mathrm{op}}^4 \right] + \mathbb{E}\left[ \left\| \frac{\overleftarrow{f}_{1-s}^{\,s}}{s} \right\|^4 \right]$$
$$= \mathbb{E}[\| X_1^{\mathrm{I}} - X_s^{\mathrm{I}} \|^8] + \mathbb{E}\left[ \left\| \frac{\overleftarrow{f}_{1-s}^{\,s}}{s} \right\|^4 \right]$$
$$\lesssim d^4 + \mathbf{m}_8[\mu] + \mathbf{m}_8[\nu^\star] + \| \nabla \log \tilde{\pi} \|_{\mathrm{L}^8(\pi)}^8 + \| \nabla \log \nu^\star \|_{\mathrm{L}^8(\nu^\star)}^8 \ .$$

Whereas, for $s \in [1/2, 1-\epsilon]$, we get

$$
\mathbb{E}\Bigg[\Bigg\|\mathbb{E}\Bigg[\frac{(X_1^{\mathrm{I}} - X_s^{\mathrm{I}})(X_1^{\mathrm{I}} - X_s^{\mathrm{I}})^{\mathrm{T}}}{(1-s)^2}\frac{X_s^{\mathrm{I}} - X_0^{\mathrm{I}}}{s}\Bigg|X_s^{\mathrm{I}}\Bigg]
$$
$$
- \mathbb{E}\Bigg[\frac{(X_1^{\mathrm{I}} - X_s^{\mathrm{I}})(X_1^{\mathrm{I}} - X_s^{\mathrm{I}})^{\mathrm{T}}}{(1-s)^2}\Bigg|X_s^{\mathrm{I}}\Bigg]\mathbb{E}\Bigg[\frac{X_s^{\mathrm{I}} - X_0^{\mathrm{I}}}{s}\Bigg|X_s^{\mathrm{I}}\Bigg]\Bigg\|^2\Bigg]\rho(s)
$$
$$
\lesssim \mathbb{E}\Bigg[\Bigg\|\mathbb{E}\Bigg[\frac{(\overrightarrow{X}_1 - \overrightarrow{X}_s)(\overrightarrow{X}_1 - \overrightarrow{X}_s)^{\mathrm{T}}}{(1-s)^2}(\overrightarrow{X}_s - \overrightarrow{X}_0)\Bigg|\overrightarrow{X}_s\Bigg]
$$
$$
- \mathbb{E}\Bigg[\frac{(\overrightarrow{X}_1 - \overrightarrow{X}_s)(\overrightarrow{X}_1 - \overrightarrow{X}_s)^{\mathrm{T}}}{(1-s)^2}\Bigg|\overrightarrow{X}_s\Bigg]\mathbb{E}\Big[\overrightarrow{X}_s - \overrightarrow{X}_0\Big|\overrightarrow{X}_s\Big]\Bigg\|^2\Bigg](1-s)^{7/8}
$$
$$
= (1-s)^{7/8}\mathbb{E}\Bigg[\Bigg\|\mathbb{E}\Bigg[\frac{\overrightarrow{f}_s^{1-s}(\overrightarrow{f}_s^{1-s})^{\mathrm{T}} + \overrightarrow{f}_s^{1-s}(\overrightarrow{g}_s^{1-s})^{\mathrm{T}} + \overrightarrow{g}_s^{1-s}(\overrightarrow{f}_s^{1-s})^{\mathrm{T}}}{(1-s)^2}(\overrightarrow{X}_s - \overrightarrow{X}_0)\Bigg|\overrightarrow{X}_s\Bigg]
$$
$$
- \mathbb{E}\Bigg[\frac{\overrightarrow{f}_s^{1-s}(\overrightarrow{f}_s^{1-s})^{\mathrm{T}} + \overrightarrow{f}_s^{1-s}(\overrightarrow{g}_s^{1-s})^{\mathrm{T}} + \overrightarrow{g}_s^{1-s}(\overrightarrow{f}_s^{1-s})^{\mathrm{T}}}{(1-s)^2}\Bigg|\overrightarrow{X}_s\Bigg]\mathbb{E}\Big[\overrightarrow{X}_s - \overrightarrow{X}_0\Big|\overrightarrow{X}_s\Big]\Bigg\|^2\Bigg]
$$
$$
\lesssim \mathbb{E}\Bigg[\frac{\big\|\overrightarrow{f}_s^{1-s}(\overrightarrow{f}_s^{1-s})^{\mathrm{T}} + \overrightarrow{f}_s^{1-s}(\overrightarrow{g}_s^{1-s})^{\mathrm{T}} + \overrightarrow{g}_s^{1-s}(\overrightarrow{f}_s^{1-s})^{\mathrm{T}}\big\|_{\mathrm{op}}^4}{(1-s)^8}\Bigg](1-s)^{7/4} + \mathbb{E}\Big[\big\|\overrightarrow{X}_s - \overrightarrow{X}_0\big\|^4\Big]
$$
$$
\lesssim \mathbb{E}\Bigg[\frac{\big\|\overrightarrow{f}_s^{1-s}\big\|^8}{(1-s)^8}\Bigg] + \mathbb{E}\Bigg[\frac{\big\|\overrightarrow{g}_s^{1-s}\big\|^8}{(1-s)^8}\Bigg](1-s)^{7/2} + + \mathbb{E}\Big[\big\|X_s^{\mathrm{I}} - X_0^{\mathrm{I}}\big\|^4\Big]
$$
$$
\lesssim \|\nabla\log\tilde{\pi}\|_{\mathrm{L}^8(\pi)}^8 + \|\nabla\log\mu\|_{\mathrm{L}^8(\mu)}^8 + d^4(1-s)^{-1/2} + d^2 + \mathbf{m}_4[\mu] + \mathbf{m}_4[\nu^\star].
$$

Consequently, we have that

$$
\int_0^{1-\epsilon}\int_{\mathbb{R}^d}\big\|A_s^6(x)\big\|^2\rho(s)p_s^{\mathrm{I}}(x)\mathrm{d}x\,\mathrm{d}s \lesssim d^4 + \mathbf{m}_8[\mu] + \mathbf{m}_8[\nu^\star] + \|\nabla\log\tilde{\pi}\|_{\mathrm{L}^8(\pi)}^8
$$
$$
+ \|\nabla\log\mu\|_{\mathrm{L}^8(\mu)}^8 + \|\nabla\log\nu^\star\|_{\mathrm{L}^8(\nu^\star)}^8.
$$

Putting together the bounds on the $\{A_s^k\}_{k=1}^6$ derived so far, we eventually obtain

$$
\int_0^{1-\epsilon}\mathbb{E}\Big[\big\|(\partial_s + \mathcal{L}_s^{\mathrm{M}})\tilde{\beta}_s(\overrightarrow{X}_s)\big\|^2\Big]\rho(s)\mathrm{d}s
$$
$$
\lesssim d^4 + \mathbf{m}_8[\mu] + \mathbf{m}_8[\nu^\star] + \|\nabla\log\tilde{\pi}\|_{\mathrm{L}^8(\pi)}^8 + \|\nabla\log\mu\|_{\mathrm{L}^8(\mu)}^8 + \|\nabla\log\nu^\star\|_{\mathrm{L}^8(\nu^\star)}^8.
\tag{41}
$$

Plugging (41), (35) and (36) into (34), we get

$$
\mathrm{KL}(\nu_{1-\epsilon}^\star|\nu_{1-\epsilon}^{\theta^\star}) \lesssim \varepsilon^2 + h(h^{1/8}+1)\Big(d^4 + \mathbf{m}_8[\mu] + \mathbf{m}_8[\nu^\star] + \|\nabla\log\tilde{\pi}\|_{\mathrm{L}^8(\pi)}^8
$$
$$
+ \|\nabla\log\mu\|_{\mathrm{L}^8(\mu)}^8 + \|\nabla\log\nu^\star\|_{\mathrm{L}^8(\nu^\star)}^8\Big).
$$

The estimate (14) then follows from the above estimate and (31).
However, for (14) to hold true, we still need to prove that

**Lemma 4.** $(\int_0^s\langle\tilde{\beta}_t, D_x\tilde{\beta}_t\rangle(X_t^{\mathrm{M}})\mathrm{d}B_t)_{s\in[0,1-\epsilon]}$ *is a martingale.*

*Proof of Lemma 4.* If we show that for any $s \in [0, 1-\epsilon]$, it holds $\mathbb{E}[\|\langle\tilde{\beta}_s, D_x\tilde{\beta}_s\rangle(X_t^{\mathrm{M}})\|^2] < C$, for some $C > 0$ independent of time, then by Fubini's theorem and [BB17, Theorem 7.3] we are done. To do so, by the Cauchy-Schwarz inequality, we just need to show that $\mathbb{E}[\|\tilde{\beta}_s(X_s^{\mathrm{M}})\|^4]$ and

$\mathbb{E}[\|D_x\tilde{\beta}_s(X_s^{\mathrm{M}})\|_{\mathrm{op}}^4]$ are bounded from above by constants which are independent of time. To this aim, note that, as a direct consequence of (8), Theorem 1, (18), Jensen inequality and **H**2, it holds

$$
\begin{aligned}
&\mathbb{E}\left[\left\|\tilde{\beta}_s(X_s^{\mathrm{M}})\right\|^4\right] \\
&\lesssim \int_{\mathbb{R}^d}\left\|\frac{\int_{\mathbb{R}^{2d}} p_s(x|x_0)\nabla_x p_{1-s}(x_1|x)\tilde{\pi}(x_0,x_1)\mathrm{d}x_0\mathrm{d}x_1}{p_s^{\mathrm{I}}(x)}\right\|^4 p_s^{\mathrm{I}}(x)\mathrm{d}x \\
&= \int_{\mathbb{R}^d}\left\|\frac{1}{p_s^{\mathrm{I}}(x)}\int_{\mathbb{R}^{2d}} p_s(x|x_0)p_{1-s}(x_1|x)\frac{\nabla_{x_1}\tilde{\pi}}{\tilde{\pi}}(x_0,x_1)\tilde{\pi}(x_0,x_1)\mathrm{d}x_0,\mathrm{d}x_1\right\|^4 p_s^{\mathrm{I}}(x)\mathrm{d}x \\
&= \mathbb{E}\left[\left\|\mathbb{E}\left[\frac{\nabla_{x_1}\tilde{\pi}}{\tilde{\pi}}(X_0^{\mathrm{I}},X_1^{\mathrm{I}})\Big|X_s^{\mathrm{I}}\right]\right\|^4\right] \leq \mathbb{E}\left[\mathbb{E}\left[\left\|\frac{\nabla_{x_1}\tilde{\pi}}{\tilde{\pi}}(X_0^{\mathrm{I}},X_1^{\mathrm{I}})\right\|^4\Big|X_s^{\mathrm{I}}\right]\right] \\
&= \mathbb{E}\left[\left\|\frac{\nabla_{x_1}\tilde{\pi}}{\tilde{\pi}}(X_0^{\mathrm{I}},X_1^{\mathrm{I}})\right\|^4\right] = \|\nabla\log\tilde{\pi}\|_{\mathrm{L}^4(\pi)}^4 \ .
\end{aligned}
\tag{42}
$$

Similarly, recalling (38), we have that

$$
\begin{aligned}
&\mathbb{E}\left[\left\|D_x\tilde{\beta}_s(X_s^{\mathrm{M}})\right\|_{\mathrm{op}}^4\right] \\
&\lesssim \int_{\mathbb{R}^d}\left\|\frac{\int_{\mathbb{R}^{2d}}\nabla_x p_{1-s}(x_1|x)(\nabla_x p_s(x|x_0))^{\mathrm{T}}\tilde{\pi}(x_0,x_1)\mathrm{d}x_0\mathrm{d}x_1}{p_s^{\mathrm{I}}(x)}\right\|_{\mathrm{op}}^4 p_s^{\mathrm{I}}(x)\mathrm{d}x \\
&+ \int_{\mathbb{R}^d}\left\|\frac{\int_{\mathbb{R}^{2d}} p_s(x|x_0)\nabla_x^2 p_{1-s}(x_1|x)\tilde{\pi}(x_0,x_1)\mathrm{d}x_0\mathrm{d}x_1}{p_s^{\mathrm{I}}(x)}\right\|_{\mathrm{op}}^4 p_s^{\mathrm{I}}(x)\mathrm{d}x \\
&+ \int_{\mathbb{R}^d}\left\|\frac{\int_{\mathbb{R}^{2d}} p_s(x|x_0)\nabla_x p_{1-s}(x_1|x)\tilde{\pi}(x_0,x_1)\mathrm{d}x_0\mathrm{d}x_1}{p_s^{\mathrm{I}}(x)}\right\|_{\mathrm{op}}^8 p_s^{\mathrm{I}}(x)\mathrm{d}x \\
&+ \int_{\mathbb{R}^d}\left\|\frac{\int_{\mathbb{R}^{2d}}\nabla_x p_s(x|x_0)p_{1-s}(x_1|x)\tilde{\pi}(x_0,x_1)\mathrm{d}x_0\mathrm{d}x_1}{p_s^{\mathrm{I}}(x)}\right\|_{\mathrm{op}}^8 p_s^{\mathrm{I}}(x)\mathrm{d}x \ .
\end{aligned}
$$

To bound the first term of the RHS of the above expression, we integrate by parts and use Lemma 1 and **H**2.

$$
\begin{aligned}
&\int_{\mathbb{R}^d}\left\|\frac{\int_{\mathbb{R}^{2d}}\nabla_x p_{1-s}(x_1|x)(\nabla_x p_s(x|x_0))^{\mathrm{T}}\tilde{\pi}(x_0,x_1)\mathrm{d}x_0\mathrm{d}x_1}{p_s^{\mathrm{I}}(x)}\right\|_{\mathrm{op}}^4 p_s^{\mathrm{I}}(x)\mathrm{d}x \\
&= \int_{\mathbb{R}^d}\left\|\frac{\int_{\mathbb{R}^{2d}}\nabla_x p_s(x|x_0)(\nabla_x p_{1-s}(x_1|x))^{\mathrm{T}}\tilde{\pi}(x_0,x_1)\mathrm{d}x_0\mathrm{d}x_1}{p_s^{\mathrm{I}}(x)}\right\|_{\mathrm{op}}^4 p_s^{\mathrm{I}}(x)\mathrm{d}x \\
&= \int_{\mathbb{R}^d}\left\|\frac{1}{p_s^{\mathrm{I}}(x)}\int_{\mathbb{R}^{2d}}\frac{\nabla_{x_0}\tilde{\pi}}{\tilde{\pi}}(x_0,x_1)\frac{(x_1-x)^{\mathrm{T}}}{1-s}p_s(x|x_0)p_{1-s}(x_1|x)\tilde{\pi}(x_0,x_1)\mathrm{d}x_0\mathrm{d}x_1\right\|_{\mathrm{op}}^4 p_s^{\mathrm{I}}(x)\mathrm{d}x \\
&\lesssim \mathbb{E}\left[\left\|\frac{\nabla_{x_0}\tilde{\pi}}{\tilde{\pi}}(X_0^{\mathrm{I}},X_1^{\mathrm{I}})\frac{(X_1^{\mathrm{I}}-X_s^{\mathrm{I}})^{\mathrm{T}}}{\epsilon}\right\|_{\mathrm{op}}^4\right] \\
&\lesssim \frac{1}{\epsilon}(d^4+\mathbf{m}_8[\mu]+\mathbf{m}_8[\nu^\star])+\|\nabla\log\tilde{\pi}\|_{\mathrm{L}^8(\pi)}^8 \ .
\end{aligned}
$$

To bound the second term, we integrate by parts and proceed as before.

$$\int_{\mathbb{R}^d} \left\| \frac{\int_{\mathbb{R}^{2d}} p_s(x|x_0)\nabla_x^2 p_{1-s}(x_1|x)\tilde\pi(x_0,x_1)\mathrm{d}x_0\mathrm{d}x_1}{p_s^{\mathrm{I}}(x)} \right\|_{\mathrm{op}}^4 p_s^{\mathrm{I}}(x)\mathrm{d}x$$

$$= \int_{\mathbb{R}^d} \left\| \frac{\int_{\mathbb{R}^{2d}} p_s(x|x_0)(\nabla_{x_1}\tilde\pi(x_0,x_1)/\tilde\pi(x_0,x_1))(\nabla_x p_{1-s}(x_1|x))^{\mathrm{T}}\ \tilde\pi(x_0,x_1)\mathrm{d}x_0\mathrm{d}x_1}{p_s^{\mathrm{I}}(x)} \right\|_{\mathrm{op}}^4$$
$$p_s^{\mathrm{I}}(x)\mathrm{d}x$$

$$= \int_{\mathbb{R}^d} \left\| \frac{1}{p_s^{\mathrm{I}}(x)} \int_{\mathbb{R}^{2d}} p_s(x|x_0)p_{1-s}(x_1|x)\frac{\nabla_{x_1}\tilde\pi}{\tilde\pi}(x_0,x_1)\frac{(x-x_1)^{\mathrm{T}}}{1-s}\tilde\pi(x_0,x_1)\mathrm{d}x_0\mathrm{d}x_1 \right\|_{\mathrm{op}}^4$$
$$p_s^{\mathrm{I}}(x)\mathrm{d}x$$

$$\lesssim \frac{1}{\epsilon}(d^4 + \mathbf{m}_8[\mu] + \mathbf{m}_8[\nu^\star]) + \|\nabla\log\tilde\pi\|_{\mathrm{L}^8(\pi)}^8 \ .$$

To bound the third and last term, we proceed as in (42), getting

$$\int_{\mathbb{R}^d} \left\| \frac{\int_{\mathbb{R}^{2d}} p_s(x|x_0)\nabla_x p_{1-s}(x_1|x)\tilde\pi(x_0,x_1)\mathrm{d}x_0\mathrm{d}x_1}{p_s^{\mathrm{I}}(x)} \right\|^8 p_s^{\mathrm{I}}(x)\mathrm{d}x \lesssim \|\nabla\log\tilde\pi\|_{\mathrm{L}^8(\pi)}^8 \ ,$$

and

$$\int_{\mathbb{R}^d} \left\| \frac{\int_{\mathbb{R}^{2d}} \nabla_x p_s(x|x_0)p_{1-s}(x_1|x)\tilde\pi(x_0,x_1)\mathrm{d}x_0\mathrm{d}x_1}{p_s^{\mathrm{I}}(x)} \right\|^8 p_s^{\mathrm{I}}(x)\mathrm{d}x \lesssim \|\nabla\log\tilde\pi\|_{\mathrm{L}^8(\pi)}^8 \ .$$

$$\square$$

## A.5    Proof of Theorem 3

Fix $\delta > 0$. Then, because of Theorem 1, (5) and **H**1, it holds

$$\mathbf{m}_8[\nu_{1-\delta}^\star] = \mathbb{E}[\|X_{1-\delta}^{\mathrm{I}}\|^8] \lesssim \delta^8 \mathbf{m}_8[\mu] + (1-\delta)^8 \mathbf{m}_8[\nu^\star] + d^4\delta^4(1-\delta)^4 < +\infty \ . \tag{43}$$

Moreover $\nu_{1-\delta}^\star \ll \mathrm{Leb}^d$ with density $p_{1-\delta}^{\mathrm{I}}$ and

$$\nabla\log\left(\frac{\mathrm{d}\nu_{1-\delta}^\star}{\mathrm{dLeb}^d}\right) \in \mathrm{L}^8(\nu_{1-\delta}^\star) \ . \tag{44}$$

Indeed because of (18), it holds

$$\nabla\log\frac{\mathrm{d}\nu_{1-\delta}^\star}{\mathrm{dLeb}^d}(x_{1-\delta})$$

$$= \frac{\nabla\left(\int_{\mathbb{R}^{2d}} p_{1-\delta}(x_{1-\delta}|x_0)p_\delta(x_1|x_{1-\delta})\tilde\pi(\mathrm{d}x_0,\mathrm{d}x_1)\right)}{p_{1-\delta}^{\mathrm{I}}(x_{1-\delta})}$$

$$= \frac{1}{p_{1-\delta}^{\mathrm{I}}(x_{1-\delta})} \int_{\mathbb{R}^{2d}} \left\{\frac{x_{1-\delta}-x_0}{1-\delta} + \frac{x_1-x_{1-\delta}}{\delta}\right\} p_{1-\delta}(x_{1-\delta}|x_0)p_\delta(x_1|x_{1-\delta})\tilde\pi(\mathrm{d}x_0,\mathrm{d}x_1)$$

$$= \frac{1}{p_{1-\delta}^{\mathrm{I}}(x_{1-\delta})} \int_{\mathbb{R}^{2d}} \frac{(2\delta-1)x_{1-\delta}-\delta x_0+(1-\delta)x_1}{\delta(1-\delta)} p_{1-\delta}(x_{1-\delta}|x_0)p_\delta(x_1|x_{1-\delta})\tilde\pi(\mathrm{d}x_0,\mathrm{d}x_1)$$

$$\lesssim \mathbb{E}\left[\frac{X_{1-\delta}^{\mathrm{I}}-\delta X_0^{\mathrm{I}}+(1-\delta)X_1^{\mathrm{I}}}{\delta(1-\delta)}\middle| X_{1-\delta}^{\mathrm{I}}=x_{1-\delta}\right] \ .$$

But then, using Jensen inequality we obtain that

$$\int_{\mathbb{R}^d} \left\| \nabla_{x_{1-\delta}} \log \frac{\mathrm{d}\nu_{1-\delta}^\star}{\mathrm{dLeb}^d} \right\|^8 \mathrm{d}\nu_{1-\delta}^\star = \mathbb{E}\left[ \left\| \mathbb{E}\left[ \frac{X_{1-\delta}^{\mathrm{I}} - \delta X_0^{\mathrm{I}} + (1-\delta)X_1^{\mathrm{I}}}{\delta(1-\delta)} \middle| X_{1-\delta}^{\mathrm{I}} \right] \right\|^8 \right]$$

$$\lesssim \mathbb{E}\left[ \left\| \frac{X_{1-\delta}^{\mathrm{I}} - \delta X_0^{\mathrm{I}} + (1-\delta)X_1^{\mathrm{I}}}{\delta(1-\delta)} \right\|^8 \right]$$

$$\lesssim \mathbf{m}_8[\nu_{1-\delta}^\star] \frac{1}{\delta^8(1-\delta)^8} + \mathbf{m}_8[\mu] \frac{1}{(1-\delta)^8} + \mathbf{m}_8[\nu^\star] \frac{1}{\delta^8}$$

$$\lesssim \mathbf{m}_8[\mu] \frac{1}{(1-\delta)^8} + \mathbf{m}_8[\nu^\star] \frac{1}{\delta^8} + d^4 \frac{1}{\delta^4(1-\delta)^4} .$$

Also, consider

$$\pi_{1-\delta}(x_0, x_{1-\delta}) = \mu(x_0) \int_{\mathbb{R}^d} p_{1-\delta|0,1}^{\mathrm{I}}(x_{1-\delta}|x_0, x_1)\nu^\star(\mathrm{d}x_1) ,$$

where $(x_0, x_1, x_{1-\delta}) \mapsto p_{1-\delta|0,1}^{\mathrm{I}}(x_{1-\delta}|x_0, x_1)$ denotes the density of $X_{1-\delta}^{\mathrm{I}}$ given $(X_0^{\mathrm{I}}, X_1^{\mathrm{I}})$ with respect to the Lebesgue measure. Then $\pi_{1-\delta} \in \Pi(\mu, \nu_{1-\delta}^\star)$ and $\pi_{1-\delta} \ll \mathrm{Leb}^{2d}$. Moreover

$$\nabla \log \left( \frac{1}{p_{1-\delta}} \frac{\mathrm{d}\pi_{1-\delta}}{\mathrm{dLeb}^{2d}} \right) \in \mathrm{L}^8(\pi_{1-\delta}) . \tag{45}$$

Indeed, because of (5),

$$p_{1-\delta|0,1}^{\mathrm{I}}(x_{1-\delta}|x_0, x_1) = \frac{1}{(4\pi\delta(1-\delta))^{d/2}} \exp\left( -\frac{\|x_{1-\delta} - \delta x_0 - (1-\delta)x_1\|^2}{4\delta(1-\delta)} \right) .$$

Therefore

$$\nabla_{x_0} p_{1-\delta|0,1}^{\mathrm{I}}(x_{1-\delta}|x_0, x_1) = \frac{x_{1-\delta} - \delta x_0 - (1-\delta)x_1}{2(1-\delta)} p_{1-\delta|0,1}^{\mathrm{I}}(x_{1-\delta}|x_0, x_1) ,$$

and

$$\nabla_{x_{1-\delta}} p_{1-\delta|0,1}^{\mathrm{I}}(x_{1-\delta}|x_0, x_1) = -\frac{x_{1-\delta} - \delta x_0 - (1-\delta)x_1}{2\delta(1-\delta)} p_{1-\delta|0,1}^{\mathrm{I}}(x_{1-\delta}|x_0, x_1) .$$

Furthermore

$$\frac{p_{1-\delta|0,1}^{\mathrm{I}}(x_{1-\delta}|x_0, x_1)\nu^\star(\mathrm{d}x_1)}{\int_{\mathbb{R}^d} p_{1-\delta|0,1}^{\mathrm{I}}(x_{1-\delta}|x_0, \tilde{x}_1)\nu^\star(\mathrm{d}\tilde{x}_1)} = \frac{p_{1-\delta|0,1}^{\mathrm{I}}(x_{1-\delta}|x_0, x_1)\mu(x_0)\nu^\star(\mathrm{d}x_1)}{\int_{\mathbb{R}^d} p_{1-\delta|0,1}^{\mathrm{I}}(x_{1-\delta}|x_0, \tilde{x}_1)\mu(x_0)\nu^\star(\mathrm{d}\tilde{x}_1)}$$

$$= p_{1|0,1-\delta}^{\mathrm{I}}(x_1|x_0, x_{1-\delta})\mathrm{d}x_1 .$$

Consequently, we have that

$$\frac{\nabla_{x_0}\pi_{1-\delta}}{\pi_{1-\delta}}(x_0, x_{1-\delta})$$

$$= \frac{\nabla\mu(x_0) \int_{\mathbb{R}^d} p_{1-\delta|0,1}^{\mathrm{I}}(x_{1-\delta}|x_0, x_1)\nu^\star(\mathrm{d}x_1) + \mu(x_0) \int_{\mathbb{R}^d} \nabla_{x_0} p_{1-\delta|0,1}^{\mathrm{I}}(x_{1-\delta}|x_0, x_1)\nu^\star(\mathrm{d}x_1)}{\mu(x_0) \int_{\mathbb{R}^d} p_{1-\delta|0,1}^{\mathrm{I}}(x_{1-\delta}|x_0, \tilde{x}_1)\nu^\star(\mathrm{d}\tilde{x}_1)}$$

$$= \frac{\nabla\mu(x_0)}{\mu(x_0)} + \int_{\mathbb{R}^d} \frac{x_{1-\delta} - \delta x_0 - (1-\delta)x_1}{2(1-\delta)} p_{1|0,1-\delta}^{\mathrm{I}}(x_1|x_0, x_{1-\delta})\mathrm{d}x_1 ,$$

and that

$$\frac{\nabla_{x_{1-\delta}}\pi_{1-\delta}}{\pi_{1-\delta}}(x_0, x_{1-\delta}) = \frac{\mu(x_0) \int_{\mathbb{R}^d} \nabla_{x_{1-\delta}} p_{1-\delta|0,1}^{\mathrm{I}}(x_{1-\delta}|x_0, x_1)\nu^\star(\mathrm{d}x_1)}{\mu(x_0) \int_{\mathbb{R}^d} p_{1-\delta|0,1}^{\mathrm{I}}(x_{1-\delta}|x_0, \tilde{x}_1)\nu^\star(\mathrm{d}\tilde{x}_1)}$$

$$= -\int_{\mathbb{R}^d} \frac{x_{1-\delta} - \delta x_0 - (1-\delta)x_1}{2\delta(1-\delta)} p_{1|0,1-\delta}^{\mathrm{I}}(x_1|x_0, x_{1-\delta})\mathrm{d}x_1 .$$

But then, if we use Jensen inequality and (43), we get

$$\int_{\mathbb{R}^{2d}} \left\| \nabla_{x_0} \log \frac{\mathrm{d}\pi_{1-\delta}}{\mathrm{d}\mathrm{Leb}^{2d}} \right\|^8 \mathrm{d}\pi_{1-\delta} \lesssim \int_{\mathbb{R}^d} \left\| \nabla_{x_0} \log \frac{\mathrm{d}\mu}{\mathrm{d}\mathrm{Leb}^d} \right\|^8 \mathrm{d}\mu$$

$$+ \mathbb{E}\left[ \left\| \mathbb{E}\left[ \left. \frac{X_{1-\delta}^{\mathrm{I}} - \delta X_0^{\mathrm{I}} - (1-\delta) X_1^{\mathrm{I}}}{1-\delta} \right| (X_0^{\mathrm{I}}, X_{1-\delta}^{\mathrm{I}}) \right] \right\|^8 \right]$$

$$\lesssim \|\nabla \log \mu\|_{\mathrm{L}^8(\mu)}^8 + \mathbb{E}\left[ \left\| \frac{X_{1-\delta}^{\mathrm{I}} - \delta X_0^{\mathrm{I}} - (1-\delta) X_1^{\mathrm{I}}}{1-\delta} \right\|^8 \right]$$

$$\lesssim \|\nabla \log \mu\|_{\mathrm{L}^8(\mu)}^8 + \mathbf{m}_8[\nu_{1-\delta}^\star] \frac{1}{(1-\delta)^8} + \mathbf{m}_8[\mu] \frac{\delta^8}{(1-\delta)^8} + \mathbf{m}_8[\nu^\star]$$

$$\lesssim \|\nabla \log \mu\|_{\mathrm{L}^8(\mu)}^8 + \mathbf{m}_8[\mu] \frac{\delta^8}{(1-\delta)^8} + \mathbf{m}_8[\nu^\star] + d^4 \frac{\delta^4}{(1-\delta)^4}$$

$$\lesssim \|\nabla \log \mu\|_{\mathrm{L}^8(\mu)}^8 + \mathbf{m}_8[\mu] \frac{1}{(1-\delta)^8} + \mathbf{m}_8[\nu^\star] \frac{1}{\delta^8} + d^4 \frac{1}{\delta^4(1-\delta)^4} ,$$

and (similarly)

$$\int_{\mathbb{R}^{2d}} \left\| \nabla_{x_{1-\delta}} \log \frac{\mathrm{d}\pi_{1-\delta}}{\mathrm{d}\mathrm{Leb}^{2d}} \right\|^8 \mathrm{d}\pi_{1-\delta} \lesssim \mathbb{E}\left[ \left\| \mathbb{E}\left[ \left. \frac{X_{1-\delta}^{\mathrm{I}} - \delta X_0^{\mathrm{I}} - (1-\delta) X_1^{\mathrm{I}}}{\delta(1-\delta)} \right| (X_0^{\mathrm{I}}, X_{1-\delta}^{\mathrm{I}}) \right] \right\|^8 \right]$$

$$\lesssim \mathbf{m}_8[\mu] \frac{1}{(1-\delta)^8} + \mathbf{m}_8[\nu^\star] \frac{1}{\delta^8} + d^4 \frac{1}{\delta^4(1-\delta)^4} .$$

Additionally, because of Remark 8 and (43), they hold

$$\int_{\mathbb{R}^{2d}} \|\nabla_{x_0} \log p_{1-\delta}(x_{1-\delta}|x_0)\|^8 \mathrm{d}\pi_{1-\delta}(x_0, x_{1-\delta})$$

$$= \int_{\mathbb{R}^{2d}} \|\nabla_{x_{1-\delta}} \log p_{1-\delta}(x_{1-\delta}|x_0)\|^8 \mathrm{d}\pi_{1-\delta}(x_0, x_{1-\delta})$$

$$= \mathbb{E}\left[ \left\| \frac{X_{1-\delta}^{\mathrm{I}} - X_0^{\mathrm{I}}}{1-\delta} \right\|^8 \right] \lesssim \mathbf{m}_8[\nu_{1-\delta}^\star] \frac{1}{(1-\delta)^8} + \mathbf{m}_8[\mu] \frac{1}{(1-\delta)^8}$$

$$\lesssim \mathbf{m}_8[\mu] \frac{\delta^8}{(1-\delta)^8} + \mathbf{m}_8[\nu^\star] + d^4 \frac{\delta^4}{(1-\delta)^4} + \mathbf{m}_8[\mu] \frac{1}{(1-\delta)^8}$$

$$\lesssim \mathbf{m}_8[\mu] \frac{1}{(1-\delta)^8} + \mathbf{m}_8[\nu^\star] \frac{1}{\delta^8} + d^4 \frac{1}{\delta^4(1-\delta)^4} .$$

It follows from **H**1, **H**2(i), (43), (44) and (45) that the probability distributions $\mu, \nu_{1-\delta}^\star$ and the coupling $\pi_{1-\delta} \in \Pi(\mu, \nu_{1-\delta}^\star)$ satisfy **H**1. The bound in Theorem 3 is now a straightforward consequence of Theorem 2 and the bounds on the scores derived so far.

