# OpenReview forum: "Theoretical guarantees in KL for Diffusion Flow Matching"
_NeurIPS.cc/2024/Conference — NeurIPS 2024 poster_

### Official Review · Reviewer_igTf · 2024-06-20

**Soundness:** 3
**Presentation:** 3
**Contribution:** 3
**Rating:** 7
**Confidence:** 3

**Summary:**

This paper proposes theoretical guarantees for diffusion flow matching, which is of the most popular generative models. Specifically, it provides an upper bound of the KL divergence between the target distribution and the one learned by the DFM. This work promotes the theoretical understanding of the discretized flow matching model.

**Strengths:**

1. The paper is clearly written, and the proofs seem to be sound.

2. The authors consider the early-stopping case and remove the Lipschitz assumption of the velocity field.

**Weaknesses:**

1. Assumption H1 is quite strong that requires both the source and target distributions to have finite 8-th moments.

2. The procedure is standard compared to some theoretical work on SGM. For example, the authors assume an L2-drift-approximation error and use Girsanov theorem for bounding the KL divergence.

**Questions:**

1. If you in addition assume the velocity field is Lipschitz, will you be able to relax assumption H1 to have finite second moments?

**Limitations:**

Yes.

---

> ### Author Rebuttal · Authors · 2024-08-05
>
> Questions:
>
> _If you in addition assume the velocity field is Lipschitz, will you be able to relax assumption H1 to have finite second moments?_
>
> We thank the reviewer for highlighting this issue. Indeed, for a Lipschitz velocity field, assumption H1 can be relaxed to only require finite second moments. This can be achieved by using the Lipschitz condition to bound the second term in equation (23) and employing the linear representation in equation (5) of the stochastic interpolant.
> However, assuming that the velocity field is Lipschitz would come at a heavy price on the data distribution and the coupling. This is certainly not true under the assumptions of our work regardless of whether or not we are in the early stopping regime. One would need to add stronger assumptions.
>
> Weaknesses:
>
> _Assumption H1 is quite strong that requires both the source and target distributions to have finite 8-th moments._
>
> We consider assumption H1 to be relatively mild, as in most applications, data distributions are compactly supported. Moreover, our paper presents the first convergence analysis for diffusion-type FMs. In contrast, early studies on SGMs required much stronger assumptions on data distributions, beyond the Lipschitz continuity of the score, as seen in [11], [12], and [14]. Furthermore, many of these studies did not achieve convergence bounds with polynomial complexity. While recent convergence guarantees for SGMs work under the finiteness of the second-order moment in the early stopping regime, it should be noted that analyzing DFMs is more challenging, as, contrary to SGMs, DFMs permit to interpolate between any two probability distributions in finite time. We appreciate the reviewer's insights and acknowledge that there is potential for improvements, albeit at the expense of additional technical complexity.
>
> [11] V. D. Bortoli, Convergence of denoising diffusion models under the manifold hypothesis.
>
> [12] V. De Bortoli, J. Thornton, J. Heng, and A. Doucet, Diffusion schrödinger bridge with applications to score-based generative modeling.
>
> [14] A. Block, Y. Mroueh, and A. Rakhlin, Generative modeling with denoising auto-encoders and langevin sampling.
>
> _The procedure is standard compared to some theoretical work on SGM. For example, the authors assume an L2-drift-approximation error and use Girsanov theorem for bounding the KL divergence._
>
> Concerning the assumption on the L2-drift-approximation error, we believe that analyzing the convergence of the proposed DFM model under such a standard assumption is a strength of our work, not a weakness. This demonstrates that our bound is applicable under reasonable conditions.
>
> Additionally, we assert that our procedure is fundamentally different from those proposed in previous works. While the application of the Girsanov theorem is now standard in the literature (see [9,10,11,12]), it is not a distinguishing feature of our approach, but rather a starting point. Beyond that, our method is entirely distinct and is not a straightforward consequence of the Girsanov theorem and assumption H3. The main innovation of the present article consists in using the Markovian projection of the interpolant as a reference measure and being able to control the $L^2$ norm of the adjoint process in the Pontryagin system associated with it. We do so by introducing a novel quantity in the generative model literature (see [13]), namely the so-called “reciprocal characteristic” of a Markov process, which may be viewed as some sort of mean acceleration field. The main efforts in our proofs are directed towards bounding the $L^2$ norm of the reciprocal characteristic whose representation in terms of either conditional moments or higher-order logarithmic derivatives of conditional densities is quite intricate, see page 26. Trying to bound each of these terms separately requires strong assumptions on the initial distributions and couplings leading to suboptimal results. However, using integration by parts both in time and space, and profiting from symmetry properties of the heat kernel, we managed to bound these terms under assumptions comparable to the minimal ones required in the analysis of SGMs. Note that the analysis of the reciprocal characteristic is not required for SGMs (it is always 0) and that controlling it also requires new tricks and ideas, since its representation contains up to three logarithmic derivatives of conditional distributions, whereas the analysis of SGMs requires at most two such derivatives to be analyzed, or, equivalently, to bound conditional covariances.
>
> Based on the reviewer’s feedback, it appears that we haven’t sufficiently highlighted the novelty of our procedure compared to existing studies. We will make it a priority to clarify and emphasize this aspect more clearly in “the proposed methodology”, starting with adding part of the previous paragraphs.
>
>
> [9] Giovanni Conforti, Alain Durmus, and Marta Gentiloni Silveri. Score diffusion models without early stopping: finite fisher information is all you need.
>
> [10] Hongrui Chen, Holden Lee, and Jianfeng Lu. Improved analysis of score-based generative modeling: User-friendly bounds under minimal smoothness assumptions.
>
> [11] V. D. Bortoli, Convergence of denoising diffusion models under the manifold hypothesis.
>
> [12] V. De Bortoli, J. Thornton, J. Heng, and A. Doucet, Diffusion schrödinger bridge with applications to score-based generative modeling.
>
> [13] A. Krener, Reciprocal diffusions in flat space.

---

> > ### Comment · Reviewer_igTf · 2024-08-07
> > **Response to the rebuttal**
> >
> > I appreciate the authors' comprehensive response. My questions and concerns are sufficiently addressed. Thus, I would like to increase the rating.

---

> > > ### Author Response · Authors · 2024-08-08
> > >
> > > We thank the reviewer for considering our response.

---

### Official Review · Reviewer_83K8 · 2024-07-12

**Soundness:** 3
**Presentation:** 2
**Contribution:** 2
**Rating:** 5
**Confidence:** 3

**Summary:**

This paper gives theoretical guarantees for Diffusion Flow Matching similar in spirit to existing results for Denoising Diffusion Probabilistic Models and Probability Flow ODE. This work requires weaker assumptions than prior work, replacing the Lipschitzness of the score function with a relative Fisher information condition.

**Strengths:**

This paper weakens this required assumptions of prior work, and is the first to develop guarantees for diffusion flow matching, instead of, for example, an analysis of DDPM or probability flow ODE. Unlike these other works, the authors further exploit properties of the heat kernel that appears in the estimation procedure. Their results also easily extend to the early-stopping setting.

**Weaknesses:**

One could make the argument that the present ideas are not as novel as its predecessors (e.g., Conforti et al. (2023) for the finite-fisher information criteria, or the line of works similar to Chen et al. (2023) in general). Moreover, the dimension-dependence in these results is quite poor (though the authors do mention this as an avenue for future work).

**Questions:**

- The authors claim to tackle "all sources of error" in the abstract, though I would argue the statistical convergence rates remain open.
- What is the primary obstacle in improving the dimension dependence? Do you conjecture that it should be the same for DDPM?

Some typos I managed to catch and write down:
- L28: "are important... " or "were an important"
- L34: I don't know what the ":" is there
- L63: "In the case where..." maybe?
- L73: Reference to LWYql23 --- might be worth fixing the authors name in the bibliography
- L77: "... an approximate sample ..."
- L144: "... a solution of the ..."
- L164: "... is therefore the solution ..." [the subsequent page has a few more instances of missing articles that are worth fixing]
- L170: "through" (not trough)
- L173: Fix the [Kry] citation (no year)
-  L190: "...gives an ideal..."

---

> ### Author Rebuttal · Authors · 2024-08-05
>
> Questions:
>
> _The authors claim to tackle "all sources of error" in the abstract, though I would argue the statistical convergence rates remain open._
>
> In our defense, we thought we have already acknowledged the lack of analysis on statistical convergence rates. In the conclusion, we suggest that “it would be interesting to complement our analysis with a statistical analysis of DFM.” Furthermore, while this is not mentioned in the abstract, we clarify in both the introduction and conclusion that "all sources of error" refer to the “drift-approximation error and the time-discretization error.” It is also important to note that, with the exception of [7] none of the previous works on FMs address the issue of statistical convergence rates. However, given the reviewer’s comment, it seems we have not been clear enough. Therefore, we will specify in the introduction that our work does not cover the statistical error of DFM.
>
> [7] Yuan Gao, Jian Huang, Yuling Jiao, and Shurong Zheng. Convergence of continuous normalizing flows for learning probability distributions.
>
> _What is the primary obstacle in improving the dimension dependence? Do you conjecture that it should be the same for DDPM?_
>
> The primary obstacle stems from the challenge of bounding the reciprocal characteristic associated with the mimicking drift, as detailed in our response to the “Weaknesses” section. However, it is important to highlight that our bound depends polynomially on the dimension, unlike previous bounds (see [3], [8])  which depend exponentially on it. Consequently, our dimension-dependence is significantly more favorable. Nonetheless, we suspect that improvements could be made at the cost of additional technical complexity and of more sophisticated choices of discretization schemes. Therefore, we mention this as a potential avenue for future work in the conclusion.
>
> [3] Michael S. Albergo and Eric Vanden-Eijnden. Building normalizing flows with stochastic interpolants.
>
> [8] Joe Benton, George Deligiannidis, and Arnaud Doucet. Error bounds for flow matching methods.
>
> Weaknesses:
>
> _One could make the argument that the present ideas are not as novel as its predecessors (e.g., Conforti et al. (2023) for the finite-fisher information criteria, or the line of works similar to Chen et al. (2023) in general). Moreover, the dimension-dependence in these results is quite poor (though the authors do mention this as an avenue for future work)._
>
> We believe that our approach contains fresh ideas that cannot be found in previous works dealing with SGMs, and is entirely distinct from those of [9] and [10].
>
> The sole resemblance to [10] lies in the application of the Girsanov theorem, which however is not the distinguishing feature of our work, but simply a starting point of our proof. In addition, this step is not a contribution of [10] and is now standard for analyzing SGMs (see e.g., [11,12]).
>
> Regarding [9], one of its main contributions is to estimate relative entropy on path space by interpreting (a modified version of) the drift of the backward process as the adjoint process in a stochastic control problem. In the context of this work, the connection with Markovian stochastic control is broken essentially because of the non-Markovian nature of stochastic interpolants. Our idea for this work was to partially restore it by using the Markovian projection of the interpolant as a reference measure. Then, the main innovation of the present article consists in being able to control the $L^2$ norm of the adjoint process in the Pontryagin system associated with the Markovian projection of the interpolant. We do so by introducing a novel quantity in the generative model literature (see [13]), namely the so-called “reciprocal characteristic” of a Markov process, which may be viewed as some sort of mean acceleration field. The main efforts in our proofs are directed towards bounding the $L^2$ norm of the reciprocal characteristic whose representation in terms of either conditional moments or higher-order logarithmic derivatives of conditional densities is quite intricate, see page 26. Trying to bound each of these terms separately requires strong assumptions on the initial distributions and couplings leading to suboptimal results. However, using integration by parts both in time and space and a double change of measure argument, and profiting from symmetry properties of the heat kernel, we managed to bound these terms under assumptions comparable to the minimal ones required in the analysis of SGMs. Note that the analysis of the reciprocal characteristic is not required for SGMs (it is always 0) and that controlling it also requires new tricks and ideas, since its representation contains up to three logarithmic derivatives of conditional distributions, whereas the analysis of SGMs requires at most two such derivatives to be analyzed.
>
> Based on the reviewer’s feedback, it appears that we haven’t sufficiently highlighted the novelty of our procedure compared to existing studies. We will make it a priority to clarify and emphasize this aspect more clearly in “the proposed methodology”, starting with adding part of the previous paragraphs.
>
>
> [9] Giovanni Conforti, Alain Durmus, and Marta Gentiloni Silveri. Score diffusion models without early stopping: finite fisher information is all you need.
>
> [10] Hongrui Chen, Holden Lee, and Jianfeng Lu. Improved analysis of score-based generative modeling: User-friendly bounds under minimal smoothness assumptions.
>
> [11] V. D. Bortoli, Convergence of denoising diffusion models under the manifold hypothesis.
>
> [12] V. De Bortoli, J. Thornton, J. Heng, and A. Doucet, Diffusion schrödinger bridge with applications to score-based generative modeling.
>
> [13] A. Krener, Reciprocal diffusions in flat space.

---

### Official Review · Reviewer_TMKS · 2024-07-14

**Soundness:** 4
**Presentation:** 4
**Contribution:** 4
**Rating:** 8
**Confidence:** 2

**Summary:**

This work provides theoretical guarantees for diffusion flow matching (DFM) models, which are a recent class of generative models similar to score-based generative models (SGM). Extensive background is given in sections 1 and 2, and section 3 contains the results, namely, bounds on the KL divergence from the target distribution to the distribution outputted by DFM. A detailed comparison with existing results is presented in section 3.2.

**Strengths:**

The presentation of the whole paper is remarkably clear, notably the background explanations of section 2 are very clear and very welcome. Regarding the guarantees of section 3, they are (claimed to be) the first ones for DFMs, which are well-motivated as alternatives to SGMs and deterministic FMs; this makes these results quite valuable.

**Weaknesses:**

(I can only comment on the presentation and context as I am not familiar with the guarantees for SGMs and FMs, sorry.)
- Two things remained unclear to me after reading section 2; see "Questions".
- The discussion of the previous work on SGMs seems a bit difficult to compare, since this paper is about DFMs, which (if I understand) are a different thing although similar.

**Questions:**

- Are SGMs a sub-case of DFMs, for a well-chosen coupling and bridge, and perhaps up to a change of time variable?
- Is the Markovian projection of an Ito process always unique, and/or is the Markovian projection of (3) unique?
- How do DFMs compare to SGMs and deterministic FMs in practice, say in generalization, speed, difficulty of implementation, robustness w.r.t. hyperparameters, ...?
- Are there settings where it makes sense to want to choose something else than $\pi$ being the independent coupling and the bridge being Brownian bridge?

**Limitations:**

The authors have adequately addressed the limitations.

---

> ### Author Rebuttal · Authors · 2024-08-05
>
> Questions:
>
> _Are SGMs a sub-case of DFMs, for a well-chosen coupling and bridge, and perhaps up to a change of time variable?_
>
> We thank the reviewer for raising this point. As suggested by the reviewer, after an appropriate time transformation ($t=\exp(-\tau)$), the Ornstein–Uhlenbeck (OU) process at the basis of SGMs can be reformulated as the process at the basis of a linear stochastic interpolant. For further insights, see Section 5 of [1]. However, the foundational concepts of SGMs and DFMs differ significantly: SGMs aim to approximate the time-reversal of a diffusion process (most often OU), whereas DFMs aim to approximate a Markovian projection of a two-sided linear stochastic interpolant.
>
> [1] Michael S. Albergo, Nicholas M. Boffi, and Eric Vanden-Eijnden. Stochastic Interpolants: A Unifying Framework for Flows and Diffusions.
>
> _Is the Markovian projection of an Ito process always unique, and/or is the Markovian projection of (3) unique?_
>
> The uniqueness of the Markovian projection of an Itô process, such as the one defined by equation (3), is not straightforward and necessitates regularity assumptions on the mimicking drift, such as those found in [2]. In the specific case of interest in our paper, we suspect that the well-posedness of equation (9) holds. We appreciate the reviewer for highlighting this interesting issue.
>
> [2] Stroock, D. W. and Varadhan, S. R. S.. Diffusion processes with continuous coefficients
>
> _How do DFMs compare to SGMs and deterministic FMs in practice, say in generalization, speed, difficulty of implementation, robustness w.r.t. hyperparameters, ...?_
>
> In Section 3.4 of [3], the authors include some results for unconditional image generation, comparing the proposed interpolant flow model with diffusion methods on CIFAR10 and the ImageNet dataset. Their models emit likelihoods, measured in bits per dim, that are competitive with state-of-the-arts diffusion models on both datasets. Similarly, Frechet Inception Distance scores are proximal to those from diffusions, though slightly behind the best results. Finally, in [1, Section 7], the authors conducted experiments indicating that stochastic interpolants generally outperform their deterministic counterparts.
>
> [1] Michael S. Albergo, Nicholas M. Boffi, and Eric Vanden-Eijnden. Stochastic Interpolants: A Unifying Framework for Flows and Diffusions.
>
> [3] Michael S. Albergo and Eric Vanden-Eijnden. Building normalizing flows with stochastic interpolants.
>
> _Are there settings where it makes sense to want to choose something else than 𝜋 being the independent coupling and the bridge being Brownian bridge?_
>
> We thank the reviewer for raising this interesting question. Our primary motivation for choosing the independent coupling in the early-stopping regime and the Brownian bridge was simplicity, as analyzing non-independent couplings or non-Brownian bridges would have been significantly more complex. Nonetheless, we believe that, in the early stopping regime, non-independent couplings could be considered as well at the cost of additional technical complexity. Moreover, in practice, the independent coupling and the Brownian bridge are one of the most common choices due to their ease of implementation. However, the independent coupling is not always the most optimal choice, and there have been works highlighting the benefits of using other couplings, such as [4]. Additionally, we can mention Rectified Flows [5] and Diffusion Schrödinger Bridge Matching (DSBM) [6], which essentially involve iterating FM or DFM and allow for defining a sequence of coupling distributions expected to be more efficient than the independent coupling.
>
> [4] A. Tong et al. Improving and generalizing flow-based generative models with minibatch optimal transport.
>
> [5] X. Liu, C. Gong, and Q. Liu. Flow straight and fast: learning to generate and transfer data with rectified flow.
>
> [6] Y. Shi, V. De Bortoli, A. Campbell A. Doucet. Diffusion Schrödinger Bridge Matching
>
> Weaknesses:
>
> _Two things remained unclear to me after reading section 2; see "Questions"._
>
> See "Questions".
>
> _The discussion of the previous work on SGMs seems a bit difficult to compare, since this paper is about DFMs, which (if I understand) are a different thing although similar._
>
> Due to the connection between SGMs and DFMs we believed that a comparison with studies on SGMs would be of interest to our readers. However, if the reviewers and the AC find this comparison irrelevant to our work, we are willing to shorten this section of the paper accordingly.

---

> > ### Comment · Reviewer_TMKS · 2024-08-08
> >
> > Thank you for your response. To improve the presentation, I would recommend adding a few words early in the paper on the fact that SGMs are a sub-case of DFMs. I maintain my rating (and low confidence score).

---

### Official Review · Reviewer_NqRW · 2024-07-15

**Soundness:** 3
**Presentation:** 3
**Contribution:** 4
**Rating:** 7
**Confidence:** 3

**Summary:**

The paper derives theoretical guarantees for a flow matching procedure which constructs a diffusion-like process which allows for samples obtained from a source distribution $\mu$ to a target distribution $\nu^*$. The goal is to build a suitable simple sampling procedure where a source sample is updated through a series of relatively simple Markovian procedure to an approximate sample from $\nu^*$. In the typical setting of diffusion models, this takes the form of a score function augmented with additional Gaussian noise. However, this suffers from two main drawbacks -- firstly, the choice of base distribution is fixed (the base function is always Gaussian) and secondly, the diffusion is truncated after a finite number of timesteps which incurs error in the approximation of the intermediate distribution by a Gaussian. Flow matching addresses both of these by allowing for transformations of arbitrary distributions to each other.

To define a flow-matching process as in the paper, one obtains samples $(X, Y)$ generated from a coupling (joint distribution) $\pi (X, Y)$ where the marginals of $X$ and $Y$ are $\mu$ and $\nu^*$ (the source and target distribution respectively). Given these samples, an interpolating distribution, $\{X_t\}_{t \in [0, 1]}$, is constructed with $X_0 \coloneqq X$ and $X_1 \coloneqq Y$ and the intermediate values are distributed based on the choice of an appropriate bridge distribution. This paper utilizes the Brownian bridge which is the conditional distribution of Brownian motion conditioned on fixing its endpoints. The main issue with this approach is that such an interpolating distribution is not \emph{Markovian} which makes sampling from it in a diffusion-like process challenging. This may be addressed through the notion of
a Markovian projection which constructs another \emph{Markovian} process with the same marginals as the interpolating process for which a score-function-like analogue which enables efficient Markovian sampling. The paper then shows that under some relatively mild assumptions on the approximation of the score function and the moments of the coupled distributions, this procedure guarantees closeness to the target distribution in Total Variation distance. All previous approaches either require additional assumptions on the estimated score function, and/or do not account for the discretization error. One of the main challenges of the proof is in analyzing the discretization error of the scheme where the continuous process is approximated with one with discrete time steps. The paper observes that the terms controlling this error may be rather large at the end points of the interval. Therefore, the paper considers a truncated process on $[\delta, 1 - \delta]$ and show that on the truncated set, the gradients of the densities with control the discretization error are bounded in ($L^8$) norm.

Overall, the results of the paper are novel and interesting. The observation that the truncated distribution is well-behaved with respect to the discretization error is valuable for future theoretical and practical work on flow models. This also illustrates the drawbacks of prior work which require additional Lipschitz assumptions on the scores/estimated scores and presents a natural approach towards circumventing them. This is a nice contribution and I recommend acceptance.

**Strengths:**

See main review

**Weaknesses:**

See main review

**Questions:**

See main review

**Limitations:**

See main review

---

> ### Author Rebuttal · Authors · 2024-08-05
>
> We sincerely thank the reviewer for the thorough review and valuable comments. We greatly appreciate the reviewer’s recognition of the novelty and significance of our work, as well as the potential of our ideas for future theoretical and practical advancements in flow models.

---

> > ### Comment · Reviewer_NqRW · 2024-08-08
> >
> > I acknowledge the authors' rebuttal and will retain my current evaluation.

---

### Author Rebuttal · Authors · 2024-08-05

We sincerely thank the reviewers for their positive evaluation of our work and their valuable feedback. Their constructive comments will undoubtedly help enhance our original work, and we are committed to incorporating the suggested modifications. The reviewers acknowledge that our work advances the field of diffusion flow matching models by providing the first convergence guarantees, removing the Lipschitz assumption on the velocity field, and addressing the early-stopping regime. While the novelty of our results is recognized, two reviewers observed some similarities in our methodology with previous studies. However, the use of standard tools, such as the decomposition of the Kullback-Leibler Divergence based on Girsanov theorem, is not the distinguishing feature of our work, which is rather the strategy used to bound the reciprocal characteristic associated with the Markovian projection. We will make every effort to clearly differentiate our procedure from existing research and highlight its originality. Below, we address each reviewer’s comments individually. We will be happy to answer any additional questions during the author-reviewer discussion period.

---

### Comment · Area_Chair_U9Fa · 2024-08-14
**Discussions**

Dear reviewers,

Thanks very much for your great efforts in the review process. Please read the authors' response and confirm it.  You are encouraged to discuss with all of us if you have any concerns.

Thanks,
AC

---

### Decision · Program_Chairs · 2024-09-25

**Decision:**

Accept (poster)

**Comment:**

Most of the reviewers think the paper is interesting.  There seems to be something new in the proof.  I recommend an acceptance.